# Learning properties of quantum states without the IID assumption

Omar Fawzi[1], Richard Kueng[2], Damian Markham [3] & Aadil Oufkir [1,4] ✉

We develop a framework for learning properties of quantum states beyond the assumption of independent and identically distributed (i.i.d.) input states. We prove that, given any learning problem (under reasonable assumptions), an algorithm designed for i.i.d. input states can be adapted to handle input states of any nature, albeit at the expense of a polynomial increase in training data size (aka sample complexity). Importantly, this polynomial increase in sample complexity can be substantially improved to polylogarithmic if the learning algorithm in question only requires non-adaptive, single-copy measurements. Among other applications, this allows us to generalize the classical shadow framework to the non-i.i.d. setting while only incurring a comparatively small loss in sample efficiency. We leverage permutation invariance and randomized single-copy measurements to derive a new quantum de Finetti theorem that mainly addresses measurement outcome statistics and, in turn, scales much more favorably in Hilbert space dimension.

The advent of quantum technologies has led to a notable amount of tools for quantum state and process learning. These are employed as tools within use cases, but also to test applications and devices themselves. However, almost all existing methods require the assumption that the devices or states being tested are prepared in the same way over time – following an identical and independent distribution (i.i.d.)[1–10]. In various situations, this assumption should not be taken for granted. For instance, in time correlated noise, states and devices change in time in a non-trivial way[11–13]. Moreover, in settings where we cannot trust the devices or states – for example, originating from an untrusted, possibly malicious manufacturer, or states that are distributed over untrusted channels – the assumption of i.i.d. state preparations can be exploited by malicious parties to mimic good behavior whilst corrupting the intended application. Avoiding this assumption is crucial for various applications such as verified quantum computation[14] or tasks using entangled states in networks[15], such as authentication of quantum communication[16], anonymous communication[17], or distributed quantum sensing[18]. At the core of the security for these applications is some verification procedure which does not assume i.i.d. resources, however they are all catering for particular states or processes, with independent proofs and with differing efficiencies.

The main contribution of this paper is to develop a framework to extend existing i.i.d. learning algorithms into a fully general (non-i.i.d.) setting while preserving rigorous performance guarantees. See Theorem 1 and Theorem 3 for the type of results we provide. The main technical ingredient is a variant of the quantum de Finetti theorem for randomized permutation invariant measurements (See Theorem 2). As a concrete example, we apply our findings to the task of feature prediction with randomized measurements (classical shadows)[7,19,20] (See Proposition 1). We then apply these results to the problem of state verification, allowing us to find the first explicit protocol for verifying an arbitrary multipartite state, showing the power of these techniques.

## Results

In the following, we start by showing how to evaluate an algorithm in the non-i.i.d. setting. Then, we show that, in principle, general algorithms can be adapted to encompass non-i.i.d. input states at the expense of an overhead in the copy complexity. Next, we reduce significantly this overhead for incoherent non-adaptive algorithms using our quantum de Finetti theorem. Finally, we apply this extension to the problems of classical shadows and verification of pure states in the non-i.i.d. setting.

[1]Inria, ENS Lyon, UCBL, LIP, Lyon, France. [2]Department of Quantum Information and Computation at Kepler (QUICK), Johannes Kepler University Linz, Linz, Austria. [3]Sorbonne Université, CNRS, Paris, France. [4]Institute for Quantum Information, RWTH Aachen University, Aachen, Germany. ✉e-mail: oufkir@physik.rwth-aachen.de

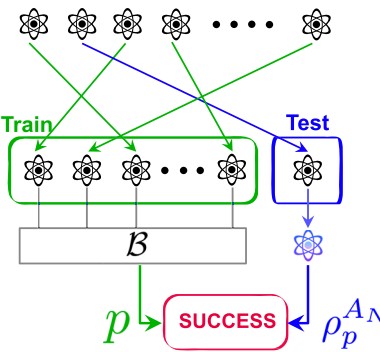

**Fig. 1 | Illustration of a general state learning algorithm.** A learning algorithm consumes $(N-1)$ copies of $\rho$ to construct a prediction $p$. Success occurs if $p$ is (approximately) compatible with the remaining post-measurement test copy $\rho_p^{A_N}$.

## Evaluating a learning algorithm

The first difficulty we face is to define what it means for a learning algorithm to achieve some learning task on a non-i.i.d. state. In the i.i.d. setting, a learning algorithm requests $N$ copies of an unknown quantum state and is provided with the quantum state $\rho = \sigma^{\otimes N} \in (\mathbb{C}^{d \times d})^{\otimes N}$. Subsequently, the learning algorithm makes predictions about a property of the quantum state $\sigma$. This algorithm is evaluated by contrasting its predictions with the actual property of the quantum state $\sigma$. To motivate our general definition, we imagine a black box from which we can request copies. On the first query, we receive a system that we call $A_1$ and on the $k$th query, we receive the system $A_k$. Learning means making a statement about some of the outputs of the black box (e.g., the state is close to $|0\rangle$). With the i.i.d. assumption, the black box always outputs the same state. Removing the i.i.d. assumption, the learning algorithm is presented with a general quantum state $\rho \in (\mathbb{C}^{d \times d})^{\otimes N}$ where $N$ is the number of requested copies. In this case, we have to specify the system about which we make the statement (this is the system that would be used for a later application for example). The most natural choice is to take a system at random among the ones that were requested. In other words, we use the common idea in machine learning of separating the data set (here the $N$ systems that we denote $A_1, ..., A_N$) into a training set used for estimation and a test set used for evaluation. We refer to Fig. 1 for a visual illustration. This idea was previously used in the context of quantum tomography[2], verification[21], and generalization bounds[22].

The choice of which systems are used for training and which are used for testing is random. More specifically, we apply a random permutation (that the learner does not have access to) to the systems $A_1...A_N$ and we fix the training set to be the first $N-1$ systems and the test set is composed of the last system. Thus, starting with the general state $\rho^{A_1 \cdots A_N}$, we obtain after the random permutation a state that we denote $\overline{\rho}^{A_1 \cdots A_N}$. Written explicitly

$$\overline{\rho}^{A_1 \cdots A_N} = \frac{1}{N!} \sum_{\pi \in \mathfrak{S}_N} \rho^\pi,$$

where $\mathfrak{S}_N$ denotes the set of permutations and $\rho^\pi$ is obtained by permuting the systems $A_1...A_N$ of $\rho$ according to $\pi$. The learning algorithm $\mathcal{B}$ is applied to the training set $A_1...A_{N-1}$ and makes a prediction that we denote $p$ and we test this prediction against the system $A_N$. The learning task will be described by a family of sets $\text{SUCCESS}_\varepsilon$ where $\varepsilon$ should be seen as a precision parameter. The pair $(p, \sigma) \in \text{SUCCESS}_\varepsilon$ if prediction $p$ is correct for the state $\sigma$ with precision $\varepsilon$. As an example, for the task of predicting $M$ observables $O_1, ..., O_M$ (shadow tomography), we would have

$\mathbf{p} = (p_1, ..., p_M) \in [0, 1]^M$ and

$$\text{SUCCESS}_\varepsilon = \{((p_1, \ldots, p_M), \sigma) : \forall i \in [M], |p_i - \text{Tr}[O_i \sigma]| \leq \varepsilon\}.$$

Note that this is precisely the learning task which has motivated (i.i.d.) classical shadows[7,20].

We evaluate a learning algorithm $\mathcal{B}$ for the task described by $\text{SUCCESS}_\varepsilon$ on the input state $\rho^{A_1 \cdots A_N}$ as follows. The algorithm $\mathcal{B}$ takes as input the systems $A_1...A_{N-1}$ and outputs a prediction $p \in \mathcal{P}$ and a calibration information $c \in \mathcal{C}$. The role of the calibration information is to determine the reduced state of $A_N$ and can range from trivial $\emptyset$ to all measurement outcomes. In other words, $(c, p)$ follows the distribution $\mathcal{B}(\overline{\rho}^{A_1 \cdots A_{N-1}})$, which we denote by: $(c, p) \sim \mathcal{B}(\overline{\rho}^{A_1 \cdots A_{N-1}})$. For an outcome $(c, p)$, we write $\rho_{c,p}^{A_N}$ for the reduced state of $A_N$ of the state $(\mathcal{B} \otimes \text{id})(\overline{\rho}^{A_1 \cdots A_N})$ conditioned on the outcome of $\mathcal{B}$ being $(c, p)$. Finally, we define

$$\delta_{\mathcal{B}}(N, \rho^{A_1 \cdots A_N}, \varepsilon) = \mathbb{P}_{(c,p) \sim \mathcal{B}(\overline{\rho})}\left[\left(p, \overline{\rho}_{c,p}^{A_N}\right) \notin \text{SUCCESS}_\varepsilon\right]. \quad (1)$$

We make a few remarks about this definition assuming $c = \emptyset$ for simplicity. First, in the i.i.d. setting we have that $\rho_p^{A_N} = \sigma$ for any $p \in \mathcal{P}$ and we recover the usual definition of error probability. Second, note that it is essential to consider the state of $A_N$ conditioned on the outcome $p$. One might be tempted to replace $\overline{\rho}_p^{A_N}$ with the marginal $\overline{\rho}^{A_N}$ but this would be both unachievable and undesirable. In fact, consider the simple example $\rho = \frac{1}{2}(|0\rangle\langle 0|^{\otimes N} + |1\rangle\langle 1|^{\otimes N})$ and we would like to estimate the value of the observable $O = |1\rangle\langle 1|$. Note that $\mathbb{P}_{p \sim \mathcal{B}(\overline{\rho})}[\cdot] = \frac{1}{2}\mathbb{P}_{p \sim \mathcal{B}(|0\rangle\langle 0|^{\otimes N})}[\cdot] + \frac{1}{2}\mathbb{P}_{p \sim \mathcal{B}(|1\rangle\langle 1|^{\otimes N})}[\cdot]$. As such, with the naive definition using the marginal $\overline{\rho}^{A_N}$ which is $\mathbb{I}/2$ in this case, the error probability would be given by $\frac{1}{2}\mathbb{P}_{p \sim \mathcal{B}(|0\rangle\langle 0|^{\otimes N})}[|\frac{1}{2} - p| > \varepsilon] + \frac{1}{2}\mathbb{P}_{p \sim \mathcal{B}(|1\rangle\langle 1|^{\otimes N})}[|\frac{1}{2} - p| > \varepsilon]$. Clearly any good learning algorithm should work for the i.i.d. states $|0\rangle\langle 0|^{\otimes N}$, $|1\rangle\langle 1|^{\otimes N}$ and this implies that the error probability $\delta_{\mathcal{B}}(N, \rho, \varepsilon)$ is close to 1 for this choice of $\rho$. For this example, it is desirable that the learning algorithm first detects which of the two states $|0\rangle\langle 0|^{\otimes N}$ or $|1\rangle\langle 1|^{\otimes N}$ has been prepared and then learns the state consistently. This is captured by the definition (1).

A third remark about the definition we use is that the error probability is evaluated for the averaged state $\overline{\rho}$, or in other words the learner does not have access to the randomly chosen permutation $\pi$. Another possibility would be to define the error probability as an average over permutations $\pi$ of the error probability evaluated for the permuted state $\rho^\pi$, i.e.,

$$\delta'_{\mathcal{B}}(N, \rho^{A_1 \cdots A_N}, \varepsilon) = \mathbb{E}_\pi\left[\mathbb{P}_{(c,p) \sim \mathcal{B}(\rho^\pi)}\left[\left(p, (\rho^\pi)_{c,p}^{A_N}\right) \notin \text{SUCCESS}_\varepsilon\right]\right]. \quad (2)$$

It turns out that this definition renders learning impossible in many cases. In fact, we show in Supplementary Note 1 that for the simplest possible classical task of estimating the expectation of a binary random variable, it is not possible to achieve $\delta'_{\mathcal{B}} < 1/4$ for all states. This shows that requiring $\delta'_{\mathcal{B}}$ to be small cannot be achieved in general and it justifies our choice in Eq. (1). We also remark that for verification problems, where the prediction is of the form Accept/Reject and we only want to express the soundness condition for all states in expectation, then the expression for the error probability is linear in the state (see Supplementary Note 4). As such, in this case, whether the permutation is available to the learner or not does not make a difference. With our definition, we have $\delta_{\mathcal{B}}(N, \rho, \varepsilon) = \delta_{\mathcal{B}}(N, \overline{\rho}, \varepsilon)$, so to make the notation lighter, we assume in the rest of the paper that $\rho$ is permutation invariant, i.e., $\overline{\rho} = \rho$.

## Adapting a learning algorithm designed for i.i.d. inputs

Our first result transforms any learning algorithm $\mathcal{A}$ for the task $\text{SUCCESS}_\varepsilon$ designed for i.i.d. input states to a learning algorithm $\mathcal{B}$ for

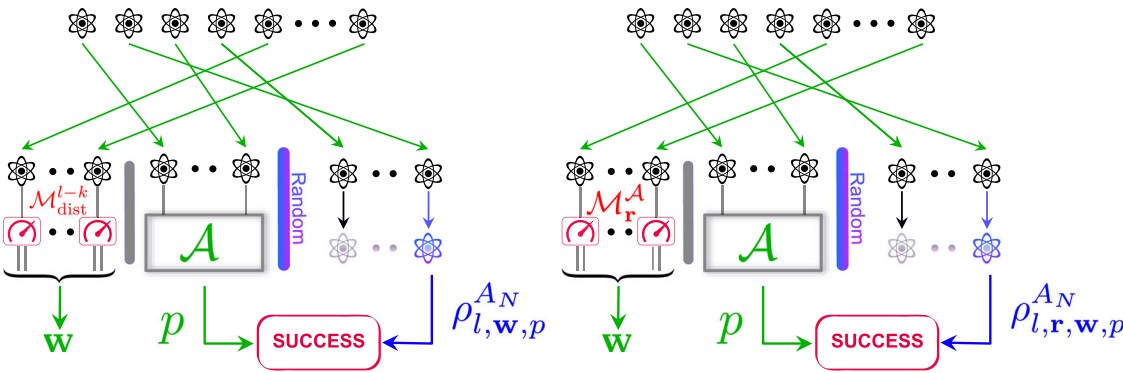

**Fig. 2 | Caricature of main results: how to lift an i.i.d. learning algorithm $\mathcal{A}$ beyond the i.i.d. setting.** Left: the performance of general learning algorithms is covered by our first main result (Theorem 1). Right: the performance of non-adaptive and incoherent learning algorithms is covered by our second main result (Theorem 3). Restricting to non-adaptive and incoherent measurement $\mathcal{M}_{\mathbf{r}}$ leads to much better theoretical performance guarantees. $\mathcal{M}_{\text{dist}}$ is a measurement device with low distortion, $\mathbf{w}$ is calibration, $p$ is prediction, $\mathcal{A}$ is the data processing of the i.i.d. algorithm and $\mathcal{M}_{\mathbf{r}}^{\mathcal{A}}$ is a measurement device uniformly chosen from $\mathcal{A}$'s set of measurements. Success occurs if $p$ is (approximately) compatible with the remaining post-measurement test copies $\rho_{l,\mathbf{w},p}^{A_N}$ or $\rho_{l,\mathbf{r},\mathbf{w},p}^{A_N}$.

the same task without requiring the i.i.d. assumption at the cost of an increased number of queries.

**Theorem 1.** (General algorithms in the non-i.i.d. setting). Let $\varepsilon > 0$, $1 \leq k < N/2$ and $d$ be the dimension of the Hilbert spaces $A_1, ..., A_N$. Let $\mathcal{A}$ be a learning algorithm designed for i.i.d. input states. There exists a learning algorithm $\mathcal{B}$ taking arbitrary inputs on $N$ systems and having an error probability (1) satisfying

$$\delta_{\mathcal{B}}(N, \rho^{A_1 \cdots A_N}, 2\varepsilon) \leq \sup_{\sigma:\text{state}} \delta_{\mathcal{A}}\left(k, \sigma^{\otimes k}, \varepsilon\right) + \mathcal{O}\left(\sqrt{\frac{k^3 d^2 \log(d)}{N\varepsilon^2}}\right).$$

Note that the evaluation of a learning algorithm is defined by first randomly permuting the systems $A_1 \ldots A_N$ so we may assume that $\rho^{A_1 \cdots A_N}$ is invariant under permutations and the systems are identically distributed. The first term in the bound of Theorem 1 is the worst case error probability in the i.i.d. setting. So, we can regard the parameter $k$ as the copy complexity within the i.i.d. setting. Hence, in order to attain a low total error probability in the non-i.i.d. setting, it is sufficient to take a total number of copies $N = \Omega(k^3 d^2 \log(d))$. This result shows in principle that any learning algorithm designed for i.i.d. states can be transformed into one for general states at an additional cost that is polynomial in the dimension $d$.

A possible algorithm $\mathcal{B}$ achieving the performance of Theorem 1, illustrated in Fig. 2 (Left) and formally described in Algorithm 2 (displayed in Box 2), partitions the training data into 3 parts. For that we choose a random number ($l \sim \text{Unif}\{k+1, \ldots, k+\frac{N}{2}\}$). The first part has size $l-k$ and each system is measured using some fixed measurement $\mathcal{M}_{\text{dist}}$ leading to an output string $\mathbf{w}$. The second part is of size $k$ and we apply the learning algorithm $\mathcal{A}$ and return this prediction. The third part consists of $N-l-1$ systems that are not used by the learning algorithm.

To control the error probability of Algorithm $\mathcal{B}$, we use the de Finetti theorem of ref. 23 (proof of Theorem 2.4) to obtain the approximation for all $1 \leq k < N/2$:

$$\mathbb{E}_{l \sim \text{Unif}\{k+1, \ldots, k+\frac{N}{2}\}, \mathbf{w} \sim \mathcal{M}_{\text{dist}}^{\otimes(l-k)}(\rho^{A_{k+1} \cdots A_l})}\left[\left\|\rho_{\mathbf{w}}^{A_1 \cdots A_k} - \left(\rho_{\mathbf{w}}^{A_1}\right)^{\otimes k}\right\|_1\right] \leq 2\sqrt{\frac{2k^3 d^2 \log(d)}{N}},$$ (gF)

where $\rho_{\mathbf{w}}^{A_1 \cdots A_k}$ is the state conditioned on observing the outcome $\mathbf{w}$ after measuring the quantum state $\rho^{A_{k+1} \cdots A_l}$ with a fixed measurement device $\mathcal{M}_{\text{dist}}^{\otimes(l-k)}$ (which should be an informationally-complete measurement satisfying a low-distortion property) and $\rho_{\mathbf{w}}^{A_1}$ denotes the reduced

quantum state derived by tracing out the systems $A_t$ for $t > 1$ from the quantum state $\rho_{\mathbf{w}}^{A_1 \cdots A_k}$. This theorem shows that when measuring a sufficiently large number of systems of a permutation invariant state, the remaining systems become approximately independent. Crucially, in (gF) the approximation of the state $\rho_{\mathbf{w}}^{A_1 \cdots A_k}$ by the i.i.d. state $\left(\rho_{\mathbf{w}}^{A_1}\right)^{\otimes k}$ is conducted using the trace-norm. This implies that any algorithm utilizing arbitrary measurement strategies that necessitate i.i.d. input states can be generalized to the non-i.i.d. setting at the cost of a new error probability bounded as in Theorem 1. Unfortunately, for some tasks, the additional cost in Theorem 1 is prohibitive. For example, for classical shadows, we expect that the dependence on the dimension $d$ be at most logarithmic.

An example of ref. 24 shows that the dependency in the dimension can not be lifted for a general de Finetti theorem with the trace-norm approximation. On the other hand, the authors of ref. 25 reduced the dependency in the dimension for the LOCC norm. Specifically, it is shown[25] that for a permutation invariant state $\rho^{A_1 \cdots A_N}$ and $1 \leq k < N$, there exists a probability measure denoted as $\nu$, such that the following inequality holds:

$$\sup_{\Lambda_2, \ldots, \Lambda_k}\left\|\text{id} \otimes \Lambda_2 \otimes \cdots \otimes \Lambda_k\left(\rho^{A_1 \cdots A_k} - \int d\nu(\sigma)\sigma^{\otimes k}\right)\right\|_1 \leq \sqrt{\frac{2k^2 \log(d)}{N-k}},$$ (IF)

where the maximization is over measurements channels (a measurement channel corresponding to a measurement device $\mathcal{M} = \{M_x\}_{x \in \mathcal{X}}$ is the quantum channel $\Lambda(\rho) = \sum_{x \in \mathcal{X}} \text{Tr}[M_x \rho] |x\rangle \langle x|$ where $\{|x\rangle\}_{x \in \mathcal{X}}$ is an orthonormal basis). Initially, this might appear adequate for relaxing the assumption of i.i.d. state preparations with a low overhead. However, the process of extending algorithms from i.i.d. inputs to a mixture of i.i.d. states (not to mention permutation-invariant states) is far from straightforward, particularly when dealing with statements that require a correctness with high probability. To address this difficulty, we use the same techniques from ref. 25 and show a randomized local quantum de Finetti theorem.

**Theorem 2.** (Randomized local de Finetti). Let $\rho^{A_1 \cdots A_N}$ be a permutation invariant quantum state, $\{\Lambda_r\}_{r \in \mathcal{R}}$ be a set of measurement channels and $q$ be a probability measure on $\mathcal{R}$. For all $1 \leq k < N/2$, the following inequality holds:

$$\mathbb{E}_{(r_1, \ldots, r_N) \sim q^{\otimes N}, l \sim \text{Unif}\{k+1, \ldots, k+\frac{N}{2}\}} \mathbb{E}_{\mathbf{w}}\left[\left\|\text{id} \otimes \Lambda_{r_2} \otimes \cdots \otimes \Lambda_{r_k}\left(\rho_{\mathbf{w}}^{A_1 \cdots A_k} - \left(\rho_{\mathbf{w}}^{A_1}\right)^{\otimes k}\right)\right\|_1\right] \leq \sqrt{\frac{4k^2 \log(d)}{N}},$$

## BOX 1

# Algorithm 1 - Predicting properties of quantum states in the non-i.i.d. setting - Non-adaptive algorithms

**Require:** The measurements $\{\mathcal{M}_t^{\mathcal{A}}\}_{1\le t\le k_{\mathcal{A}}}$ of algorithm $\mathcal{A}$. A permutation invariant state $\rho^{A_1\cdots A_N}$.

**Ensure:** Adapt the algorithm $\mathcal{A}$ to non-i.i.d. inputs $\rho^{A_1\cdots A_N}$.

1. For $k = k_{\mathcal{A}} \log(k_{\mathcal{A}}/\delta_{\mathcal{A}})$, sample $l \sim \text{Unif}\{k+1, \ldots, k+\frac{N}{2}\}$ and $\mathbf{r} = (r_1, \ldots, r_l) \overset{\text{iid}}{\sim} \text{Unif}\{1, \ldots, k_{\mathcal{A}}\}$.
2. For $t = k+1, ..., l$, apply $\mathcal{M}_{r_t}^{\mathcal{A}}$ to system $A_t$ and obtain outcome $\mathbf{w} \leftarrow \otimes_{t=k+1}^{l} \mathcal{M}_{r_t}^{\mathcal{A}}(\rho)$.
3. For $t = 1, ..., k$, apply $\mathcal{M}_{r_t}^{\mathcal{A}}$ to system $A_t$ and obtain outcome $\mathbf{v} \leftarrow \otimes_{t=1}^{k} \mathcal{M}_{r_t}^{\mathcal{A}}(\rho_{\mathbf{w}})$.
4. For $t = 1, \ldots, k_{\mathcal{A}}$, let $s(t) \in [k_{\mathcal{A}} \log(k_{\mathcal{A}}/\delta_{\mathcal{A}})]$ be the first integer such that $r_{s(t)} = t$.
5. Run the prediction of algorithm $\mathcal{A}$ to the measurement outcomes $v_{s(1)}, \ldots, v_{s(k_{\mathcal{A}})}$ and obtain $p$.
6. Return: $(l, \mathbf{r}, \mathbf{w}, p)$.

## BOX 2

# Algorithm 2 - Predicting properties of quantum states in the non-i.i.d. setting - General algorithms

**Require:** Measurement $\mathcal{A}: L(A_1 \ldots A_k) \to \mathbb{C}^{\mathcal{P}}$. A permutation invariant state $\rho^{A_1\cdots A_N}$.

**Ensure:** Adapt the algorithm $\mathcal{A}$ to non-i.i.d. inputs $\rho^{A_1\cdots A_N}$.

1. Sample $l \sim \text{Unif}\{k+1, \ldots, k+\frac{N}{2}\}$.
2. Apply $\mathcal{M}_{\text{dist}}$ to each system $A_{k+1}$ to $A_l$ and obtain the outcome $\mathbf{w} \leftarrow \mathcal{M}_{\text{dist}}^{\otimes(l-k)}(\rho)$.
3. Run algorithm $\mathcal{A}$ on systems $A_1...A_k$ and obtain the outcome $p \leftarrow \mathcal{A}(\rho)$.
4. Return: $(l, \mathbf{w}, p)$.

where $\mathbf{w}$ is obtained by applying the channel $\Lambda_{r_{k+1}} \otimes \cdots \otimes \Lambda_{r_l}$ to the systems $A_{k+1} \cdots A_l$ of $\rho$.

The result we establish in Theorem 4 is actually slightly stronger: we do not need $\rho^{A_1\cdots A_N}$ to be permutation invariant, it suffices to choose a permutation of the systems $(A_1, \ldots, A_N)$ at random, and the result above holds in expectation over this choice. Moreover, it suffices to sample $(r_1, \ldots, r_N) \sim q^N$ from a permutation-invariant measure on $\mathcal{R}^N$.

Observe that our de Finetti theorem requires stronger assumptions than the local de Finetti theorem (1F)[25]: the distribution of the measurement channels should be permutation invariant (as opposed to arbitrary). However, the implications of our de Finetti theorem are also stronger than the local de Finetti theorem (1F) in that it approximates the projection of the permutation invariant state to exactly i.i.d. states (instead of mixture of i.i.d. states).

It is worth noting that the approximation error in Theorem 2 is significantly smaller than the previous approximation error (gF). Notably, the dependence on the local dimension $d$ is logarithmic, which implies that the total number of copies $N$ only needs to scale as $\Omega(k^2 \log(d))$, as opposed to the more demanding $\Omega(k^3 d^2 \log(d))$. However, the approximation of the state $\rho_{\mathbf{w}}^{A_1\cdots A_k}$ by the i.i.d. state $(\rho_{\mathbf{w}}^{A_1})^{\otimes k}$ in the general trace-norm is no longer guaranteed. This assertion now holds only when applying independent local measurement channels drawn from $\{\Lambda_r\}_{r\in\mathcal{R}}$ according to the distribution $q$ on the quantum state $\rho_{\mathbf{w}}^{A_1\cdots A_k}$. For learning algorithms that are non-adaptive and incoherent (performing single copy measurements using a set of measurement devices chosen before starting the learning procedure), this is enough to bound their error probability and leads to the following theorem.

**Theorem 3.** (Non-adaptive algorithms in the non-i.i.d. setting). Let $\varepsilon > 0$ and $1 \le k < N/2$. Let $\mathcal{A}$ be a learning algorithm designed for i.i.d. input states and performing non-adaptive incoherent measurements. There is an algorithm $\mathcal{B}$ that takes as input an arbitrary state on $N$ systems and possessing an error probability:

$$\delta_{\mathcal{B}}(N, \rho^{A_1\cdots A_N}, 2\varepsilon) \le \sup_{\sigma : \text{state}} \delta_{\mathcal{A}}\left(k, \sigma^{\otimes k}, \varepsilon\right) + \mathcal{O}\left(\sqrt{\frac{k^2 \log^2(N) \log(d)}{N\varepsilon^2}}\right).$$

In terms of copy complexity, to ensure an error probability $\delta$, the number of copies in the non-i.i.d. setting should be

$$N_{\text{non-iid}} = \Omega\left(\frac{\log(d)}{\delta^2 \varepsilon^2} \cdot k_{\text{iid}}(\varepsilon, \delta)^2 \log^2(k_{\text{iid}}/\delta)\right),$$

where $k_{\text{iid}}(\varepsilon, \delta)$ is a sufficient number of copies needed to achieve $\delta/2$ correctness in the i.i.d. setting with a precision parameter $\varepsilon/2$.

To prove Theorem 3, we provide an algorithm $\mathcal{B}$, illustrated in Fig. 2 (Right) and formally described in Algorithm 1 (displayed in Box 1). Note that as $\mathcal{A}$ is assumed to be incoherent and non-adaptive, it is described by some measurements $\{\mathcal{M}_r\}_{r\in\mathcal{R}}$. The algorithm $\mathcal{B}$ partitions the training data into 3 parts. We choose a random number ($l \sim \text{Unif}\{k+1, \ldots, k+\frac{N}{2}\}$). The first part has size $l-k$ and each system is measured using some measurement $\mathcal{M}_r$, where $r$ is chosen at random. This step gives an output string that we denote $\mathbf{w}$. The second part is of size $k$ and we apply the learning algorithm $\mathcal{A}$ and return this prediction. The third part consists of $N-l-1$ systems that are not used by the learning algorithm. Besides this, Algorithm $\mathcal{B}$ returns also the outcomes $\mathbf{w}$ as calibration data.

Many problems of learning properties of quantum states can be solved using algorithms that perform non-adaptive incoherent measurements - that is, measurements which are local on copies and chosen non-adaptively (see Definition 5 for a formal definition). This includes state tomography[26], shadow tomography using classical shadows[7], testing mixedness[27], fidelity estimation[1], verification of pure states[5] among others. For all these problems, we can apply Theorem 3 to extend these algorithms so that they can operate even for non-i.i.d. input states (see Methods' subsection "Applications"). Here, we present this extension for observable prediction via classical shadows. The learning task is to $\varepsilon$-approximate $M$ target observables $\mathrm{tr}(O_i\rho)$ in an unknown $d$-dimensional state $\rho$.

**Proposition 1.** (Classical shadows in the non i.i.d. setting). Fix a collection of $M$ observables $O_i$ on an $n$-qubit system that are also $k$-local. Then, we can use (global or local) Clifford measurements to successfully $\varepsilon$-approximate all target observables in the reduced test state with probability at least 2/3. The number of copies required depends on the measurement process (global/local Clifford) and scales as

$$N = \tilde{\mathcal{O}}\left(\frac{n^3 \max_{i \in [M]} \| O_i \|_2^4 \log^2(M)}{\varepsilon^6}\right) \quad \text{(global Clifford)},$$

$$N = \tilde{\mathcal{O}}\left(\frac{nk^2 16^k \max_{i \in [M]} \| O_i \|_\infty^4 \log^2(M)}{\varepsilon^6}\right) \quad \text{(local Clifford)},$$

where $\tilde{\mathcal{O}}$ hides $\log\log(M)$ and $\log(1/\varepsilon)$ factors.

Notably, taking classical shadows techniques allows us to perform verification in the non-i.i.d. setting[21] without even revealing or making assumptions on the verified target state.

## Application: verification of pure states

The verification of pure states plays an important role in quantum information, notably in the cryptographic setting, where devices, channels or parties are not trusted[8]. This stems from the view of quantum states as resources for certain tasks, which is the case for many applications in quantum information, where the most challenging part is the preparation (and/or distribution) of large entangled states, with which various applications can be carried out by easier, usually local, operations. In measurement-based quantum computing, computation is carried out by single qubit measurements on a large entangled graph state[28]. In networks, many applications rely on the sharing of particular entangled resource states, such as anonymous communication[29], secret sharing[30], and distributed sensing[31]. In these cases, what this means is that, once we can be sure we have the good resource state, we can confirm the application itself. The ability to verify the resource state is then very useful, especially, for example, if the resource state is issued by an untrusted server, or shared over an untrusted network. In these cases, we would clearly not like to make the assumption of an i.i.d. source since this would correspond to assuming i.i.d. attacks by the malicious party. In the simplest case the malicious party would behave well on some runs (in order to convince the user the state is a good resource), and badly on the others (potentially corrupting the application). We then require verification of pure resources states, without the i.i.d. assumption. Once armed with this, for example, verified quantum computation, can be achieved by verifying the underlying resource graph state[32]. Similarly, verifying the underlying resource states provides security over untrusted networks for anonymous communication[33], secret sharing[34] and distributed sensing[18].

As an application of Theorem 3 and Proposition 1, we can show that any $n$-qubit pure state can be verified with either $\tilde{\mathcal{O}}\left(\frac{n}{\varepsilon^6}\right)$ Clifford measurements (see Proposition 4) or $\tilde{\mathcal{O}}(\frac{n^3 16^n}{\varepsilon^6})$ Pauli measurements (see Proposition 5). In words, a verification algorithm should accept only when the test set (post-measurement state) is $\varepsilon$-close to the ideal state in fidelity. Our proposed algorithm offers two significant advantages:

(a) it does not rely on the assumption of i.i.d. state preparations, and (b) it does not demand prior knowledge of the target pure state during the data acquisition phase (that is, the measurements in the algorithm are independent of the state we wish to verify).

Notably, existing verification protocols in the non-i.i.d. setting are state-dependent, such as stabilizer states[15,32,35], weighted graph states, hypergraph states[36], and Dicke states[37]. In contrast, our protocol is independent of the state to be verified. This not only adds to its simplicity but also offers potential advantages in concealing information from the measurement devices regarding the purpose of the test. This blindness is a crucial aspect of many protocols for the verification of computation[14], making this feature valuable in such contexts. Moreover, in both network and computational settings, having a universal protocol simplifies the management of verification steps in broader scenarios where different states may be used for various applications.

## Discussion

We will now give an overview of the relationship between these results and previous works.

The foundational de Finetti theorem, initially introduced by de Finetti[38], states that exchangeable Bernoulli random variables behaves as a mixture of i.i.d. Bernoulli random variables. Subsequently, this statement was quantified and generalized to finite sample sizes and arbitrary alphabets by refs. 39,40. This theorem was further extended to quantum states. Initially in refs. 41,42, the authors established asymptotic generalizations, while in refs. 24,43, the authors presented finite approximations in terms of trace-norm. Later works[25,44] improved these approximations for weaker norms: exponential improvements in the dimension dependence are achieved using the one-way LOCC norm, initially for $k = 2$ by ref. 44, and subsequently for general $k$ by ref. 25. In the mentioned works, the permutation-invariant state was approximated by a mixture of i.i.d. states. In ref. 23, the authors introduced an approximation to i.i.d. states in terms of the trace-norm. In this work, we improve the dimension dependence of this approximation, employing a randomized LOCC norm instead of the trace-norm. Lastly, it is worth noting that information-theoretic proofs for classical finite de Finetti theorems were provided by refs. 45–47.

For the problem of state tomography, the copy complexity in the i.i.d. setting is well-established: $\Theta(d^2/\varepsilon^2)$ with coherent measurements[3,48], and $\Theta(d^3/\varepsilon^2)$ with incoherent measurements[4,26,49], where $\varepsilon$ denotes the approximation accuracy. In the non-i.i.d. setting, the authors of ref. 2 introduced a formulation for the state tomography problem and presented a result using confidence regions. This result pertains to the asymptotic regime, specifically when the state can be represented as a mixture of i.i.d. states. In this article, we build upon the formulation of ref. 2, and we discern between algorithms that return calibration information and those that do not. Furthermore, we introduce a state tomography algorithm with a finite copy complexity (in the non asymptotic regime). Finally, the authors of ref. 50 have also proposed non-i.i.d. tomography algorithms tailored for matrix product states.

The problem of shadow tomography is known to be solvable with a complexity that grows poly-logarithmically with respect to both the dimension and the number of observables, provided (almost) all i.i.d. copies can be coherently measured[6,9,51]. However, if we seek to extend this result to the non-i.i.d. setting using our framework, the copy complexity would be polynomial in the dimension. In the case of incoherent measurements, classical shadows[7,19,52–57] offer efficient algorithms for estimating properties of certain observable classes. Leveraging our findings, these algorithms can be adapted to the non-i.i.d. setting while maintaining comparable performance guarantees. Importantly, this extension retains efficiency for the same class of observables. Finally, refs. 55,58,59 derived shadow tomography results assuming receipt of independent (though not necessarily identical) copies of states. However, it

is worth noting that the assumption of independence, which we overcome in this article, is necessary for their analysis.

Regarding the verification of pure states, optimal and efficient protocols have been proposed in scenarios where the verifier receives independent or product states[5,37,60]. Recently, considerable attention has been given to the verification of pure quantum states in the adversarial scenario, where the received states can be arbitrarily correlated and entangled[21,35,36,61–63]. For instance, in ref. 61, the authors proposed efficient protocols for verifying the ground states of Hamiltonians (subject to certain conditions) and polynomial-time-generated hypergraph states. Meanwhile, in ref. 21, the authors introduced protocols to efficiently verify bipartite pure states, stabilizer states, and Dicke states. Noteworthy attention has also been directed towards the verification of graph states[21,36,62,63]. Furthermore, the authors of ref. 64 studied device-independent verification of quantum states beyond the i.i.d. assumption. Lastly, the verification of continuous-variable quantum states in the adversarial scenario is studied in refs. 65–67. Note that in all these cases the protocols depend explicitly on the state in question.

In summary, we have developed a framework for learning properties of quantum states in the non-i.i.d. setting. The only requirement we impose on the property we aim to learn is the robustness assumption (Definition 3). It would be interesting to analyze the significance of this assumption in the context of the beyond i.i.d. generalizations we prove in the paper (Theorems 5 and 8). Furthermore, while only non-adaptive algorithms that employ incoherent measurements are shown to be extended to encompass non-i.i.d. input states without a loss of efficiency, an open research direction is to investigate whether general algorithms can achieve a similar extension or if there exists an information-theoretic limit.

One of the applications of our results provides the first explicit protocol for verifying any multiparty quantum state, accompanied by clear efficiency statements. However, our results have certain limitations. As discussed in Results' subsection "Evaluating a learning algorithm", the choice of the random permutation should be hidden from the learner in general. In addition, for local Pauli measurements, the scaling is exponential in the number of qubits, and while the scaling for Clifford measurements is close to optimal, they are non-local across each copy. In addition, the scaling in the error parameters is not optimal. Nevertheless, we see our results as a first proof-of-principle showing that beyond i.i.d. learning is feasible in many settings with performance guarantees that are comparable to the i.i.d. guarantees. We expect that further work will improve the bounds we obtain both for the general statements as well as using specificities of classes of learning tasks. In addition, we believe that this work will contribute to the transfer of techniques between the areas of learning theory and quantum verification.

## Methods

We first present the necessary notation and preliminaries in the next section. This section is essential for a complete understanding of the evaluation of an algorithm in the non-i.i.d. setting and the distinction we make between general and non-adaptive algorithms.

### Notation and preliminaries

Let $[d]$ denote the set of integers from 1 to $d$ and $[t, s]$ denote the set of integers from $t$ to $s$. Hilbert spaces are denoted $A, B, \ldots$ and we will use these symbols for both the label of a quantum system and the system itself. We let $d_A$ be the dimension of the Hilbert space $A$. Let $L(A)$ denote the set of linear maps from $A$ to itself. A quantum state on $A$ is defined as

$$\rho \in L(A) : \rho \succcurlyeq 0 \text{ and } \mathrm{Tr}[\rho] = 1,$$

where $\rho \succcurlyeq 0$ means that $\rho$ is positive semidefinite. The set of quantum states on $A$ is denoted by $D(A)$. For an integer $N \geq 2$, we denote the $N$-partite composite system by $A_1 A_2 \cdots A_N = A_1 \otimes A_2 \otimes \cdots \otimes A_N$. A classical-quantum state is a bipartite states that can be written in the form

$$\rho^{XB} = \sum_{x \in \mathcal{X}} p_x |x\rangle \langle x|^X \otimes \rho_x^B,$$

for some orthonormal basis $\{|x\rangle\}_{x \in \mathcal{X}}$ of the classical outcome space $X$, where $\mathbf{p} = (p_x)_{x \in \mathcal{X}}$ is a probability distribution and for $x \in \mathcal{X}$, $\rho_x^B$ is a quantum state. It will also be useful to interpret a classical quantum state as $\rho^{XB} \in \mathbb{C}^{\mathcal{X}} \otimes L(B)$, i.e., as a vector $(p_x \rho_x^B)_{x \in \mathcal{X}}$ of operators acting on $B$. This interpretation is more appropriate when the classical system takes continuous values. In this case, technically $\mathbb{C}^{\mathcal{X}}$ should be interpreted as the space $L_1(\mathcal{X}, \mu)$ with some measure $\mu$ on $\mathcal{X}$. Quantum channels are linear maps $\mathcal{N} : L(A) \to L(B)$ that can be written in the form

$$\mathcal{N}(\rho) = \sum_{x \in \mathcal{X}} K_x \rho K_x^\dagger \quad \text{for all } \rho \in L(A).$$

Here, the Kraus operators $\{K_x\}_{x \in \mathcal{X}}$ are linear maps from $A$ to $B$ and satisfy $\sum_{x \in \mathcal{X}} K_x^\dagger K_x = \mathbb{I}_{d_A}$, where $\mathbb{I}_{d_A}$ is the identity matrix in $d_A$ dimensions ($[\mathbb{I}_{d_A}]_{i,j} = \delta_{i,j}$). Equivalently, $\mathcal{N}$ is trace preserving and completely positive. The partial trace $\mathrm{Tr}_B[.]$ is a quantum channel from $AB$ to $A$ defined as

$$\mathrm{Tr}_B[\rho] = \sum_{i=1}^{d_B} \left( \mathbb{I}_{d_A}^A \otimes \langle i|^B \right) \rho \left( \mathbb{I}_{d_A}^A \otimes |i\rangle^B \right) \quad \text{for all } \rho \in L(A) \otimes L(B).$$

For bipartite state $\rho^{AB}$, we denote the reduced state on $A$ by $\rho^A = \mathrm{Tr}_B[\rho^{AB}]$. In general, for an $N$-partite state $\rho^{A_1 \cdots A_N}$ and for two integers $t \leq s \in [N]$, we denote by $\rho^{A_t \cdots A_s}$ the quantum state obtained by tracing out the systems $A_i$ for $i < t$, as well as $i > s$. In formulas:

$$\rho^{A_t \cdots A_s} = \mathrm{Tr}_{A_1 \cdots A_{t-1} A_{s+1} \cdots A_N} \left[ \rho^{A_1 \cdots A_N} \right].$$

In the situation where all systems except one ($A_t$ for $t \in [N]$) are traced out, we use the notation

$$\mathrm{Tr}_{-A_t}[\rho] = \mathrm{Tr}_{A_1 \cdots A_{t-1} A_{t+1} \cdots A_N} \left[ \rho^{A_1 \cdots A_N} \right].$$

A quantum channel $\Lambda$ with classical output system is called a measurement channel, and is described by a POVM (positive operator-valued measure) $\{M_x\}_{x \in \mathcal{X}} \in (L(A))^{\mathcal{X}}$ where the measurement operators satisfy $M_x \succcurlyeq 0$ and $\sum_{x \in \mathcal{X}} M_x = \mathbb{I}_{d_A}$. After performing the measurement on a quantum state $\rho \in D(A)$ we observe the outcome $x \in \mathcal{X}$ with probability $\mathrm{Tr}[M_x \rho]$. The measurement channel $\Lambda$ should be viewed as a linear map $\Lambda : L(A) \to \mathbb{C}^{\mathcal{X}}$ (preserving positivity and normalization) defined by:

$$\forall \rho \in L(A) : \Lambda(\rho) = (\mathrm{Tr}[M_x \rho])_{x \in \mathcal{X}}.$$

For a measurement operator $0 \preccurlyeq M_x \preccurlyeq \mathbb{I}$ acting on $A$, we write $\rho$ conditioned on observing the outcome $x$ by:

$$\rho_x^B = \frac{1}{\mathrm{Tr}[M_x \rho^A]} \mathrm{Tr}_A \left[ \left( M_x \otimes \mathbb{I}^B \right) \cdot \rho^{AB} \right].$$

Note that this display is only well-defined if $\mathrm{Tr}[M_x \rho] > 0$. We extend it consistently to $\mathrm{Tr}[M_x \rho] = 0$ by identifying $\rho_x^B$ with a single fixed density matrix, e.g. the maximally mixed state. The state $\rho$ and the measurement $\Lambda$ define a probability measure $\mathbb{P}_{\Lambda(\rho)}[.]$ on $\mathcal{X}$ by $\mathbb{P}_{\Lambda(\rho)}[x] = \mathrm{Tr}[M_x \rho^A]$ and we will usually write $x$ to be a random variable associated with this measure $\mathbb{P}_{\Lambda(\rho)}[.]$.

**I.i.d. setting - input state.** A common assumption in the field of quantum learning is that the learning algorithm is provided with $N$

independent and identically distributed (i.i.d.) copies of the unknown quantum state.

**Definition 1.** (I.i.d. states). Let $N \geq 1$ be a positive integer and $A_1 \cong A_2 \cong \cdots \cong A_N$ be $N$ isomorphic quantum systems of dimension $d$. An i.i.d. state refers to an $N$-partite quantum state $\rho \in D(A_1 \cdots A_N)$ that can be expressed as $\rho = \sigma^{\otimes N}$ where $\sigma \in D(A_1)$ is a quantum state.

An i.i.d. state possesses the characteristic of permutation invariance: if we permute the arrangement of the constituent states $\sigma$, the overall state $\rho = \sigma^{\otimes N}$ remains unchanged. For a formal definition of permutation invariance, let $\mathfrak{S}_N$ be the permutation group of $N$ elements.

**Definition 2.** (Permutation invariant states). For $\pi \in \mathfrak{S}_N$, let $C_\pi$ be the permutation operator corresponding to the permutation $\pi$, that is:

$$C_\pi |i_1\rangle \otimes \cdots \otimes |i_N\rangle = |i_{\pi^{-1}(1)}\rangle \otimes \cdots \otimes |i_{\pi^{-1}(N)}\rangle, \quad \forall i_1, \ldots, i_N \in [d].$$

A state $\rho \in D(A_1 \cdots A_N)$ is permutation invariant if for all $\pi \in \mathfrak{S}_N$ we have

$$\rho^{A_1 \cdots A_N} = C_\pi \rho^{A_1 \cdots A_N} C_\pi^\dagger.$$

Note that every i.i.d. state $\rho = \sigma^{\otimes N}$ is permutation-invariant. The converse is not necessarily true, however. Take, for example an $N$-qubit GHZ state: $\rho = |GHZ_N\rangle\langle GHZ_N|$ with $|GHZ_N\rangle = (|0\cdots0\rangle + |1\cdots1\rangle)/\sqrt{2}$. This state is unaffected under permutation operators, but it is very far from an i.i.d. tensor product. It is worthwhile to point out that permutation invariance plays nicely with partial measurements. If $\rho$ is permutation invariant then for an operator $0 \preceq M_x \preceq \mathbb{I}$ acting on $A_1 \cdots A_t$, the post-measurement state $\rho_x^{A_{t+1} \cdots A_N}$ is also permutation invariant. So we can define the reduced state conditioned on observing $x$ as:

$$\rho_x^{A_N} = \text{Tr}_{-A_N}\left[\rho_x^{A_{t+1} \cdots A_N}\right] = \text{Tr}_{-A_j}\left[\rho_x^{A_{t+1} \cdots A_N}\right] = \rho_x^{A_j}, \quad \forall j \in [t+1, N].$$

**Problems/tasks.** In this article, we consider problems of learning quantum states' properties. These problems can be formulated using a SUCCESS event:

**Definition 3.** (Success formulation of learning properties of quantum states). A quantum learning problem for states on the system $A$ is defined by: a set $\mathcal{P}$ of possible predictions together with a set of successful predictions SUCCESS $\subseteq \mathcal{P} \times D(A)$. If $(p, \sigma) \in$ SUCCESS, then $p$ is considered a correct prediction for $\sigma$. Otherwise, it is considered incorrect.

Many problems have a precision parameter $\varepsilon$, we write in this case SUCCESS$_\varepsilon$ for the pairs $(p, \sigma)$ for which $p$ is a correct prediction for $\sigma$ within precision $\varepsilon$.

We say that the property SUCCESS$_\varepsilon$ satisfies the robustness assumption whenever

$$\forall(\sigma, \xi) \in D(A)^2, \text{ if } \|\sigma - \xi\|_1 \leq \varepsilon', \text{ then } (p, \sigma) \in \text{SUCCESS}_\varepsilon \Rightarrow (p, \xi) \in \text{SUCCESS}_{\varepsilon + \varepsilon'}.$$

**Example 1.** We illustrate the SUCCESS set for the shadow tomography, full state tomography, verification of a pure state, and testing mixedness problems:

- Shadow tomography: for some family of $M$ observables $O_1, \ldots, O_M$ satisfying $0 \preceq O_i \preceq \mathbb{I}$, the objective is to estimate all their expectation values within an additive error $\varepsilon$. In this case, a prediction is an $M$-tuple of numbers in [0, 1], i.e., $\mathcal{P} = [0,1]^M$ and the correct pairs are given by

$$\text{SUCCESS}_\varepsilon = \{((\mu_1, \ldots, \mu_M), \sigma) \,|\, \forall 1 \leq i \leq M : |\mu_i - \text{Tr}[O_i\sigma]|_1 \leq \varepsilon\} \subset [0,1]^M \times D(A).$$

- State tomography: the objective is to obtain a description of the full state. In this case, a prediction is a description of a density

operator, i.e., $\mathcal{P} = D(A)$ and we have

$$\text{SUCCESS}_\varepsilon = \{(\rho, \sigma) \,|\, \|\rho - \sigma\|_1 \leq \varepsilon\} \subset D(A) \times D(A).$$

- (Tolerant) verification of pure states: in this problem, the objective is to output 0 if the state we have is $\varepsilon$-close to $|\Psi\rangle$ and output 1 if it is $2\varepsilon$-far from $|\Psi\rangle$. In this case, the prediction is a bit, i.e., $\mathcal{P} = \{0, 1\}$ and notice that this is a promise problem in the sense that there are inputs for which any output is valid. For this reason, it is simpler to define the incorrect prediction pairs:

$$(\text{SUCCESS}_\varepsilon)^c = \{(1, \sigma)\,|\,\langle\Psi|\sigma|\Psi\rangle \geq 1 - \varepsilon\} \cup \{(0, \sigma)\,|\,\langle\Psi|\sigma|\Psi\rangle \leq 1 - 2\varepsilon\} \subset \{0, 1\} \times D(A).$$

- (Tolerant) testing mixedness of quantum states: this problem is similar to the previous one, except that we are testing if the state is maximally mixed or not. In this case, we have

$$(\text{SUCCESS}_\varepsilon)^c = \left\{(1, \sigma)\,\Big|\,\Big\|\sigma - \frac{\mathbb{I}}{d}\Big\|_1 \leq \varepsilon\right\} \cup \left\{(0, \sigma)\,\Big|\,\Big\|\sigma - \frac{\mathbb{I}}{d}\Big\|_1 \geq 2\varepsilon\right\} \subset \{0, 1\} \times D(A).$$

Observe that all these problems, by the triangle inequality, satisfy the robustness assumption.

Before specifying the algorithms we consider, let us first recall how one could formulate a problem when the input state is non-i.i.d.

**Non-i.i.d. setting - input state.** Given a learning problem defined by SUCCESS$_\varepsilon$, in the usual setting, an algorithm takes as an input an i.i.d. state $\rho^{A_1 \cdots A_N} = \sigma^{\otimes N}$ and outputs a prediction $p$. Then, we say that this algorithm succeeds if $(p, \sigma)$ belongs to the SUCCESS set. In the setting where the input state $\rho^{A_1 \cdots A_N}$ is no longer an i.i.d. state, it is not clear when the algorithm succeeds. In what follows, we follow[2,21] and present a way to evaluate algorithms with possibly non-i.i.d. input states.

Consider a collection of $N$ finite dimensional quantum systems $A_1 \cong \cdots \cong A_N$. We denote the dimension of $A_1$ by $d$ (for an $n$-qubit system $A_1$, we have $d = 2^n$). This collection is shuffled uniformly at random so that the state $\rho^{A_1 \cdots A_N} \in D(A_1 \cdots A_N)$ is permutation invariant. We need to form two sets:

- The train set which consists of the first $N-1$ copies of the state. Some of these copies are measured in order to construct the estimations necessary for the learning task, and
- The test set which consists of the last copy (the state on $A_N$) that is used to test the accuracy of the estimations deduced from the train set. This copy should not be measured.

Since the state $\rho^{A_1 \cdots A_N}$ can now be entangled, it is possible that the train and test sets cannot be separated from each other. In particular, the measurements we perform on the train set may affect the test set. In addition, the choice of measuring a copy or not can also affect the test set. At the end, we compare the estimations from the train set with the single copy of the test set (see Fig. 3 for an illustration).

Note that in the i.i.d. setting, i.e., $\rho = \sigma^{\otimes N}$, the train set will be of the form $\sigma^{\otimes N-1}$ and the test set of the form $\sigma$ where we compare the estimations deduced from measuring the state $\sigma$ with the test state $\sigma$. Thus we recover the usual setting. The following example illustrates the importance of choosing the test state as the post-measurement state.

**Example 2.** Consider the following permutation invariant state

$$\rho^{A_1 \cdots A_N} = \frac{1}{d}\sum_{i=1}^{d} |i\rangle\langle i|^{\otimes N}.$$

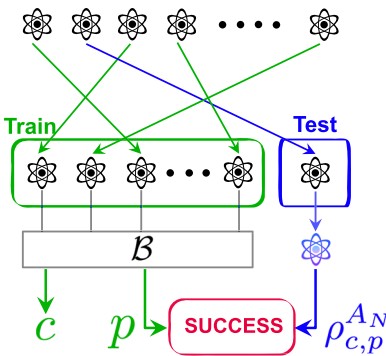

**Fig. 3 | A general algorithm for learning properties of quantum states in the non-i.i.d. setting.** A learning algorithm $\mathcal{B}$ takes as input the $N-1$ copies of the train set and returns a prediction $p$ and a calibration $c$. Success occurs if $p$ is (approximately) compatible with the remaining post-measurement test copy $\rho_{c,p}^{A_N}$.

If we measure the first system $A_1$ with the canonical basis $\mathcal{M} = \{|i\rangle\langle i|\}_{i\in[d]}$, we observe $m \in [d]$ with probability $1/d$ and the state collapses to:

$$\rho_m^{A_2\cdots A_N} = |m\rangle\langle m|^{\otimes N-1}.$$

After this initial measurement, the state of the last system $A_N$ is always equal to $|m\rangle\langle m|$. Therefore, it is more appropriate to compare the prediction to $\rho_m^{A_N} = |m\rangle\langle m|$ rather than the reduced measurement state $\rho^{A_N} = \frac{1}{d}\sum_{i=1}^d |i\rangle\langle i| = \frac{\mathbb{1}}{d}$.

**Algorithms.** In a general algorithm, the prediction can be an arbitrary quantum channel from the train set to a prediction.

**Definition 4.** (General algorithm). Let $N \geq 1$ be a positive integer and $A_1 \cong A_2 \cong \cdots \cong A_N$ be $N$ isomorphic quantum systems. An algorithm for a learning problem with prediction set $\mathcal{P}$ is simply a measurement channel $\mathcal{B} : \mathrm{L}(A_1\cdots A_{N-1}) \to \mathbb{C}^{\mathcal{P}}$.

We will also be interested in a special class of learning algorithms: non-adaptive incoherent algorithms that can only measure each system separately and then apply an arbitrary classical post-processing function.

**Definition 5.** (Non-adaptive algorithm). Let $N \geq 1$ be a positive integer and $A_1 \cong A_2 \cong \cdots \cong A_N$ be $N$ isomorphic quantum systems. For a non-adaptive algorithm, the prediction channel $\mathcal{B}$ should be of the form $\mathcal{B} = \mathcal{D} \circ (\mathcal{M}_1 \otimes \cdots \otimes \mathcal{M}_{N-1})$ where $\mathcal{M}_i : \mathrm{L}(A_i) \to \mathbb{C}^{\mathcal{X}_i}$ are measurement channels, and $\mathcal{D} : \mathbb{C}^{\mathcal{X}_1} \otimes \cdots \otimes \mathbb{C}^{\mathcal{X}_{N-1}} \to \mathbb{C}^{\mathcal{P}}$ is an arbitrary post-processing channel (aka a classical data processing algorithm).

**Error probability.** We can assess an algorithm based on its probability of error, which represents the likelihood that its outcomes do not satisfy the desired property for a given test set or state. Note that if a learning algorithm outputs more information than simply the prediction $p$, this may influence the post-measurement state that we are comparing against and influence the error probability. This leads us to the following definition which allows the learning algorithm to output auxiliary information, which we refer to as calibration. See Fig. 3 for an illustration of algorithms with calibration information.

**Definition 6.** (Error probability in the non-i.i.d. setting with calibration). Let $N \geq 1$ be a positive integer and $A_1 \cong A_2 \cong \cdots \cong A_N$ be $N$ isomorphic quantum systems. Let $\rho^{A_1\cdots A_N} \in \mathrm{D}(A_1\cdots A_N)$ be permutation invariant. A learning algorithm with calibration is given by a

quantum channel $\mathcal{B} : \mathrm{L}(A_1\ldots A_{N-1}) \to \mathbb{C}^{\mathcal{C}} \otimes \mathbb{C}^{\mathcal{P}}$. The error probability of the algorithm on input $\rho$ is:

$$\delta_{\mathcal{B}}(N, \rho^{A_1\cdots A_N}, \varepsilon) = \mathbb{P}_{(c,p)\sim\mathcal{B}(\rho)}\left[\left(p, \rho_{c,p}^{A_N}\right) \notin \mathrm{SUCCESS}_\varepsilon\right],$$

where $(c, p)$ is a random variable having distribution $\mathcal{B}(\rho^{A_1\cdots A_{N-1}})$.

Note that, if $\rho$ is i.i.d., the conditioning on $c$, $p$ does not have any effect on the post-measurement state and Definition 6 coincides with the usual definition of the error probability.

We refer to Supplementary Note 3 for the distinction between error probabilities with and without calibration. In particular, we are able to extend algorithms to the non-i.i.d. setting without calibration for a wide range of learning problems that can be formulated using a function with reasonable assumptions.

In the following, we state and prove a randomized local de Finetti theorem. We then concentrate on non-adaptive algorithms employing incoherent measurements and illustrate how to extend their applicability to handle non-i.i.d. input states. In Methods' subsection "Applications", we apply the results we obtained for non-adaptive algorithms (Theorem 5) to specific examples, including observable prediction with classical shadows, verification of pure states, fidelity estimation, quantum state tomography, and testing the mixedness of states. Finally, in Methods' subsection "General algorithms in the non-i.i.d. setting", we detail the process of adapting any algorithm to function within the non-i.i.d. framework.

**Randomized local de Finetti Theorem**
In this section, we state and prove a randomized local de Finetti theorem. Note that the statement does not need the state $\rho^{A_1A_2\cdots A_N}$ to be permutation invariant, but we show that for most choices of permutations of the systems $(A_1, A_2, \ldots, A_N)$, the conditional state of the first few copies is close to product.

**Theorem 4.** (Randomized local de Finetti). Let $N \geq 1$ be a positive integer and $A_1 \cong A_2 \cong \cdots \cong A_N$ be $N$ isomorphic quantum systems of dimension $d$. Let $1 \leq k < \sqrt{\frac{N}{\log(d)}}$. Let $\rho^{A_1\cdots A_N}$ be a state and let $q^N$ be a permutation-invariant measure on $\mathcal{R}^N$. Let $\{\Lambda_r\}_{r\in\mathcal{R}}$ be a set of measurement channels with input system $A$ and output system $X$. Let $\mathbf{j} = (j_1, \ldots, j_N)$ be a random permutation of $\{1, \ldots, N\}$, $l \sim \mathrm{Unif}\{k+1, \ldots, k+\frac{N}{2}\}$, $\mathbf{r} = (r_1, \ldots, r_N) \sim q^N$ and $\mathbf{w} = (w_{l+1}, \ldots, w_{k+N/2})$ be the outcomes of measuring the systems $A_{j_{l+1}}, \ldots, A_{j_{k+N/2}}$ using the measurements $\Lambda_{r_{l+1}}, \ldots, \Lambda_{r_{k+N/2}}$. The following inequality holds:

$$\mathbb{E}_{\mathbf{j}, l, \mathbf{r}\sim q^N}\left[\sum_{\mathbf{w}} p_{\mathbf{r}}(\mathbf{w}) \left\|\mathrm{id} \otimes \left(\bigotimes_{i=2}^{k+1}\Lambda_{r_i}\right)\left(\rho_{l,\mathbf{r},\mathbf{w}}^{A_{j_1}\cdots A_{j_{k+1}}} - \bigotimes_{i=2}^{k+1}\rho_{l,\mathbf{r},\mathbf{w}}^{A_{j_i}}\right)\right\|_1\right]$$
$$\leq \sqrt{\frac{4k^2\log(d)}{N}},$$

where $p_{\mathbf{r}}(\mathbf{w}) = \mathrm{Tr}\left[\langle\mathbf{w}|(\Lambda_{r_{l+1}} \otimes \cdots \otimes \Lambda_{r_{k+N/2}})(\rho^{A_{j_1}\cdots A_{j_N}})|\mathbf{w}\rangle\right]$ and we defined the conditional state $\rho_{l,\mathbf{r},\mathbf{w}}$ as

$$\rho_{l,\mathbf{r},\mathbf{w}}^{A_{j_1}\cdots A_{j_{k+1}}} = \frac{1}{p_{\mathbf{r}}(\mathbf{w})}\mathrm{Tr}_{A_{j_{k+2}}\cdots A_{j_N}}\left[\langle\mathbf{w}|(\Lambda_{r_{l+1}} \otimes \cdots \otimes \Lambda_{r_{k+N/2}})(\rho^{A_{j_1}\cdots A_{j_N}})|\mathbf{w}\rangle\right].$$

Note that if $\rho^{A_1\cdots A_N}$ is permutation invariant, the random permutation $\mathbf{j}$ is not needed and we can replace $j_i$ by $i$ and $\otimes_{i=1}^{k+1}\rho_{l,\mathbf{r},\mathbf{w}}^{A_{j_i}}$ by $(\rho_{l,\mathbf{r},\mathbf{w}}^{A_N})^{\otimes k+1}$ in the above expressions.

The proof is inspired by refs. 25,68 and [23].
Proof. The mutual information is defined as follows:

$$\mathcal{I}(A_1 : A_2 : \cdots : A_N)_\rho = S(\rho^{A_1}) + \cdots + S(\rho^{A_N}) - S(\rho^{A_1\cdots A_N})$$

where $S(\rho) = -\operatorname{Tr}[\rho \log(\rho)]$ is the Von Neumann entropy of $\rho$. The mutual information of quantum-classical state $\xi^{A_{j_1}\cdots A_{j_k}C} = \sum_m p_m \rho_m^{A_{j_1}\cdots A_{j_k}} \otimes |m\rangle\langle m|^C$ is defined as follows:

$$\mathcal{I}(A_{j_1}:\cdots:A_{j_k}|C)_\xi = \sum_m p_m \mathcal{I}(A_{j_1}:\cdots:A_{j_k})_{\rho_m}.$$

The chain rule implies:

$$\mathcal{I}(A_{j_1}:\cdots:A_{j_k}|C)_\xi = \mathcal{I}(A_{j_1}:A_{j_2}|C)_\xi + \mathcal{I}(A_{j_1}A_{j_2}:A_{j_3}|C)_\xi \\ +\cdots+\mathcal{I}(A_{j_1}\cdots A_{j_{k-1}}:A_{j_k}|C)_\xi.$$

Moreover, we can apply the data-processing inequality locally, for all quantum channels $\Gamma_i : L(A_{j_i}) \to L(X_{j_i})$, let $\zeta = \Gamma_1 \otimes \cdots \otimes \Gamma_k \otimes \mathrm{id}(\xi^{A_{j_1}\cdots A_{j_k}C})$ we have:

$$\mathcal{I}(X_{j_1}:\cdots:X_{j_k}|C)_\zeta \le \mathcal{I}(A_{j_1}:\cdots:A_{j_k}|C)_\xi.$$

For every $\mathbf{r} = (r_1, \ldots, r_N)$ define the state:

$$\pi_{\mathbf{r}}^{A_{j_1}X_{j_2}\ldots X_{j_N}} = \mathrm{id} \otimes \Lambda_{r_2} \otimes \cdots \otimes \Lambda_{r_N}\left(\rho^{A_{j_1}\cdots A_{j_N}}\right).$$

We have by the chain rule:

$$\mathbb{E}_{\mathbf{r}\sim q^N}\left[\mathcal{I}\left(A_{j_1}X_{j_2}\cdots X_{j_k}:X_{j_{k+1}}\cdots X_{j_{k+N/2}}\right)_{\pi_{\mathbf{r}}}\right] \\ = \mathbb{E}_{\mathbf{r}\sim q^N}\left[\sum_{l=k+1}^{k+N/2}\mathcal{I}(A_{j_1}X_{j_2}\cdots X_{j_k}:X_{j_l}|X_{j_{l+1}}\cdots X_{j_{k+N/2}})_{\pi_{\mathbf{r}}}\right]. \quad (3)$$

By taking the average over the random permutation $\mathbf{j}$ and using the fact that the distribution $q^N$ is invariant under the permutation of the systems $k+1$ and $l$ we have for all $k+1 \le l \le k+N/2$:

$$\mathbb{E}_{\mathbf{j},\mathbf{r}\sim q^N}\left[\mathcal{I}(A_{j_1}X_{j_2}\cdots X_{j_k}:X_{j_l}|X_{j_{l+1}}\cdots X_{j_{k+N/2}})_{\pi_{\mathbf{r}}}\right] \\ = \mathbb{E}_{\mathbf{j},\mathbf{r}\sim q^N}\left[\mathcal{I}(A_{j_1}X_{j_2}\cdots X_{j_k}:X_{j_{k+1}}|X_{j_{l+1}}\cdots X_{j_{k+N/2}})_{\pi_{\mathbf{r}}}\right],$$

hence:

$$\mathbb{E}_{\mathbf{j},\mathbf{r}\sim q^N}\left[\sum_{l=k+1}^{k+N/2}\mathcal{I}(A_{j_1}X_{j_2}\cdots X_{j_k}:X_{j_l}|X_{j_{l+1}}\cdots X_{j_{k+N/2}})_{\pi_{\mathbf{r}}}\right] \\ = \mathbb{E}_{\mathbf{j},\mathbf{r}\sim q^N}\left[\sum_{l=k+1}^{k+N/2}\mathcal{I}(A_{j_1}X_{j_2}\cdots X_{j_k}:X_{j_{k+1}}|X_{j_{l+1}}\ldots X_{j_{k+N/2}})_{\pi_{\mathbf{r}}}\right]. \quad (4)$$

Now using the data-processing inequality for the partial trace channel and the fact that $q^N$ is permutation invariant and averaging over $\mathbf{j}$, we obtain for all $2 \le i \le k$ and $k+1 \le l \le k+N/2$:

$$\mathbb{E}_{\mathbf{j},\mathbf{r}\sim q^N}\left[\mathcal{I}(A_{j_1}X_{j_2}\cdots X_{j_k}:X_{j_{k+1}}|X_{j_{l+1}}\ldots X_{j_{k+N/2}})_{\pi_{\mathbf{r}}}\right] \\ \ge \mathbb{E}_{\mathbf{j},\mathbf{r}\sim q}\left[\mathcal{I}(A_{j_1}X_{j_2}\cdots X_{j_i}:X_{j_{k+1}}|X_{j_{l+1}}\ldots X_{j_{k+N/2}})_{\pi_{\mathbf{r}}}\right] \quad (5) \\ = \mathbb{E}_{\mathbf{j},\mathbf{r}\sim q^N}\left[\mathcal{I}(A_{j_1}X_{j_2}\cdots X_{j_i}:X_{j_{l+1}}|X_{j_{l+1}}\ldots X_{j_{k+N/2}})_{\pi_{\mathbf{r}}}\right].$$

Then we can apply the chain rule to get for all $k+1 \le l \le k+N/2$:

$$\sum_{i=2}^{k+1}\mathbb{E}_{\mathbf{j},\mathbf{r}\sim q^N}\left[\mathcal{I}(A_{j_1}X_{j_2}\cdots X_{j_{i-1}}:X_{j_i}|X_{j_{l+1}}\ldots X_{j_{k+N/2}})_{\pi_{\mathbf{r}}}\right] \\ = \mathbb{E}_{\mathbf{j},\mathbf{r}\sim q^N}\left[\mathcal{I}(A_{j_1}:X_{j_2}:\cdots:X_{j_{k+1}}|X_{j_{l+1}}\ldots X_{j_{k+N/2}})_{\pi_{\mathbf{r}}}\right]. \quad (6)$$

Now, for each $k+1 \le l \le k+N/2$, we introduce the notations $\pi_{\mathbf{r},\mathbf{w}}$ for the states conditioned on the systems $(X_{j_{l+1}}, \ldots, X_{j_{k+N/2}})$ taking the value $\mathbf{w}$, and $p_{\mathbf{r}}(\mathbf{w})$ for the probability of obtaining outcome $\mathbf{w}$. Hence using Pinsker's inequality then Cauchy Schwarz's inequality, we obtain:

$$\mathbb{E}_{\mathbf{j},\mathbf{r}\sim q^N}\left[\mathcal{I}(A_{j_1}:X_{j_2}:\cdots:X_{j_{k+1}}|X_{j_{l+1}}\cdots X_{j_{k+N/2}})_{\pi_{\mathbf{r}}}\right] \\ = \mathbb{E}_{\mathbf{j},\mathbf{r}\sim q^N}\left[\sum_{\mathbf{w}}p_{\mathbf{r}}(\mathbf{w})\mathcal{I}(A_{j_1}:X_{j_2}:\cdots:X_{j_{k+1}})_{\pi_{\mathbf{r},\mathbf{w}}}\right] \\ \ge \frac{1}{2}\mathbb{E}_{\mathbf{j},\mathbf{r}\sim q^N}\left[\sum_{\mathbf{w}}p_{\mathbf{r}}(\mathbf{w})\left\|\pi_{\mathbf{r},\mathbf{w}}^{A_{j_1}X_{j_2}\cdots X_{j_{k+1}}} - \pi_{\mathbf{r},\mathbf{w}}^{A_{j_1}}\otimes\pi_{\mathbf{r},\mathbf{w}}^{X_{j_2}}\otimes\cdots\otimes\pi_{\mathbf{r},\mathbf{w}}^{X_{j_{k+1}}}\right\|_1^2\right] \\ = \frac{1}{2}\mathbb{E}_{\mathbf{j},\mathbf{r}\sim q^N}\left[\sum_{\mathbf{w}}p_{\mathbf{r}}(\mathbf{w})\left\|(\mathrm{id}\otimes\bigotimes_{i=2}^{k+1}\Lambda_{r_i})(\rho_{l,\mathbf{r},\mathbf{w}}^{A_{j_1}A_{j_2}\cdots A_{j_{k+1}}}) - \rho_{l,\mathbf{r},\mathbf{w}}^{A_{j_1}}\otimes\bigotimes_{i=2}^{k+1}\Lambda_{r_i}(\rho_{l,\mathbf{r},\mathbf{w}}^{A_{j_i}})\right\|_1^2\right] \\ \ge \frac{1}{2}\left(\mathbb{E}_{\mathbf{j},\mathbf{r}\sim q^N}\left[\sum_{\mathbf{w}}p_{\mathbf{r}}(\mathbf{w})\left\|(\mathrm{id}\otimes\bigotimes_{i=2}^{k+1}\Lambda_{r_i})(\rho_{l,\mathbf{r},\mathbf{w}}^{A_{j_1}\cdots A_{j_{k+1}}} - \rho_{l,\mathbf{r},\mathbf{w}}^{A_{j_1}}\otimes\cdots\otimes\rho_{l,\mathbf{r},\mathbf{w}}^{A_{j_{k+1}}})\right\|_1\right]\right)^2. \quad (7)$$

Combining the (In)Eqs. (3)–(7) we obtain:

$$\mathbb{E}_{\mathbf{j},\mathbf{r}\sim q^N}\left[\mathcal{I}\left(A_{j_1}X_{j_2}\cdots X_{j_k}:X_{j_{k+1}}\cdots X_{j_{k+N/2}}\right)_{\pi_{\mathbf{r}}}\right] \\ \ge \frac{1}{k}\sum_{l=k+1}^{k+N/2}\sum_{i=2}^{k+1}\mathbb{E}_{\mathbf{j},\mathbf{r}\sim q^N}\left[\mathcal{I}(A_{j_1}X_{j_2}\cdots X_{j_{i-1}}:X_{j_i}|X_{j_{l+1}}\cdots X_{j_{k+N/2}})_{\pi_{\mathbf{r}}}\right] \\ = \frac{N}{2k}\mathbb{E}_{\mathbf{j},l,\mathbf{r}\sim q^N}\left[\mathcal{I}(A_{j_1}:X_{j_2}:\cdots:X_{j_{k+1}}|X_{j_{l+1}}\cdots X_{j_{k+N/2}})_{\pi_{\mathbf{r}}}\right] \\ \ge \frac{N}{4k}\left(\mathbb{E}_{\mathbf{j},l,\mathbf{r}\sim q^N}\left[\sum_{\mathbf{w}}p_{\mathbf{r}}(\mathbf{w})\left\|(\mathrm{id}\otimes\bigotimes_{i=2}^{k+1}\Lambda_{r_i})(\rho_{l,\mathbf{r},\mathbf{w}}^{A_{j_1}\cdots A_{j_{k+1}}} - \rho_{l,\mathbf{r},\mathbf{w}}^{A_{j_1}}\otimes\cdots\otimes\rho_{l,\mathbf{r},\mathbf{w}}^{A_{j_{k+1}}})\right\|_1\right]\right)^2.$$

Since $\mathcal{I}(A_{j_1}X_{j_2}\cdots X_{j_k}:X_{j_{k+1}}\cdots X_{j_{k+N/2}})_{\pi_{\mathbf{r}}} \le \log(d^k) = k\log(d)$ for all $\mathbf{r}\in\mathcal{R}^N$, we obtain finally the desired inequality:

$$\mathbb{E}_{\mathbf{j},l,\mathbf{r}\sim q^N}\left[\sum_{\mathbf{w}}p_{\mathbf{r}}(\mathbf{w})\left\|(\mathrm{id}\otimes\bigotimes_{i=2}^{k+1}\Lambda_{r_i})(\rho_{l,\mathbf{r},\mathbf{w}}^{A_{j_1}\cdots A_{j_{k+1}}} - \rho_{l,\mathbf{r},\mathbf{w}}^{A_{j_1}}\otimes\cdots\otimes\rho_{l,\mathbf{r},\mathbf{w}}^{A_{j_{k+1}}})\right\|_1\right] \\ \le \sqrt{\frac{4k}{N}\cdot\mathbb{E}_{\mathbf{j},\mathbf{r}\sim q^N}\left[\mathcal{I}\left(A_{j_1}X_{j_2}\cdots X_{j_k}:X_{j_{k+1}}\cdots X_{j_{k+N/2}}\right)_{\pi_{\mathbf{r}}}\right]} \\ \le \sqrt{\frac{4k}{N}\cdot\sup_{\mathbf{j}\in\mathfrak{S}_N,\,\mathbf{r}\in\mathcal{R}^N}\mathcal{I}\left(A_{j_1}X_{j_2}\cdots X_{j_k}:X_{j_{k+1}}\cdots X_{j_{k+N/2}}\right)_{\pi_{\mathbf{r}}}} \\ \le \sqrt{\frac{4k^2\log(d)}{N}}. \quad (8)$$

We refer to Supplementary Note 2 for an illustration of Theorem 4 for a specific permutation invariant state and a specific distribution of measurements.

### Non-adaptive algorithms in the non-i.i.d. setting

In this section, our emphasis is on problems related to learning properties of quantum states (as defined in Definition 3) and algorithms that operate through non-adaptive incoherent measurements (as defined in Definition 5). We present a method to extend the applicability of these algorithms beyond the constraint of i.i.d. input states.

Let SUCCESS$_\varepsilon$ define a property of quantum states. We consider a fixed non-adaptive algorithm $\mathcal{A}$ that performs non-adaptive measurements on the systems which make up the train set. Our approach introduces a strategy $\mathcal{B}$ outlined in Algorithm 1 (displayed in Box 1) and illustrated in Fig. 4, which extends the functionality of the algorithm $\mathcal{A}$ to encompass non-i.i.d. states. The input state, denoted as $\rho^{A_1\cdots A_N} \in D(A_1\cdots A_N)$, is now an $N$-partite state that can be entangled.

In words, given a non-adaptive incoherent algorithm $\mathcal{A}$ that uses a set of measurement devices $\{\mathcal{M}_t\}_t$, Algorithm 1 measures a large number of the state's subsystems using measurement devices uniformly

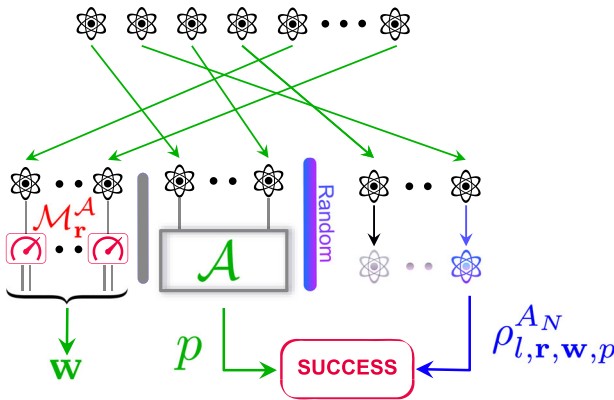

**Fig. 4 | Illustration of Algorithm 1.** Algorithm 1 measures a large number of the state's subsystems using $\mathcal{M}_{\mathbf{r}}^{\mathcal{A}}$ that represents measurement devices uniformly chosen from the i.i.d. algorithm's set of measurements (red and green parts). Then, Algorithm 1 applies the data processing of Algorithm $\mathcal{A}$ to the outcomes of a part of these subsystems (green part), leading to a prediction $p$. Algorithm 1 returns the remaining outcomes as calibration $\mathbf{w}$. Success occurs if $p$ is (approximately) compatible with the remaining post-measurement test copy $\rho_{l,\mathbf{r},\mathbf{w},p}^{A_N}$.

chosen from $\{\mathcal{M}_t\}_t$ (see Fig. 4, red and green parts). This ensures that the (small) portion of measured subsystems intended for the learning algorithm approximately behave like i.i.d. copies (see Fig. 4, green part). Then, in order to predict the property, Algorithm 1 applies the data processing of Algorithm $\mathcal{A}$ to the outcomes of these subsystems.

More precisely, since $\mathcal{A}$ is a non-adaptive algorithm, it performs measurements using the measurements devices $\{\mathcal{M}_t^{\mathcal{A}}\}_{1 \le t \le k_{\mathcal{A}}}$. We sample at each time a POVM uniformly at random from the set $\{\mathcal{M}_t^{\mathcal{A}}\}_{1 \le t \le k_{\mathcal{A}}}$ so we need slightly more copies $k_{\mathcal{A}} \log(k_{\mathcal{A}}/\delta_{\mathcal{A}})$ to span $\{\mathcal{M}_t^{\mathcal{A}}\}_{1 \le t \le k_{\mathcal{A}}}$:

Let $l \sim \text{Unif}\{k_{\mathcal{A}} \log(k_{\mathcal{A}}/\delta_{\mathcal{A}})+1, \ldots, k_{\mathcal{A}} \log(k_{\mathcal{A}}/\delta_{\mathcal{A}})+\frac{N}{2}\}$. For each $i \in [l]$, we choose $r_i \in \text{Unif}\{1, \ldots, k_{\mathcal{A}}\}$ and we measure the system $A_i$ using the measurement $\mathcal{M}_{r_i}^{\mathcal{A}}$.

To compute the prediction, Algorithm $\mathcal{B}$ considers the $k_{\mathcal{A}} \log(k_{\mathcal{A}}/\delta_{\mathcal{A}})$ outcomes $\mathbf{v}$ of measurements $\mathcal{M}_{r_1}, \ldots, \mathcal{M}_{r_{k_{\mathcal{A}} \log(k_{\mathcal{A}}/\delta_{\mathcal{A}})}}$. Provided $r_1, \ldots, r_{k_{\mathcal{A}} \log(k_{\mathcal{A}}/\delta_{\mathcal{A}})}$ span the set $\{1, \ldots, k_{\mathcal{A}}\}$, the prediction algorithm of $\mathcal{A}$ is applied to the relevant systems (as described in Algorithm 1). The coupon collector's problem ensures that $r_1, \ldots, r_{k_{\mathcal{A}} \log(k_{\mathcal{A}}/\delta_{\mathcal{A}})}$ spans all elements in $\{1, \ldots, k_{\mathcal{A}}\}$ with high probability.

We can support this algorithm with the following rigorous bound on the failure probability that only depends on problem-specific parameters, as well as the performance of an ideal i.i.d. learning algorithm.

**Theorem 5.** (Non-adaptive algorithms in the non-i.i.d. setting). Let $N \ge 1$ be a positive integer and $A_1 \cong A_2 \cong \cdots \cong A_N$ be $N$ isomorphic quantum systems of dimension $d$. Let $\varepsilon > 0$ and $k_{\mathcal{A}} \le N/\log(N)$. Let $\mathcal{A}$ be a non-adaptive algorithm suitable for i.i.d. input states and performing measurements with $\{\mathcal{M}_t^{\mathcal{A}}\}_{1 \le t \le k_{\mathcal{A}}}$. Algorithm 1 has an error probability satisfying:

$$\delta_{\mathcal{B}}(N, \rho^{A_1 \cdots A_N}, 2\varepsilon) \le 2\sup_{l,\mathbf{r},\mathbf{w}} \delta_{\mathcal{A}}\left(k_{\mathcal{A}}, \left(\rho_{l,\mathbf{r},\mathbf{w}}^{A_N}\right)^{\otimes k_{\mathcal{A}}}, \varepsilon\right) + 6\sqrt{\frac{k_{\mathcal{A}}^2 \log^2(k_{\mathcal{A}}/\delta_{\mathcal{A}})\log(d)}{N\varepsilon^2}}.$$

**Remark 1.** The first component of this upper bound essentially represents the error probability of algorithm $\mathcal{A}$ when applied to an i.i.d. input state $\sigma^{\otimes k_{\mathcal{A}}}$, where $\sigma \in \{\rho_{l,\mathbf{r},\mathbf{w}}^{A_N}\}_{l,\mathbf{r},\mathbf{w}}$. Note that here we are not required to control this error probability over all states but only over the post-measurement states $\{\rho_{l,\mathbf{r},\mathbf{w}}^{A_N}\}_{l,\mathbf{r},\mathbf{w}}$. The second component

consists of an error term that accounts for the possibility of the input state $\rho^{A_1 \cdots A_N}$ being non i.i.d..

**Remark 2.** To achieve an error probability of at most $\delta$, one could start by determining a value for $k_{\mathcal{A}} = k(\mathcal{A}, \delta, \varepsilon)$ such that for all $l$, $\mathbf{r}$, $\mathbf{w}$, $\delta_{\mathcal{A}}(k_{\mathcal{A}}, (\rho_{l,\mathbf{r},\mathbf{w}}^{A_N})^{\otimes k_{\mathcal{A}}}, \varepsilon/2) \le \delta/6$. Subsequently, the total number of copies can be set to

$$N_{\text{non--iid}} = \frac{18^2 \log(d)}{\delta^2 \varepsilon^2} \cdot k_{\mathcal{A}}^2 \log^2(6k_{\mathcal{A}}/\delta).$$

This choice of training data size ensures that the overall probability of failure obeys $\delta_{\mathcal{B}}(N_{\text{non--iid}}, \rho^{A_1 \cdots A_N}, \varepsilon) \le \delta$, as desired.

**Remark 3.** The second error term of this upper bound can be improved to $6\sqrt{\frac{k \sup_{\mathbf{j},\mathbf{r}} \mathcal{I}_{\mathbf{j}}(\pi_{\mathbf{r}})}{N\varepsilon^2}}$ through the same proof outlined in Theorem 4 (see Inequality (8)). When the state $\rho^{A_1 \cdots A_N} = \sigma^{\otimes N}$ is i.i.d., the mutual information $\mathcal{I}_{\mathbf{j}}(\pi_{\mathbf{r}}) = \mathcal{I}(A_{j_1} X_{j_2} \cdots X_{j_k} : X_{j_{k+1}} \cdots X_{j_{k+N/2}})_{\pi_{\mathbf{r}}}$ becomes zero for all local quantum channels $\Lambda_{\mathbf{r}} = \text{id} \otimes \Lambda_{r_2} \otimes \cdots \otimes \Lambda_{r_{k+N/2}}$. Consequently, the second error term vanishes in the i.i.d. setting and we recover the i.i.d. error probability, albeit with a minor loss: substituting $\varepsilon$ with $2\varepsilon$ and $k_{\mathcal{A}}$ with $k_{\mathcal{A}} \log(2k_{\mathcal{A}}/\delta)$.

**Remark 4.** In Algorithm 1, the initial stage of measuring systems $A_{k+1} \cdots A_l$ (corresponding to outcomes $\mathbf{w}$) can be thought as a projection phase, while the subsequent stage involving measuring the systems $A_1 \cdots A_k$ (corresponding to outcomes $\mathbf{v}$) can be regarded as a learning phase. Note that we utilize only the outcomes $\mathbf{v}$ for the prediction component $p$; however, the outcomes $\mathbf{w}$ hold significance in enabling the application of the randomized de Finetti Theorem 4.

**Remark 5.** Algorithm 1 extends only non-adaptive incoherent algorithms to the non-i.i.d. setting as it applies the measurements of the i.i.d. algorithm chosen uniformly at random. Adaptive algorithms are shown to outperform their non-adaptive counterparts for some learning[49,69] and testing[70] problems. We leave the question of extending adaptive incoherent algorithms for future work.

The remaining of this section is dedicated to the proof of Theorem 5.

Proof of Theorem 5. In this proof we differentiate between $k_{\mathcal{A}}$ and $k$. The former is the copy complexity of the non-adaptive algorithm $\mathcal{A}$ while the latter is a parameter we use for the proof to ensure that all the measurement devices used by the non-adaptive algorithm $\mathcal{A}$ are sampled. Let $l \sim \text{Unif}\{k + 1, \ldots, k + N/2\}$ and $\mathbf{r} = (r_1, \ldots, r_l) \overset{\text{iid}}{\sim} \text{Unif}\{1, \ldots, k_{\mathcal{A}}\}$. Algorithm $\mathcal{B}$ applies measurement $\mathcal{M}_{r_i}^{\mathcal{A}}$ to system $A_i$ for all $i \in [l]$.

Our proof strategy will be to approximate the reduced post-measurement state $\rho_{l,\mathbf{r},\mathbf{w},p}^{A_N}$ by the reduced post-measurement state $\rho_{l,\mathbf{r},\mathbf{w}}^{A_N}$. Then, we approximate the state $\rho_{l,\mathbf{r},\mathbf{w}}^{A_1 \cdots A_k}$ by the i.i.d. state $\left(\rho_{l,\mathbf{r},\mathbf{w}}^{A_N}\right)^{\otimes k}$ using the de Finetti Theorem 4.

More precisely, we write the error probability:

$$\begin{aligned}
\delta_{\mathcal{B}}(N, \rho^{A_1 \cdots A_N}, \varepsilon + \varepsilon') &= \mathbb{P}_{l,\mathbf{r},\mathbf{w},p}\left[(p, \rho_{l,\mathbf{r},\mathbf{w},p}^{A_N}) \notin \text{SUCCESS}_{\varepsilon+\varepsilon'}\right] \\
&= \mathbb{P}_{l,\mathbf{r},\mathbf{w},p}\left[(p, \rho_{l,\mathbf{r},\mathbf{w},p}^{A_N}) \notin \text{SUCCESS}_{\varepsilon+\varepsilon'}, \left\|\rho_{l,\mathbf{r},\mathbf{w},p}^{A_N} - \rho_{l,\mathbf{r},\mathbf{w}}^{A_N}\right\|_1 \le \varepsilon'\right] \\
&\quad + \mathbb{P}_{l,\mathbf{r},\mathbf{w},p}\left[(p, \rho_{l,\mathbf{r},\mathbf{w},p}^{A_N}) \notin \text{SUCCESS}_{\varepsilon+\varepsilon'}, \left\|\rho_{l,\mathbf{r},\mathbf{w},p}^{A_N} - \rho_{l,\mathbf{r},\mathbf{w}}^{A_N}\right\|_1 > \varepsilon'\right] \\
&\le \mathbb{P}_{l,\mathbf{r},\mathbf{w},p}\left[(p, \rho_{l,\mathbf{r},\mathbf{w}}^{A_N}) \notin \text{SUCCESS}_{\varepsilon}\right] + \mathbb{P}_{l,\mathbf{r},\mathbf{w},p}\left[\left\|\rho_{l,\mathbf{r},\mathbf{w},p}^{A_N} - \rho_{l,\mathbf{r},\mathbf{w}}^{A_N}\right\|_1 > \varepsilon'\right]
\end{aligned}$$

(9)

where we used the robustness condition for the problem defined by $\text{SUCCESS}_\varepsilon$.

Let us start with the second term by relating the reduced post-measurement state $\rho_{l,\mathbf{r},\mathbf{w},p}^{A_N}$ with $\rho_{l,\mathbf{r},\mathbf{w}}^{A_N}$. Note that as $p$ is a function of $\mathbf{v}$, it suffices to bound the distance between $\rho_{l,\mathbf{r},\mathbf{w}}^{A_N}$ and $\rho_{l,\mathbf{r},\mathbf{w},\mathbf{v}}^{A_N}$, which is done in the following lemma.

**Lemma 1.** We have for all $\varepsilon'>0$:

$$\mathbb{P}_{l,\mathbf{r},\mathbf{w},\mathbf{v}}\left[\left\|\rho_{l,\mathbf{r},\mathbf{w},\mathbf{v}}^{A_N}-\rho_{l,\mathbf{r},\mathbf{w}}^{A_N}\right\|_1>\varepsilon'\right]\leq\sqrt{\frac{16k^2\log(d)}{N\varepsilon'^2}}.$$

*Proof of Lemma 1.* We use the notation $M_{\mathbf{w}}=\otimes_{t=k+1}^l M_{w_t}^t$ and $M_{\mathbf{v}}=\otimes_{t=1}^k M_{v_t}^t$ where $\mathcal{M}_{r_t}^A=\{M_x^t\}_{x\in\mathcal{X}}$ for $t\in[N]$. We have:

$$\left\|\mathcal{M}_{r_1}\otimes\cdots\otimes\mathcal{M}_{r_k}\otimes\mathrm{id}\left(\rho_{l,\mathbf{r},\mathbf{w}}^{A_1\cdots A_k A_N}-\left(\rho_{l,\mathbf{r},\mathbf{w}}^{A_N}\right)^{\otimes k+1}\right)\right\|_1$$

$$=\sum_{\mathbf{v}}\left\|\mathrm{Tr}_{A_1\cdots A_k}\left[(M_{\mathbf{v}}\otimes\mathbb{I})\left(\rho_{l,\mathbf{r},\mathbf{w}}^{A_1\cdots A_k A_N}-\left(\rho_{l,\mathbf{r},\mathbf{w}}^{A_N}\right)^{\otimes k+1}\right)\right]\right\|_1$$

$$=\sum_{\mathbf{v}}\left\|\mathrm{Tr}_{A_1\cdots A_k}\left[(M_{\mathbf{v}}\otimes\mathbb{I})\rho_{l,\mathbf{r},\mathbf{w}}^{A_1\cdots A_k A_N}\right]-\mathrm{Tr}\left[M_{\mathbf{v}}\left(\rho_{l,\mathbf{r},\mathbf{w}}^{A_N}\right)^{\otimes k}\right]\rho_{l,\mathbf{r},\mathbf{w}}^{A_N}\right\|_1$$

and similarly by the data processing inequality we have

$$\left\|\mathcal{M}_{r_1}\otimes\cdots\otimes\mathcal{M}_{r_k}\otimes\mathrm{id}\left(\rho_{l,\mathbf{r},\mathbf{w}}^{A_1\cdots A_k A_N}-\left(\rho_{l,\mathbf{r},\mathbf{w}}^{A_N}\right)^{\otimes k+1}\right)\right\|_1$$

$$\geq\left\|\mathcal{M}_{r_1}\otimes\cdots\otimes\mathcal{M}_{r_k}\left(\rho_{l,\mathbf{r},\mathbf{w}}^{A_1\cdots A_k}-\left(\rho_{l,\mathbf{r},\mathbf{w}}^{A_N}\right)^{\otimes k}\right)\right\|_1$$

$$=\sum_{\mathbf{v}}\left|\mathrm{Tr}\left[M_{\mathbf{v}}\left(\left(\rho_{l,\mathbf{r},\mathbf{w}}^{A_N}\right)^{\otimes k}-\rho_{l,\mathbf{r},\mathbf{w}}^{A_1\cdots A_k}\right)\right]\right|$$

$$=\sum_{\mathbf{v}}\left\|\mathrm{Tr}\left[M_{\mathbf{v}}\left(\rho_{l,\mathbf{r},\mathbf{w}}^{A_N}\right)^{\otimes k}\right]\rho_{l,\mathbf{r},\mathbf{w}}^{A_N}-\mathrm{Tr}\left[(M_{\mathbf{v}}\otimes\mathbb{I})\rho_{l,\mathbf{r},\mathbf{w}}\right]\rho_{l,\mathbf{r},\mathbf{w}}^{A_N}\right\|_1.$$

So the triangle inequality implies:

$$\mathbb{E}_{l,\mathbf{r}}\left[\sum_{\mathbf{v},\mathbf{w}}\mathrm{Tr}[(M_{\mathbf{v}}\otimes M_{\mathbf{w}}\otimes\mathbb{I})\rho]\left\|\rho_{l,\mathbf{r},\mathbf{w},\mathbf{v}}^{A_N}-\rho_{l,\mathbf{r},\mathbf{w}}^{A_N}\right\|_1\right]$$

$$=\mathbb{E}_{l,\mathbf{r}}\left[\sum_{\mathbf{v},\mathbf{w}}\mathrm{Tr}[(M_{\mathbf{w}}\otimes\mathbb{I})\rho]\,\mathrm{Tr}[(M_{\mathbf{v}}\otimes\mathbb{I})\rho_{l,\mathbf{r},\mathbf{w}}]\left\|\rho_{l,\mathbf{r},\mathbf{w},\mathbf{v}}^{A_N}-\rho_{l,\mathbf{r},\mathbf{w}}^{A_N}\right\|_1\right]$$

$$=\mathbb{E}_{l,\mathbf{r}}\left[\sum_{\mathbf{v},\mathbf{w}}\mathrm{Tr}[(M_{\mathbf{w}}\otimes\mathbb{I})\rho]\left\|\mathrm{Tr}_{A_1\cdots A_k}\left[(M_{\mathbf{v}}\otimes\mathbb{I})\rho_{l,\mathbf{r},\mathbf{w}}^{A_1\cdots A_k A_N}\right]-\mathrm{Tr}[(M_{\mathbf{v}}\otimes\mathbb{I})\rho_{l,\mathbf{r},\mathbf{w}}]\rho_{l,\mathbf{r},\mathbf{w}}^{A_N}\right\|_1\right]$$

$$\leq\mathbb{E}_{l,\mathbf{r}}\left[\sum_{\mathbf{v},\mathbf{w}}\mathrm{Tr}[(M_{\mathbf{w}}\otimes\mathbb{I})\rho]\left\|\mathrm{Tr}_{A_1\cdots A_k}\left[(M_{\mathbf{v}}\otimes\mathbb{I})\rho_{l,\mathbf{r},\mathbf{w}}^{A_1\cdots A_k A_N}\right]-\mathrm{Tr}\left[M_{\mathbf{v}}\left(\rho_{l,\mathbf{r},\mathbf{w}}^{A_N}\right)^{\otimes k}\right]\rho_{l,\mathbf{r},\mathbf{w}}^{A_N}\right\|_1\right]$$

$$+\mathbb{E}_{l,\mathbf{r}}\left[\sum_{\mathbf{v},\mathbf{w}}\mathrm{Tr}[(M_{\mathbf{w}}\otimes\mathbb{I})\rho]\left\|\mathrm{Tr}\left[M_{\mathbf{v}}\left(\rho_{l,\mathbf{r},\mathbf{w}}^{A_N}\right)^{\otimes k}\right]\rho_{l,\mathbf{r},\mathbf{w}}^{A_N}-\mathrm{Tr}[(M_{\mathbf{v}}\otimes\mathbb{I})\rho_{l,\mathbf{r},\mathbf{w}}]\rho_{l,\mathbf{r},\mathbf{w}}^{A_N}\right\|_1\right]$$

$$\leq 2\,\mathbb{E}_{l,\mathbf{r}}\left[\sum_{\mathbf{w}}\mathrm{Tr}[(M_{\mathbf{w}}\otimes\mathbb{I})\rho]\left\|\mathcal{M}_{r_1}\otimes\cdots\otimes\mathcal{M}_{r_k}\otimes\mathrm{id}\left(\rho_{l,\mathbf{r},\mathbf{w}}^{A_1\cdots A_k A_N}-\left(\rho_{l,\mathbf{r},\mathbf{w}}^{A_N}\right)^{\otimes k+1}\right)\right\|_1\right].$$

On the other hand, we have by the randomized local de Finetti Theorem 4:

$$\mathbb{E}_{l,\mathbf{r}}\left[\sum_{\mathbf{w}}\mathrm{Tr}[(M_{\mathbf{w}}\otimes\mathbb{I})\rho]\left\|\mathcal{M}_{r_1}\otimes\cdots\otimes\mathcal{M}_{r_k}\otimes\mathrm{id}\left(\rho_{l,\mathbf{r},\mathbf{w}}^{A_1\cdots A_k A_N}-\left(\rho_{l,\mathbf{r},\mathbf{w}}^{A_N}\right)^{\otimes k+1}\right)\right\|_1\right]\leq\sqrt{\frac{4k^2\log(d)}{N}}.$$

Hence we can deduce the following inequality:

$$\mathbb{E}_{l,\mathbf{r},\mathbf{w},p}\left[\left\|\rho_{l,\mathbf{r},\mathbf{w},\mathbf{v}}^{A_N}-\rho_{l,\mathbf{r},\mathbf{w}}^{A_N}\right\|_1\right]\leq 2\sqrt{\frac{4k^2\log(d)}{N}}. \tag{10}$$

Finally, the Markov's inequality implies:

$$\mathbb{P}_{l,\mathbf{r},\mathbf{w},\mathbf{v}}\left[\left\|\rho_{l,\mathbf{r},\mathbf{w},\mathbf{v}}^{A_N}-\rho_{l,\mathbf{r},\mathbf{w}}^{A_N}\right\|_1>\varepsilon'\right]\leq\frac{\mathbb{E}_{l,\mathbf{r},\mathbf{w},\mathbf{v}}\left[\|\rho_{l,\mathbf{r},\mathbf{w},\mathbf{v}}^{A_N}-\rho_{l,\mathbf{r},\mathbf{w}}^{A_N}\|_1\right]}{\varepsilon'}\leq\sqrt{\frac{16k^2\log(d)}{N\varepsilon'^2}}.$$

This completes the proof of Lemma 1.

We now go back to (9) and consider the first term. Let us denote $\mathcal{M}_{\mathbf{r}}=\otimes_{i=1}^l\mathcal{M}_{r_i}$ and $\mathcal{D}$ for the channel mapping the outcomes $\mathbf{v}$ and outputting a prediction $p$ (as described in Algorithm 1). We have

$$\mathbb{P}_{l,\mathbf{r},\mathbf{w},p}\left[(p,\rho_{l,\mathbf{r},\mathbf{w}}^{A_N})\notin\text{SUCCESS}_\varepsilon\right]$$

$$=\mathbb{E}_{l,\mathbf{r},\mathbf{w}}\left[\mathbb{P}_{p\sim\mathcal{D}(\mathcal{M}_{\mathbf{r}}(\rho_{l,\mathbf{r},\mathbf{w}}^{A_1\cdots A_k}))}\left[\left(p,\rho_{l,\mathbf{r},\mathbf{w}}^{A_N}\right)\notin\text{SUCCESS}_\varepsilon\right]\right]$$

$$\leq\mathbb{E}_{l,\mathbf{r},\mathbf{w}}\left[\mathbb{P}_{p\sim\mathcal{D}(\mathcal{M}_{\mathbf{r}}((\rho_{l,\mathbf{r},\mathbf{w}}^{A_N})^{\otimes k}))}\left[\left(p,\rho_{l,\mathbf{r},\mathbf{w}}^{A_N}\right)\notin\text{SUCCESS}_\varepsilon\right]\right]+\sqrt{\frac{4k^2\log(d)}{N}}.$$

using the randomized local de Finetti Theorem 4. To relate

$$\mathbb{E}_{l,\mathbf{r},\mathbf{w}}\left[\mathbb{P}_{p\sim\mathcal{D}(\mathcal{M}_{\mathbf{r}}((\rho_{l,\mathbf{r},\mathbf{w}}^{A_N})^{\otimes k}))}\left[\left(p,\rho_{l,\mathbf{r},\mathbf{w}}^{A_N}\right)\notin\text{SUCCESS}_\varepsilon\right]\right]$$

to the behavior of algorithm $\mathcal{A}$, we introduce the event that all the measurement devices that algorithm $\mathcal{A}$ needs are sampled before $k$:

$$\mathcal{G}=\left\{[k_{\mathcal{A}}]\subset\{r_t\}_{1\leq t\leq k}\right\}.$$

The union bound implies:

$$\mathbb{P}[\mathcal{G}^c]=\mathbb{P}\left[\exists 1\leq s\leq k_{\mathcal{A}}:s\notin\{r_t\}_{1\leq t\leq k}\right]$$

$$\leq\sum_{s=1}^{k_{\mathcal{A}}}\mathbb{P}\left[\forall 1\leq t\leq k:r_t\neq s\right]=k_{\mathcal{A}}\left(1-\frac{1}{k_{\mathcal{A}}}\right)^k\leq k_{\mathcal{A}}e^{-k/k_{\mathcal{A}}}.$$

Under $\mathcal{G}$ we let $s(t)\in[k]$ be the smallest integer such that $r_{s(t)}=t$ for $t=1,\ldots,k_{\mathcal{A}}$. Then

$$\mathbb{E}_{l,\mathbf{r},\mathbf{w}}\left[\mathbb{P}_{p\sim\mathcal{D}(\mathcal{M}_{\mathbf{r}}((\rho_{l,\mathbf{r},\mathbf{w}}^{A_N})^{\otimes k}))}\left[\left(p,\rho_{l,\mathbf{r},\mathbf{w}}^{A_N}\right)\notin\text{SUCCESS}_\varepsilon\right]\right]$$

$$\leq\mathbb{E}_{l,\mathbf{r},\mathbf{w}}\left[\mathbb{P}_{p\sim\mathcal{D}(\mathcal{M}_{\mathbf{r}}((\rho_{l,\mathbf{r},\mathbf{w}}^{A_N})^{\otimes k}))}\left[\left(p,\rho_{l,\mathbf{r},\mathbf{w}}^{A_N}\right)\notin\text{SUCCESS}_\varepsilon\right]\mathbf{1}\{\mathcal{G}\}\right]+\mathbb{P}[\mathcal{G}^c]$$

$$\leq\mathbb{E}_{l,\mathbf{r},\mathbf{w}}\left[\mathbb{P}_{p\sim\mathcal{A}\left((\rho_{l,\mathbf{r},\mathbf{w}}^{A_N})^{\otimes k_{\mathcal{A}}}\right)}\left[\left(p,\rho_{l,\mathbf{r},\mathbf{w}}^{A_N}\right)\notin\text{SUCCESS}_\varepsilon\right]\right]+k_{\mathcal{A}}e^{-k/k_{\mathcal{A}}}$$

$$\leq\sup_{l,\mathbf{r},\mathbf{w}}\delta_{\mathcal{A}}\left(k_{\mathcal{A}},\left(\rho_{l,\mathbf{r},\mathbf{w}}^{A_N}\right)^{\otimes k_{\mathcal{A}}},\varepsilon\right)+k_{\mathcal{A}}e^{-k/k_{\mathcal{A}}}.$$

Choosing $k=k_{\mathcal{A}}\log(k_{\mathcal{A}}/\delta_{\mathcal{A}})$, $\varepsilon'=\varepsilon$ and bounding $\sqrt{\frac{k^2\log(d)}{N}}\leq\sqrt{\frac{k^2\log(d)}{N\varepsilon^2}}$ we obtain the desired bound on the error probability.

## Applications

In this section, we apply the non i.i.d. framework that we have developed in Methods' subsection "Non-adaptive algorithms in the non-i.i.d. setting" to address specific and concrete examples. These examples include classical shadows for shadow tomography, the verification of pure states, fidelity estimation, state tomography, and testing mixedness of states.

**Classical shadows for shadow tomography.** In the shadow tomography problem, we have $M\geq 1$ known observables denoted as $O_1,\ldots,O_M$, with each observable satisfying $0\preceq O_i\preceq\mathbb{I}$, along with $N$ i.i.d. copies of an unknown quantum state $\sigma$. The task is now to $\varepsilon$-approximate all $M$ observable values $\mathrm{tr}(O\sigma)$ with success probability (at least) $1-\delta$. In ref. 7, the authors have introduced two specific protocols known as classical shadows, which employ (global) Clifford and Pauli

(or local Clifford) measurements to tackle this problem. In their analysis, the authors crucially rely on the assumption of input states being i.i.d., which is essential for the successful application (concentration) of the median of means technique (estimator). Given that both algorithms proposed by ref. 7 are non-adaptive (as defined in Definition 5), we can leverage Theorem 5 to extend the applicability of these algorithms to encompass input states that are not i.i.d..

The initial algorithm employs measurements that follow either the Haar or Clifford distributions. The Haar probability measure stands as the unique invariant probability measure over the unitary (compact) group and is denoted $\mathcal{L}_{\mathbf{Haar}}$. For the Clifford distribution, certain definitions need to be introduced. We consider an $n$-qubit quantum system denoted as $A \cong \mathbb{C}^d$ where $d = 2^n$. First define the set of Pauli matrices as follows:

$$\mathbf{P}_n = \left\{ e^{i\theta\pi/2}\sigma_1 \otimes \cdots \otimes \sigma_n \mid \theta = 0,1,2,3, \ \sigma_i \in \{\mathbb{I}, X, Y, Z\} \right\}.$$

Subsequently, the Clifford group is defined as the centralizer of the aforementioned set of Pauli matrices:

$$\mathbf{Cl}(2^n) = \{U \in \mathbb{U}_d : U\mathbf{P}_n U^\dagger = \mathbf{P}_n\}.$$

It is known[71,72] that the Clifford group is generated by the Hadamard ($H$), phase ($S$) and CNOT gates:

$$H = \frac{1}{\sqrt{2}}\begin{pmatrix} 1 & 1 \\ 1 & -1 \end{pmatrix}, S = \begin{pmatrix} 1 & 0 \\ 0 & i \end{pmatrix} \text{ and } \mathrm{CNOT} = \begin{pmatrix} 1 & 0 & 0 & 0 \\ 0 & 1 & 0 & 0 \\ 0 & 0 & 0 & 1 \\ 0 & 0 & 1 & 0 \end{pmatrix}.$$

Moreover, the Clifford group is finite (of order at most $\exp(\mathcal{O}(n^2))$)[72]. Sampling a Clifford unitary matrix is given by selecting an element uniformly and randomly from the Clifford group $\mathbf{Cl}(2^n)$. We denote this distribution by $\mathcal{L}_{\mathbf{Clifford}}$. Importantly, Clifford distribution is a 3-design[73–75], that is for all $s = 0, 1, 2, 3$:

$$\mathbb{E}_{U \sim \mathcal{L}_{\mathbf{Clifford}}}\left[U^{\otimes s} \otimes \overline{U}^{\otimes s}\right] = \mathbb{E}_{U \sim \mathcal{L}_{\mathbf{Haar}}}\left[U^{\otimes s} \otimes \overline{U}^{\otimes s}\right].$$

This property of the Clifford distribution has a significant implication: unitaries distributed according to $\mathcal{L}_{\mathbf{Clifford}}$ or $\mathcal{L}_{\mathbf{Haar}}$ distributions yield identical performance for the classical shadows[7]. Now we can state the first result of ref. 7:

**Theorem 6.** (Ref. 7, rephrased). Let $\{O_i\}_{i\in[M]}$ be $M$ observables. There is an algorithm for predicting the expected values of the observables $\{O_i\}_{i\in[M]}$ under the state $\sigma$ to within $\varepsilon$ with an error probability $\delta$. This algorithm performs i.i.d. measurements following the distribution $\mathcal{L}_{\mathbf{Clifford}}$ (or $\mathcal{L}_{\mathbf{Haar}}$), and it requires a total number of i.i.d. copies of the state $\sigma$ satisfying:

$$N = \mathcal{O}\left(\frac{\max_{i\in[M]}\mathrm{Tr}\left[O_i^2\right]\log(M/\delta)}{\varepsilon^2}\right).$$

Hence by Theorem 5 there is an algorithm $\mathcal{B}$ in the non-i.i.d. setting with an error probability:

$$\delta_{\mathcal{B}}(N, \rho^{A_1\cdots A_N}, 2\varepsilon) \leq 2\sup_{\sigma:\text{ state }} \delta_{\mathcal{A}}\left(k_{\mathcal{A}}, \sigma^{\otimes k_{\mathcal{A}}}, \varepsilon\right) + 6\sqrt{\frac{k_{\mathcal{A}}^2\log^2(k_{\mathcal{A}}/\delta_{\mathcal{A}})\log(d)}{N\varepsilon^2}}.$$

By taking $k_{\mathcal{A}} = \mathcal{O}\left(\frac{\max_{i\in[M]}\mathrm{Tr}[O_i^2]\log(M/\delta)}{\varepsilon^2}\right)$ as the complexity of classical shadows in the i.i.d. setting, we deduce that a total number of copies

sufficient to achieve $\delta$-correctness in the non-i.i.d. setting is given by:

$$N = \mathcal{O}\left(\frac{k_{\mathcal{A}}^2\log^2(k_{\mathcal{A}}/\delta)\log(d)}{\delta^2\varepsilon^2}\right) = \mathcal{O}\left(\frac{\| O \|^2\log^2(M/\delta)\log^2(\| O \| \log(M/\delta)/\varepsilon\delta)\log(d)}{\delta^2\varepsilon^6}\right)$$

where $\| O \| = \max_{i\in[M]}\mathrm{Tr}\left[O_i^2\right]$.

**Proposition 2.** (Classical shadows in the non-i.i.d. setting - Clifford). Let $\{O_i\}_{i\in[M]}$ be $M$ observables. There is an algorithm in the non-i.i.d. setting for predicting the expected values of the observables $\{O_i\}_{i\in[M]}$ under the post-measurement state to within $\varepsilon$ with a copy complexity

$$N = \mathcal{O}\left(\frac{\max_{i\in[M]}\mathrm{Tr}\left[O_i^2\right]^2\log^2(M/\delta)\log^2\left(\max_{i\in[M]}\mathrm{Tr}\left[O_i^2\right]\log(M/\delta)/\varepsilon\delta\right)\log(d)}{\delta^2\varepsilon^6}\right).$$

The algorithm is described in Algorithm 1, where the non-adaptive algorithm/statistic $\mathcal{A}$ is the classical shadows algorithm of ref. 7 and the distribution of measurements is $\mathcal{L}_{\mathbf{Clifford}}$ (or $\mathcal{L}_{\mathbf{Haar}}$).

The second protocol introduced by ref. 7 involves the use of Pauli measurements. This is given by measuring using an orthonormal basis that corresponds to a non-identity Pauli matrix. On the level of the unitary matrix, we can generate this sample by taking $U = u_1 \otimes \cdots \otimes u_N$ where $u_1, \ldots, u_n \overset{\mathrm{iid}}{\sim} \mathrm{Unif}(\mathbf{Cl}(2))$. We denote this distribution by $\mathcal{L}_{\mathbf{Pauli}}$. The classical shadows with Pauli measurement have better performance for estimating expectations of local observables.

**Theorem 7.** (Ref. 7, rephrased). Let $\{O_i\}_{i\in[M]}$ be $M$ $k$-local observables. There is an algorithm for predicting the expected values of the observables $\{O_i\}_{i\in[M]}$ under the state $\sigma$ to within $\varepsilon$ with an error probability $\delta$. This algorithm performs i.i.d. measurements following the distribution $\mathcal{L}_{\mathbf{Pauli}}$, and requires a total number of i.i.d. copies of the state $\sigma$ satisfying:

$$N = \mathcal{O}\left(\frac{2^{2k}\max_{i\in[M]}\| O_i \|_\infty^2\log(M/\delta)}{\varepsilon^2}\right).$$

Now, combining this theorem and Theorem 5, we obtain the following generalization for estimating local properties in the non-i.i.d. setting.

**Proposition 3.** (Classical shadows in the non-i.i.d. setting - Pauli). Let $\{O_i\}_{i\in[M]}$ be $M$ $k$-local observables. There is an algorithm in the non-i.i.d. setting for predicting the expected values of the observables $\{O_i\}_{i\in[M]}$ under the post-measurement state to within $\varepsilon$ with an error probability $\delta$ and a copy complexity satisfying:

$$N = \mathcal{O}\left(\frac{2^{4k}\max_{i\in[M]}\| O_i \|_\infty^4\log^2(M/\delta)\log^2(2^{2k}\log(M)/\varepsilon\delta)\log(d)}{\delta^2\varepsilon^6}\right).$$

Recently, the authors of[53] provide protocols with depth-modulated randomized measurement that interpolates between Clifford and Pauli measurements. Since their algorithms are also non-adaptive, they can be generalized as well to the non-i.i.d. setting using Theorem 5. Other classical shadows protocols[54,56,57,76] could also be extended to the non-i.i.d. setting.

Classical shadows can be used for learning quantum states and unitaries of bounded gate complexity[77]. Our generalization of classical shadows permits to immediately extend the state learning protocol of ref. 77 beyond the i.i.d. assumption and a similar extension should be possible for their unitary learning results.

**Verification of pure states.** The verification of pure states is the task of determining whether a received state precisely matches the ideal pure state or significantly deviates from it. In this context, we will extend

this problem to scenarios where we have $M$ potential pure states represented as $\{|\Psi_i\rangle\langle\Psi_i|\}_{1\le i\le M}$, and our objective is to ascertain whether the received state corresponds to one of these pure states or is substantially different from all of them. The traditional problem constitutes a special case with $M=1$. To formalize, a verification protocol $\mathcal{B}$ satisfies:

1. the completeness condition if it accepts, with high probability, upon receiving one of the pure i.i.d. states $\{|\Psi_i\rangle\langle\Psi_i|^{\otimes N}\}_{1\le i\le M}$, i.e., for all $i \in [M]$, we have $\mathbb{P}_{p\sim\mathcal{B}(|\Psi_i\rangle\langle\Psi_i|^{\otimes N-1})}[p=0]\ge 1-\delta$. Here, the symbol 0 represents the outcome 'Accept' or the null hypothesis.

2. the soundness condition if when the algorithm accepts, the quantum state passing the verification protocol (post-measurement state conditioned on a passing event) is close to one of the pure states $\{|\Psi_i\rangle\langle\Psi_i|\}_{1\le i\le M}$ with high probability, i.e.,

$$\mathbb{P}_{(c,p)\sim\mathcal{B}(\rho^{A_1\cdots A_N})}\left[p=0, \forall i\in[M]: \langle\Psi_i|\rho_{c,0}^{A_N}|\Psi_i\rangle<1-\varepsilon\right]\le\delta. \quad (11)$$

In this latter scenario, the protocol can receive a possibly highly entangled state $\rho^{A_1\cdots A_N}$.

Note that as the prediction for this problem is binary (Accept/Reject), a verification protocol is modeled by an operator $\Pi_{\text{Accept}}$, which is given by $\mathcal{B}^\dagger(|0\rangle\langle 0|)$. The usual way (see e.g., refs. [21,35,36,61–63]) of writing the completeness and soundness conditions of a protocol for the case $M=1$ of verifying a single pure state is as follows. The completeness condition is

$$\text{Tr}\left[\Pi_{\text{Accept}}|\Psi\rangle\langle\Psi|^{\otimes N-1}\right]\ge 1-\delta_c,$$

where $\delta_c$ is the completeness parameter, which is the same as what we expressed in terms of $\mathcal{B}$. The soundness condition is

$$\text{Tr}\left[\Pi_{\text{Accept}}\otimes(\mathbb{I}-|\Psi\rangle\langle\Psi|)\rho^{A_1\cdots A_N}\right]\le\delta_s. \quad (12)$$

Note that this quantity evaluates the expected infidelity of the state conditioned on acceptance, whereas Eq. (11) is slightly different: it evaluates the probability (over $p$ and $c$) of having a fidelity below $1-\varepsilon$. It is simple to see that Eq. (11) implies $\delta_s\le\varepsilon+\delta$. Conversely, using Markov's inequality, Eq. (12) implies Eq. (11) with $\varepsilon=\delta=\sqrt{\delta_s}$. We can, using the same methods, express our findings directly in terms of expectations for the task of verifying one pure state, see Supplementary Note 4 for more details. Here we prove the following verification result with high probability.

**Proposition 4.** (Verification of pure states in the non-i.i.d. setting - Clifford). Let $\rho^{A_1\cdots A_N}$ be a permutation invariant state. Let $\{|\Psi_i\rangle\langle\Psi_i|\}_{1\le i\le M}$ be $M$ pure states. There is an algorithm using Clifford measurements for verifying whether the (post-measurement) state $\rho^{A_N}$ is a member of $\{|\Psi_i\rangle\langle\Psi_i|\}_{1\le i\le M}$ or is at least $\varepsilon$-far from them in terms of fidelity with a probability at least $1-\delta$ and a number of copies satisfying

$$N=\mathcal{O}\left(\frac{\log^2(M/\delta)\log^2(\log(M)/\varepsilon\delta)\log(d)}{\delta^2\varepsilon^6}\right).$$

Proof. We can apply Proposition 2 to estimate the expectation of the observables $\{O_i=|\Psi_i\rangle\langle\Psi_i|\}_{1\le i\le M}$ under the post-measurement state $\rho_{l,\mathbf{r},\mathbf{w}}^{A_N}$ to within $\varepsilon/4$ and with a probability at least $1-\delta$ using a number of copies $N=\mathcal{O}\left(\frac{\log^2(M/\delta)\log^2(\log(M)/\varepsilon\delta)\log(d)}{\delta^2\varepsilon^6}\right)$. More concretely, we have a set of predictions $\boldsymbol{\mu}=\{\mu_i\}_{1\le i\le M}$ satisfying (Proposition 2 and

Lemma 1):

$$\mathbb{P}_{l,\mathbf{r},\mathbf{w},\boldsymbol{\mu}}\left[\forall i\in[M]:\left|\mu_i-\text{Tr}\left[|\Psi_i\rangle\langle\Psi_i|\rho_{l,\mathbf{r},\mathbf{w},\boldsymbol{\mu}}^{A_N}\right]\right|\le\varepsilon/4, \left\|\rho_{l,\mathbf{r},\mathbf{w},\boldsymbol{\mu}}^{A_N}-\rho_{l,\mathbf{r},\mathbf{w}}^{A_N}\right\|_1\le\varepsilon/8\right]\ge 1-\delta.$$

$$(13)$$

Then, our proposed algorithm accepts if, and only if there is some $i\in[M]$ such that $\mu_i\ge 1-\varepsilon/2$. We can verify the completeness and soundness conditions for this algorithm.

1. Completeness. If the verifier receives one pure state of the form $\rho^{A_1\cdots A_N}=|\Psi_i\rangle\langle\Psi_i|^{\otimes N}$ for some $i\in[M]$ then every post-measurement state is pure, i.e., $\rho_{l,\mathbf{r},\mathbf{w},\boldsymbol{\mu}}^{A_N}=|\Psi_i\rangle\langle\Psi_i|$ and Inequality (13) implies $\mathbb{P}_{\boldsymbol{\mu}}[|\mu_i-1|\le\varepsilon/2]=\mathbb{P}_{\boldsymbol{\mu}}[|\mu_i-\text{Tr}[|\Psi_i\rangle\langle\Psi_i|\rho^{A_N}]|\le\varepsilon/2]\ge 1-\delta$. Hence the algorithm accepts with a probability $\ge\mathbb{P}_{\boldsymbol{\mu}}[\mu_i\ge 1-\varepsilon/2]\ge 1-\delta$. Observe that for this algorithm, we can even relax the assumption that the input state is i.i.d.. For instance, we can only ask that the input state is product $\rho^{A_1\cdots A_N}=\otimes_{t=1}^N\sigma_t$ where for all $t\in[N]$, $\langle\Psi_i|\sigma_t|\Psi_i\rangle\ge 1-\varepsilon/4$.

2. Soundness. Here, we want to prove the following:

$$\mathbb{P}_{l,\mathbf{r},\mathbf{w},\boldsymbol{\mu}}\left[\mathcal{B}(\rho^{A_1\cdots A_N})=0, \forall i\in[M]:\langle\Psi_i|\rho_{l,\mathbf{r},\mathbf{w},0}^{A_N}|\Psi_i\rangle<1-\varepsilon\right]\le\delta.$$

If $\mathcal{B}(\rho)=0$ then for some $j\in[M]$ we have $\mu_j\ge 1-\varepsilon/2$. Hence $\langle\Psi_j|\rho_{l,\mathbf{r},\mathbf{w},0}^{A_N}|\Psi_j\rangle<1-\varepsilon$ implies $\langle\Psi_j|\rho_{l,\mathbf{r},\mathbf{w},\boldsymbol{\mu}}^{A_N}|\Psi_j\rangle\le\langle\Psi_j|\rho_{l,\mathbf{r},\mathbf{w}}^{A_N}|\Psi_j\rangle+\varepsilon/8\le\langle\Psi_j|\rho_{l,\mathbf{r},\mathbf{w},0}^{A_N}|\Psi_j\rangle+\varepsilon/4<\mu_j-\varepsilon/4$ therefore:

$$\mathbb{P}_{l,\mathbf{r},\mathbf{w},\boldsymbol{\mu}}\left[\mathcal{B}(\rho^{A_1\cdots A_N})=0, \forall i\in[M]:\langle\Psi_i|\rho_{l,\mathbf{r},\mathbf{w},0}^{A_N}|\Psi_i\rangle<1-\varepsilon\right]$$
$$\le\mathbb{P}_{l,\mathbf{r},\mathbf{w},\boldsymbol{\mu}}\left[\exists j\in[M]:\mu_j\ge 1-\varepsilon/2, \langle\Psi_j|\rho_{l,\mathbf{r},\mathbf{w},0}^{A_N}|\Psi_j\rangle<1-\varepsilon\right]$$
$$\le\mathbb{P}_{l,\mathbf{r},\mathbf{w},\boldsymbol{\mu}}\left[\exists j\in[M]:\langle\Psi_j|\rho_{l,\mathbf{r},\mathbf{w},\boldsymbol{\mu}}^{A_N}|\Psi_j\rangle<\mu_j-\varepsilon/4\right]$$
$$\le\mathbb{P}_{l,\mathbf{r},\mathbf{w},\boldsymbol{\mu}}\left[\exists j\in[M]:\left|\mu_j-\text{Tr}\left[|\Psi_j\rangle\langle\Psi_j|\rho_{l,\mathbf{r},\mathbf{w},\boldsymbol{\mu}}^{A_N}\right]\right|>\varepsilon/4\right]$$
$$\le\delta$$

where we used Inequality Eq. (13).

The above result uses Clifford measurements, which are non-local. If our primary concern lies in verification with local measurements, an alternative approach would be to apply the non-i.i.d. shadow tomography result for local measurements (Proposition 3). Using the same analysis of this section, we can prove the following proposition.

**Proposition 5.** (Verification of pure states in the non-i.i.d. setting - Pauli). Let $\rho^{A_1\cdots A_N}$ be a permutation invariant state. Let $\{|\Psi_i\rangle\langle\Psi_i|\}_{1\le i\le M}$ be $M$ pure states. There is an algorithm using local (pauli) measurements for verifying whether the (post-measurement) state $\rho^{A_N}$ is a member of $\{|\Psi_i\rangle\langle\Psi_i|\}_{1\le i\le M}$ or is at least $\varepsilon$-far from them in terms of fidelity with a probability at least $1-\delta$ and a number of copies satisfying

$$N=\mathcal{O}\left(\frac{n^3 2^{4n}\log^2(M/\delta)\log^2(\log(M)/\varepsilon\delta)}{\delta^2\varepsilon^6}\right).$$

**Discussion and comparison with previous works on verification of pure states.** The main contribution here compared to previous results is that we give the first explicit protocol which works for all multipartite states. This stands in contrast to previous protocols where the desired state must be a ground state of a Hamiltonian satisfying certain conditions[61] or a graph state[35,36,62,63], or Dicke states[21]. However, the more efficient protocol uses Clifford measurements, which are nonlocal. The Pauli measurement case is local, but comes at a cost in scaling with number of systems.

We now go into more detail regarding the different scalings. The optimal copy complexity, or scaling for the number of copies required, with the fidelity error $\varepsilon$, is $1/\varepsilon^2$[21,78]. The scaling with the number of systems $n$ depends on the protocol (e.g. for stabilizer states there are protocols that do not scale with $n$, but known protocols for the W state scales with $n$[21]). Applying our results using Clifford (i.e. entangled over the systems) gives scaling with $\varepsilon$ and $n$ as $\tilde{\mathcal{O}}(n/\varepsilon^6)$, and for random local Pauli scaling (local) the scaling is $\tilde{\mathcal{O}}(n^3 16^n/\varepsilon^6)$. For the Clifford protocol, then, we have similar scaling to optimal known for W states (though with $\varepsilon$ scaling as $1/\varepsilon^6$ instead of $1/\varepsilon$), but our protocol works for all states. The cost here is that measurements are in non-local across each copy. However for certain applications this is not an issue. For example verifying output of computations, Clifford are reasonably within the sets of easy gates, so we have a close to optimal verification for all states that can be implemented. In the case of random Paulis, where measurements are local on copies, we have the same scaling with $\varepsilon$ but we get an exponential penalty of $n$ scaling in the error. Given the generality of our protocol to all states though, it is perhaps not so surprising that we have a high dimensional cost. Furthermore, depending on the situation, this scaling may not be the major cost one cares about. Indeed, for small networks dimension will not be the most relevant scaling. We can imagine many applications in this regime. For example small networks of sensors, such as satellites or gravimeters[18,31], this scaling would not be prohibitive, but our results would allow for different resource states to be used, for example spin squeezed states, or other symmetric states which exhibit better robustness to noise[79]. Another example would be small communication networks, where, for example GHZ states can be used for anonymous communication[29] or W states for leader election[80]. On such small scale networks our results would allow for verified versions of these applications over untrusted networks, in a way that is blind to which communication protocol is being applied.

We also point out that we have not optimized over these numbers (rather we were concerned with showing something that works for all states). It is highly likely that these complexities can be improved and we expect that for particular families of states one can find variants where the scaling in the number of systems is polynomial or better. One perspective in this direction coming directly from our results, is the observation that the protocols in the framework of ref. 5, which assume i.i.d. states, use random i.i.d. measurements, therefore our theorem allows them to be applied directly to the non-i.i.d. case. This allows us to take any protocol assuming i.i.d. states, and it works for general (non-i.i.d.) sources with a small cost.

Lastly, our formulation is naturally robust to noise. Such robustness is an important issue for any practical implementation, and indeed it has been addressed for several of the protocols mentioned, see for example[35,63,81,82]. In terms of the completeness condition, we can easily make out statements robust to noise. For instance, we can relax the requirement to only ask that the input state is a product state $\rho^{A_1 \cdots A_N} = \otimes_{t=1}^N \sigma_t$ where for all $t \in [N]$, $\langle \Psi_i | \sigma_t | \Psi_i \rangle \geq 1 - \varepsilon/4$.

**Fidelity estimation.** The problem of direct fidelity estimation[1,83] consists of estimating the fidelity $\langle \Psi | \rho | \Psi \rangle$ between the target known pure state $|\Psi\rangle\langle\Psi|$ and the unknown quantum state $\rho$ by measuring independent copies of $\rho$. The algorithm of ref. 1 proceeds by sampling i.i.d. random Pauli matrices

$$P_1, \ldots, P_l \sim \left\{ \frac{\langle \Psi | P | \Psi \rangle^2}{d} \right\}_{P \in \{\mathbb{I}, X, Y, Z\}^{\otimes n}}$$

where $l = \lceil 1/(\varepsilon^2 \delta) \rceil$. Then for each $i = 1, \ldots, l$, the algorithm measures the state $\rho$ with the POVM $\mathcal{M}_{P_i} = \left\{ \frac{\mathbb{I} - P_i}{2}, \frac{\mathbb{I} + P_i}{2} \right\}$ $m_i$ times where $m_i$ is defined as

$$m_i = \left\lceil \frac{2 \log(2/\delta)\delta}{\langle \Psi | P_i | \Psi \rangle^2} \right\rceil.$$

The algorithm observes $A_{i,j} \sim \left\{ \frac{1 - \mathrm{Tr}[P_i \rho]}{2}, \frac{1 + \mathrm{Tr}[P_i \rho]}{2} \right\}$ where $i \in \{1, \ldots, l\}$ and $j \in \{1, \ldots, m_i\}$. The estimator of the fidelity is then given as follows

$$S = \frac{1}{l} \sum_{i=1}^l \frac{1}{m_i \langle \Psi | P_i | \Psi \rangle} \sum_{j=1}^{m_i} (2A_{i,j} - 1).$$

In general, in ref. 1, it is proven that the copy complexity satisfies:

$$\mathbb{E}\left[ \sum_{i=1}^l m_i \right] \leq \left( 1 + \frac{12}{\varepsilon^2} + \frac{2d}{\varepsilon^2} \log(24) \right)$$

to conclude that $|S - \langle \Psi | \rho | \Psi \rangle| \leq 2\varepsilon$ with probability at least 5/6. This algorithm is non-adaptive and performs independent measurements from the set:

$$\mathcal{M}_{P_1, \ldots, P_l} = \bigcup_{i=1}^{\lceil 12/\varepsilon^2 \rceil} \left\{ \mathcal{M}_{P_i} \text{ repeated } \left\lceil \frac{2 \log(2/\delta)\delta}{\langle \Psi | P_i | \Psi \rangle^2} \right\rceil \text{ times} \right\} \text{ where } P_1, \ldots, P_l \sim \left\{ \frac{\langle \Psi | P | \Psi \rangle^2}{d} \right\}_{P \in \{\mathbb{I}, X, Y, Z\}^{\otimes n}}$$

To extend this result to the non-i.i.d. setting, we apply Theorem 5 with the set of measurements $\mathcal{M}_{P_1, \ldots, P_l}$ and a copy complexity given by $k_{\mathcal{A}} = \sum_{i=1}^l m_i = \sum_{i=1}^{\lceil 12/\varepsilon^2 \rceil} \lceil \frac{\log(24)}{6 \langle \Psi | P_i | \Psi \rangle^2} \rceil$. Theorem 5 ensures that we can estimate the fidelity between the ideal state $|\Psi\rangle\langle\Psi|$ and the post-measurement state $\rho_{\mathbf{w}}^{A_N}$ to within $3\varepsilon$ with probability at least 5/6 if the total number of copies $N$ satisfies:

$$N = \frac{48^2 \log(d)}{\varepsilon^2} \cdot k_{\mathcal{A}}^2 \log^2(18 k_{\mathcal{A}}).$$

By Markov's inequality we have with probability at least 5/6:

$$k_{\mathcal{A}} \leq 6 \, \mathbb{E}\left[ \sum_{i=1}^l m_i \right] \leq 6 \left( 1 + \frac{12}{\varepsilon^2} + \frac{2d}{\varepsilon^2} \log(24) \right) \leq \frac{12^2 d}{\varepsilon^2}.$$

Therefore, by the union bound, our non-i.i.d. algorithm is 1/3-correct and its complexity satisfies:

$$N \leq \frac{48^2 \cdot 12^2 d^2 \log^2(18 \cdot 12^2 d/\varepsilon^2) \log(d)}{\varepsilon^6} = \mathcal{O}\left( \frac{d^2 \log^3(d/\varepsilon)}{\varepsilon^6} \right).$$

**Proposition 6.** (Fidelity estimation in the non-i.i.d. setting). There is an algorithm in the non-i.i.d. setting for fidelity estimation with a precision parameter $\varepsilon$, a success probability at least 2/3 and a copy complexity:

$$N = \mathcal{O}\left( \frac{d^2 \log^3(d/\varepsilon)}{\varepsilon^6} \right).$$

Moreover, in ref. 1, it is shown that for well-conditioned states $|\Psi\rangle\langle\Psi|$ satisfying for all $P \in \{\mathbb{I}, X, Y, Z\}^{\otimes n}$, $|\langle \Psi | P | \Psi \rangle| \geq \alpha$ for some $\alpha > 0$, the copy complexity is bounded in expectation as follows:

$$\mathbb{E}[k_{\mathcal{A}}] = \mathbb{E}\left[ \sum_{i=1}^l m_i \right] = \mathcal{O}\left( \frac{\log(12)}{\alpha^2 \varepsilon^2} \right).$$

Similarly, by applying Theorem 5 and Markov's inequality we can show the following proposition.

**Proposition 7.** (Fidelity estimation in the non-i.i.d. setting - Well-conditioned states). Let $|\Psi\rangle$ be a well-conditioned state with parameter $\alpha > 0$. There is an algorithm in the non-i.i.d. setting for fidelity estimation with a precision parameter $\varepsilon$, a success probability at least 2/3

and a copy complexity:

$$N = \mathcal{O}\left(\frac{\log^3(d/\alpha\varepsilon)}{\alpha^4\varepsilon^6}\right).$$

**State tomography.** In the problem of state tomography, we are given $N$ copies of an unknown quantum state $\sigma$ and the objective is to construct a (classical description) of a quantum state $\hat{\sigma}$ satisfying $\|\sigma - \hat{\sigma}\|_1 \leq \varepsilon$ with a probability at least $1 - \delta$.

In the i.i.d. setting, a sufficient number of copies for state tomography in the incoherent setting with a precision $\varepsilon$ and an error probability $\delta$ is[4]:

$$k_{\mathcal{A}} = \mathcal{O}\left(\frac{d^2\log(1/\delta)}{\varepsilon^2} + \frac{d^3}{\varepsilon^2}\right).$$

Hence by Theorem 5 there is an algorithm $\mathcal{B}$ in the non-i.i.d. setting with an error probability:

$$\delta_{\mathcal{B}}(N, \rho^{A_1\cdots A_N}, 2\varepsilon) \leq 2\sup_{\sigma:\text{ state}} \delta_{\mathcal{A}}\left(k_{\mathcal{A}}, \sigma^{\otimes k_{\mathcal{A}}}, \varepsilon\right) + 6\sqrt{\frac{k_{\mathcal{A}}^2\log^2(k_{\mathcal{A}}/\delta_{\mathcal{A}})\log(d)}{N\varepsilon^2}}.$$

So a total number of copies sufficient to achieve $\delta$-correctness in the non-i.i.d. setting is:

$$N = \frac{256 k_{\mathcal{A}}^2 \log^2(6k_{\mathcal{A}}/\delta)\log(d)}{\delta^2\varepsilon^2}$$
$$= \mathcal{O}\left(\frac{d^4\log^2(d/\delta\varepsilon)\log^2(1/\delta)\log(d)}{\delta^2\varepsilon^6} + \frac{d^6\log^2(d/\delta\varepsilon)\log(d)}{\delta^2\varepsilon^6}\right).$$

**Proposition 8.** (State tomography in the non-i.i.d. setting). There is an algorithm in the non-i.i.d. setting for state tomography with a precision parameter $\varepsilon$, a success probability at least $1 - \delta$ and a copy complexity:

$$N = \mathcal{O}\left(\frac{d^4\log^5(d/\delta\varepsilon)}{\delta^2\varepsilon^6} + \frac{d^6\log^3(d/\delta\varepsilon)}{\delta^2\varepsilon^6}\right).$$

Observe that, unlike the statement of state tomography in the i.i.d. setting[4], here we do not have an explicit dependency on the rank of the approximated state. This can be explained by the fact that if the state $\rho^{A_1\cdots A_N}$ is not i.i.d. then the post-measurement states $\{\rho_{c,p}^{A_N}\}_{c,p}$ can have a full rank even if we start with a pure input state $\rho^{A_1\cdots A_N}$. For instance, let $\rho = |\Psi\rangle\langle\Psi|$ where $|\Psi\rangle = \frac{1}{\sqrt{d}}\sum_{i\in[d]}|i\rangle \otimes |i\rangle$ is the maximally entangled state, and let $X = \sum_{i\in[d]}\alpha_i|i\rangle\langle i|$ be an observable. In this case, we have rank$(\rho^{A_1A_2}) = 1$ and rank$(\rho_X^{A_2}) = $ rank$\left(\sum_{i\in[d]}\frac{\alpha_i}{\|\alpha\|_1}|i\rangle\langle i|\right) = d$ if all the coefficients $\{\alpha_i\}_{i\in[d]}$ are non-zero.

**Testing mixedness of states.** In the problem of testing mixedness of states, we are given an unknown quantum state $\sigma$, which can either be $\frac{1}{d}$ (null hypothesis) or $\varepsilon$-far from it in the trace-norm (alternate hypothesis). The objective is to determine the true hypothesis with a probability of at least $1 - \delta$. However, this problem does not satisfy the robustness assumption required in Definition 3. Due to this reason, we introduced the tolerant version of this problem in Example 1. To the best of our knowledge, there is no algorithm for the tolerant testing mixedness problem that outperforms the tomography algorithm (naive testing by learning approach). Thus, in this section, we concentrate on the standard (non-tolerant) formulation of testing mixedness of states.

Under the null hypothesis, we assume that the learning algorithm is given the i.i.d. state $\rho = \left(\frac{1}{d}\right)^{\otimes N}$ and is expected to respond with

0 with a probability of at least $1 - \delta$. On the other hand, under the alternate hypothesis, the learning algorithm receives a (potentially entangled) state $\rho^{A_1\cdots A_N}$. In this scenario, the learning algorithm should output 1 with a probability of at least $1-\delta$ if the post-measurement state $\rho_{c,p}^{A_N}$ is $\varepsilon$-far from $\frac{1}{d}$. In the i.i.d. case, a sufficient number of copies for testing mixedness of states problem in the incoherent setting with a precision parameter $\varepsilon$ and an error probability $\delta$ is given by ref. 27:

$$k_{\mathcal{A}} = \mathcal{O}\left(\frac{\sqrt{d^3}\log(1/\delta)}{\varepsilon^2}\right).$$

Hence by Theorem 5

$$\delta_{\mathcal{B}}(N, \rho^{A_1\cdots A_N}, 2\varepsilon) \leq 2\sup_{\sigma:\text{ state}} \delta_{\mathcal{A}}\left(k_{\mathcal{A}}, \sigma^{\otimes k_{\mathcal{A}}}, \varepsilon\right) + 6\sqrt{\frac{k_{\mathcal{A}}^2\log^2(k_{\mathcal{A}}/\delta_{\mathcal{A}})\log(d)}{N\varepsilon^2}}. \tag{14}$$

We can apply Theorem 5 only under the alternate hypothesis where the robustness assumption holds. Under the null hypothesis, the robustness assumption no longer holds; however, since we are assuming that the input state is i.i.d., i.e., $\rho = \left(\frac{1}{d}\right)^{\otimes N}$, we can directly apply the result from ref. 27 in this case. So, from Eq. (14), we deduce that a total number of copies sufficient to achieve $\delta$-correctness in the non-i.i.d. setting is:

$$N = \frac{256 k_{\mathcal{A}}^2 \log^2(6k_{\mathcal{A}}/\delta\varepsilon)\log(d)}{\delta^2\varepsilon^2} = \mathcal{O}\left(\frac{d^3\log^2(1/\delta)\log^2(d/\delta\varepsilon)\log(d)}{\delta^2\varepsilon^6}\right).$$

**Proposition 9.** (Testing mixedness of quantum states in the non-i.i.d. setting). There is an algorithm in the non-i.i.d. setting for testing mixedness of quantum states with a precision parameter $\varepsilon$, a success probability at least $1-\delta$ and a copy complexity:

$$N = \mathcal{O}\left(\frac{d^3\log^5(d/\delta\varepsilon)}{\delta^2\varepsilon^6}\right).$$

## General algorithms in the non-i.i.d. setting

In this section, we present a general framework for extending algorithms designed to learn properties of a quantum state using i.i.d. input states, to general possibly entangled input states. The distinction from Methods' subsection "Non-adaptive algorithms in the non-i.i.d. setting" lies in the relaxation of the requirement for algorithms to be non-adaptive; meaning, they can now involve adaptive measurements, potentially coherent or entangled (see Definition 4). Coherent measurements are proved to be more powerful than incoherent ones (let alone non-adaptive ones) for tasks such as state tomography[3,49], shadow tomography[6,9,84] and testing mixedness of states[10,85].

As we now consider general algorithms that encompass (possibly) coherent measurements, a suitable candidate for the measurement device in the projection phase (the **w** part in Algorithm 1) becomes less clear. Furthermore, we require an approximation that excels under the more stringent trace-norm condition, particularly when addressing non-local (non product) observables. To address this challenge, we adopt the approach outlined in ref. 23, utilizing any informationally complete measurement device. We will use the measurement device $\mathcal{M}_{\text{dist}}$, having a low distortion with side information, of ref. 86. It satisfies the following important property: the application of the corresponding measurement channel $\mathcal{M}_{\text{dist}}$ to the system $A_2$ does not diminish the distinguishability between two bipartite states on $A_1A_2$ by a factor greater than $2d_{A_2}$, wherein $d_{A_2}$ represents the dimension of $A_2$. To be precise, the measurement channel $\mathcal{M}_{\text{dist}}$ satisfies the following

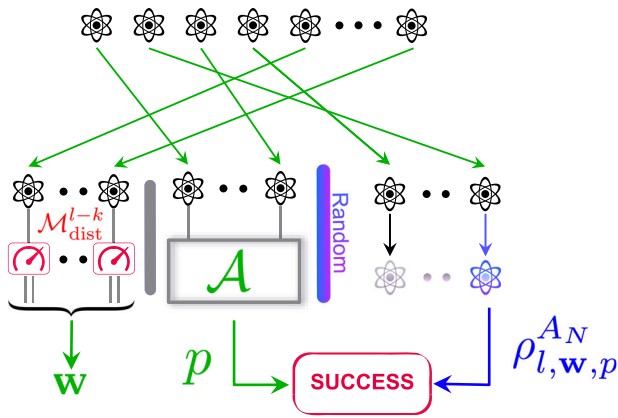

**Fig. 5 | Illustration of Algorithm 2.** Algorithm 2 measures a large number of the state's subsystems using the measurement device with low distortion $\mathcal{M}_{\text{dist}}^{l-k}$ (red and green parts). Then, in order to predict the property, Algorithm 2 applies the data processing of Algorithm $\mathcal{A}$ to the outcomes of a part these subsystems (green part) leading to a prediction $p$. Algorithm 2 returns the remaining outcomes as calibration $\mathbf{w}$. Success occurs if $p$ is (approximately) compatible with the remaining post-measurement test copy $\rho_{l,\mathbf{w},p}^{A_N}$.

inequality for all bipartite states $\rho^{A_1 A_2}$ and $\sigma^{A_1 A_2}$:

$$\left\| \rho^{A_1 A_2} - \sigma^{A_1 A_2} \right\|_1 \le 2 d_{A_2} \left\| \text{id}^{A_1} \otimes \mathcal{M}_{\text{dist}}^{A_2} (\rho^{A_1 A_2} - \sigma^{A_1 A_2}) \right\|_1 .$$

The measurement device $\mathcal{M}_{\text{dist}}$ will play a crucial role in our algorithm. By applying this channel to a large fraction of the subsystems of a quantum state, we can show that the post-measurement state behaves as an i.i.d. state. Thus, we will be able to use the same algorithm on a small number of the remaining systems.

For a learning algorithm $\mathcal{A}$ designed for i.i.d. inputs, we construct the algorithm $\mathcal{B}$ explicitly described in Algorithm 2 (displayed in Box 2) and illustrated in Fig. 5.

In the following theorem, we relate the error probability of Algorithm 2 with the error probability of the algorithm $\mathcal{A}$.

**Theorem 8.** (General algorithms in the non-i.i.d. setting) Let $N \ge 1$ be a positive integer and $A_1 \cong A_2 \cong \cdots \cong A_N$ be $N$ isomorphic quantum systems of dimension $d$. Let $\varepsilon, \varepsilon' > 0$ and $1 \le k < N/2$. Let $\mathcal{A}$ be a general algorithm. Algorithm 2 has an error probability satisfying:

$$\delta_{\mathcal{B}}(N, \rho^{A_1\cdots A_N}, \varepsilon + \varepsilon') \le \sup_{l,\mathbf{w}} \delta_{\mathcal{A}}\left(k, \left(\rho_{l,\mathbf{w}}^{A_N}\right)^{\otimes k}, \varepsilon\right) + 12\sqrt{\frac{2k^3 d^2 \log(d)}{N\varepsilon'^2}} + 2\sqrt{\frac{2k^3 d^2 \log(d)}{N}} .$$

**Remark 6.** To achieve an error probability of at most $\delta$, one could start by determining a value for $k(\mathcal{A}, \delta, \varepsilon)$ such that for all $\mathbf{w}$, $\delta_{\mathcal{A}}(k, (\rho_{\mathbf{w}}^{A_N})^{\otimes k}, \varepsilon/2) \le \delta/2$. Subsequently, the total number of copies can be set to

$$N = \frac{32 \cdot 14^2 d^2 \log(d)}{\delta^2 \varepsilon^2} \cdot k(\mathcal{A}, \delta, \varepsilon)^3 .$$

This choice of sample complexity ensures that $\delta_{\mathcal{B}}(N, \rho^{A_1\cdots A_N}, \varepsilon) \le \delta$, as desired.

In what follows we proceed to prove Theorem 8.

Proof of Theorem 8. First, since we are using the informationally complete measurement device $\mathcal{M}_{\text{dist}}$, we can relate the difference between post-measurement states and the actual states. This along with an information theoretical analysis using the mutual information show that measuring using $\mathcal{M}_{\text{dist}}$ a sufficiently large number of times, transforms the state approximately to an i.i.d. one. Infact, the proof of

Theorem 2.4. of ref. 23 together with the distortion with side information measurement device $\mathcal{M}_{\text{dist}}$ of ref. 86 imply that for $k < N/2$:

**Lemma 2.** (Ref. 23, rephrased) Let $\rho^{A_1\cdots A_N}$ be a permutation invariant state. For $k < N/2$, we have

$$\frac{2}{N} \sum_{l=k+1}^{k+N/2} \mathbb{E}_{\mathbf{w} \sim \mathcal{M}_{\text{dist}}^{l-k}(\rho)} \left[ \left\| \rho_{\mathbf{w}}^{A_1\cdots A_k} - \left(\rho_{\mathbf{w}}^{A_N}\right)^{\otimes k} \right\|_1 \right] \le 2\sqrt{\frac{2k^3 d^2 \log(d)}{N}} .$$

where $\mathbf{w} = (w_{k+1}, ..., w_l)$ is the outcome of measuring each of the systems $A_{k+1}...A_l$ with the measurement $\mathcal{M}_{\text{dist}}$.

We write the error probability as

$$
\begin{aligned}
\delta_{\mathcal{B}}(N, \rho^{A_1\cdots A_N}, \varepsilon + \varepsilon') &= \mathbb{P}_{l,\mathbf{w},p}\left[ (p, \rho_{l,\mathbf{w},p}^{A_N}) \notin \text{SUCCESS}_{\varepsilon+\varepsilon'} \right] \\
&= \mathbb{P}_{l,\mathbf{w},p}\left[ (p, \rho_{l,\mathbf{w},p}^{A_N}) \notin \text{SUCCESS}_{\varepsilon+\varepsilon'}, \left\| \rho_{l,\mathbf{w},p}^{A_N} - \rho_{l,\mathbf{w}}^{A_N} \right\|_1 \le \varepsilon' \right] \\
&\quad + \mathbb{P}_{l,\mathbf{w},p}\left[ (p, \rho_{l,\mathbf{w},p}^{A_N}) \notin \text{SUCCESS}_{\varepsilon+\varepsilon'}, \left\| \rho_{l,\mathbf{w},p}^{A_N} - \rho_{l,\mathbf{w}}^{A_N} \right\|_1 > \varepsilon' \right] \\
&\le \mathbb{P}_{l,\mathbf{w},p}\left[ (p, \rho_{l,\mathbf{w}}^{A_N}) \notin \text{SUCCESS}_{\varepsilon} \right] + \mathbb{P}_{l,\mathbf{w},p}\left[ \left\| \rho_{l,\mathbf{w},p}^{A_N} - \rho_{l,\mathbf{w}}^{A_N} \right\|_1 > \varepsilon' \right],
\end{aligned}
\tag{15}
$$

where we use the robustness condition. Using Lemma 2 and the triangle inequality, the first term can be bounded as follows:

$$
\begin{aligned}
&\mathbb{P}_{l,(p,\mathbf{w}) \sim (\mathcal{A} \otimes \mathcal{M}_{\text{dist}}^{\otimes(l-k)})(\rho^{A_1\cdots A_l})}\left[ (p, \rho_{l,\mathbf{w}}^{A_N}) \notin \text{SUCCESS}_{\varepsilon} \right] \\
&= \mathbb{E}_{l,\mathbf{w}}\left[ \mathbb{P}_{p \sim \mathcal{A}(\rho_{l,\mathbf{w}}^{A_1\cdots A_k})}\left[ (p, \rho_{l,\mathbf{w}}^{A_N}) \notin \text{SUCCESS}_{\varepsilon} \right] \right] \\
&\le \mathbb{E}_{l,\mathbf{w}}\left[ \mathbb{P}_{p \sim \mathcal{A}\left((\rho_{l,\mathbf{w}}^{A_N})^{\otimes k}\right)}\left[ (p, \rho_{l,\mathbf{w}}^{A_N}) \notin \text{SUCCESS}_{\varepsilon} \right] \right] + 2\sqrt{\frac{2k^3 d^2 \log(d)}{N}} \\
&\le \sup_{l,\mathbf{w}} \delta_{\mathcal{A}}\left(k, \left(\rho_{l,\mathbf{w}}^{A_N}\right)^{\otimes k}, \varepsilon\right) + 2\sqrt{\frac{2k^3 d^2 \log(d)}{N}}
\end{aligned}
$$

For the second term of Eq. (15), we apply the following lemma:

**Lemma 3.** Let $\varepsilon' > 0$, $1 \le k < N/2$ and $l \sim \text{Unif}\{k+1, ..., k+N/2\}$. Let $\mathbf{w} = (w_{k+1}, ..., w_l)$ and $p$ be the outcomes of measuring the state $\rho$ with the measurement $\mathcal{M}_{\text{dist}}^{\otimes(l-k)}$ on systems $A_{k+1}...A_l$ and $\mathcal{A}$ on $A_1...A_k$. The following inequality holds:

$$\mathbb{P}_{l,\mathbf{w},p}\left[ \left\| \rho_{l,\mathbf{w},p}^{A_N} - \rho_{l,\mathbf{w}}^{A_N} \right\|_1 > \varepsilon' \right] \le 12\sqrt{\frac{2k^3 d^2 \log(d)}{N\varepsilon'^2}} .$$

Proof. Denote by $\{M_p\}_p$ the elements of the POVM corresponding to $\mathcal{A}$. Lemma 2 together with the triangle inequality imply:

$$
\begin{aligned}
\mathbb{E}_{l,\mathbf{w},p}\left[ \left\| \rho_{l,\mathbf{w},p}^{A_N} - \rho_{l,\mathbf{w}}^{A_N} \right\|_1 \right] &= \mathbb{E}_{l,\mathbf{w}}\left[ \sum_p \text{Tr}\left[ M_p \rho_{l,\mathbf{w}}^{A_1\cdots A_k} \right] \left\| \rho_{l,\mathbf{w},p}^{A_N} - \rho_{l,\mathbf{w}}^{A_N} \right\|_1 \right] \\
&= \mathbb{E}_{l,\mathbf{w}}\left[ \sum_p \left\| \text{Tr}\left[ M_p \rho_{l,\mathbf{w}}^{A_1\cdots A_k} \right] \rho_{l,\mathbf{w},p}^{A_N} - \text{Tr}\left[ M_p \rho_{l,\mathbf{w}}^{A_1\cdots A_k} \right] \rho_{l,\mathbf{w}}^{A_N} \right\|_1 \right] \\
&\le \mathbb{E}_{l,\mathbf{w}}\left[ \sum_p \left\| \text{Tr}\left[ M_p \rho_{l,\mathbf{w}}^{A_1\cdots A_k} \right] \rho_{l,\mathbf{w},p}^{A_N} - \text{Tr}\left[ M_p (\rho_{l,\mathbf{w}}^{A_N})^{\otimes k} \right] \rho_{l,\mathbf{w}}^{A_N} \right\|_1 \right] \\
&\quad + \mathbb{E}_{l,\mathbf{w}}\left[ \sum_p \left\| \text{Tr}\left[ M_p (\rho_{l,\mathbf{w}}^{A_N})^{\otimes k} \right] \rho_{l,\mathbf{w}}^{A_N} - \text{Tr}\left[ M_p \rho_{l,\mathbf{w}}^{A_1\cdots A_k} \right] \rho_{l,\mathbf{w}}^{A_N} \right\|_1 \right] \\
&= \mathbb{E}_{l,\mathbf{w}}\left[ \sum_p \left\| \text{Tr}_{A_1\cdots A_k}\left[ M_p \otimes \mathbb{I}\left( \rho_{l,\mathbf{w}}^{A_1\cdots A_{k+1}} - \left(\rho_{l,\mathbf{w}}^{A_N}\right)^{\otimes k+1} \right) \right] \right\|_1 \right] \\
&\quad + \mathbb{E}_{l,\mathbf{w}}\left[ \sum_p \left| \text{Tr}\left[ M_p (\rho_{l,\mathbf{w}}^{A_N})^{\otimes k} \right] - \text{Tr}\left[ M_p \rho_{l,\mathbf{w}}^{A_1\cdots A_k} \right] \right| \right] \\
&\le 2\,\mathbb{E}_{l,\mathbf{w}}\left[ \left\| \rho_{l,\mathbf{w}}^{A_1\cdots A_{k+1}} - \left(\rho_{l,\mathbf{w}}^{A_N}\right)^{\otimes k+1} \right\|_1 \right] \\
&\le 4\sqrt{\frac{2(k+1)^3 d^2 \log(d)}{N}} \le 12\sqrt{\frac{2k^3 d^2 \log(d)}{N}},
\end{aligned}
$$

where we used the equality between states $\mathrm{Tr}_{A_1\cdots A_k}[(M_p \otimes \mathbb{I})\rho_{\mathbf{w}}^{A_1\cdots A_{k+1}}] = \mathrm{Tr}[M_p\rho_{\mathbf{w}}^{A_1\cdots A_k}]\rho_{\mathbf{w},p}^{A_N}$ and the inequality $\sum_p \| M_p X \|_1 \leq \sum_p \mathrm{Tr}\left[M_p|X|\right] = \| X \|_1$ as $\sum_p M_p = \mathbb{I}$. Therefore, by Markov's inequality we deduce:

$$\mathbb{P}_{l,\mathbf{w},p}\left[\| \rho_{l,\mathbf{w},p}^{A_N} - \rho_{l,\mathbf{w}}^{A_N} \|_1 > \varepsilon'\right] \leq \frac{\mathbb{E}_{l,\mathbf{w},p}\left[\| \rho_{l,\mathbf{w},p}^{A_N} - \rho_{l,\mathbf{w}}^{A_N} \|_1\right]}{\varepsilon'} \leq 12\sqrt{\frac{2k^3 d^2 \log(d)}{N\varepsilon'^2}}.$$

## Data availability

Data sharing not applicable to this article as no datasets were generated or analysed during the current study.

## Code availability

Code availability is not applicable to this article as no code was generated or analysed during the current study.

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

## Acknowledgements

We would like to thank Mario Berta and Philippe Faist for helpful discussions. We acknowledge support from the European Research Council (ERC Grant AlgoQIP, Agreement No. 851716) (O.F. and A.O.), (ERC Grant Agreement No. 948139) (A.O.), (ERC Grant Agreement No. 101117138) (R.K.), from the European Union's Horizon 2020 research and innovation program under Grant Agreement No 101017733 within the

QuantERA II Programme (O.F.) and from the PEPR integrated project EPiQ ANR-22-PETQ-0007 part of Plan France 2030 (O.F., D.M., and A.O.), as well as the QuantumReady and HPQC projects of the Austrian Research Promotion Agency (FFG) (R.K.).

## Author contributions

O.F., R.K., D.M., and A.O. contributed extensively to this work.

## Funding

## Competing interests

The authors declare no competing interests.
