## [Transparent Peer Review File · Nature Communications]

Learning Properties of Quantum States Without the IID Assumption

Corresponding Author: Dr Aadil Oufkir

Version 0:

Reviewer comments:

Reviewer #1

(Remarks to the Author)

Summary:

This submission deals with learning with the non-iid quantum states, in which the authors propose a generic approach to upgrade an algorithm designed for i.i.d. input states to be the one to handle non-i.i.d. input states in any learning problem (under some mild assumptions), albeit at the expense of an increase in copy complexity.

To that end, the authors propose to consider the permutation-invariant state $\rho = \rho^{A_1 A_2 \dots A_N}$, and call this state non-iid input state with copy complexity N . Because the state is no longer iid, the authors propose to define the learning target as the average of the reduced quantum state of ρ over all possible outcomes of a given learning algorithm, which is deemed as a quantum-to-classical channel.

Under this assumption/setup, the authors prove that any learning algorithm for iid input states can be generically upgraded to be a learning algorithm for non-iid input states in the above sense. The authors further analyze the special case of learning algorithms with non-adaptive incoherent measurements. The authors also investigate the application of the above generic learning algorithms to many fundamental tasks like shadow tomography, pure state verification, state tomography, and so on.

The central idea in proving these results is a new “randomized” local quantum de Finetti theorem, and the proof reads sound. I have attached a more technical pdf explaining these results.

Strengths: (1) It is conceptually interesting to study the learning in the non-iid case.

(2) The development of a variant of the local quantum de Finetti theorem could be of independent interest.

Weakness: (1) I feel the definition of non-iid setting needs better first-principle motivation and justification. In general, I feel the new definition of the target learning states and treating N as the corresponding copy complexity is at least confusing, and sometimes hard to make sense in the applications. For example, what is the first-principle operational meaning of learning a non-iid input state, especially when your definition of non-iid input states seems motivated by the technique rather than the first principles? I have more detailed comments in the attached PDF.

(2) It is not clear why you can assume your input state is permutation-invariant without loss of generality. (e.g., line 99-100). In your current definition, your learning target is the average of reduced quantum states over this permutation. Here you make an implicit assumption that your learning algorithm does not memorize the random permutation itself. One can also come up with a definition where the learning target is the reduced quantum state per permutation, where you calculate the success probability per each permutation, and average them to get the average success probability. These two definitions are generally not equivalent because your learning procedure could be non-linear. Your technique does not seem able to handle the other case. Please justify your setting.

Typos: see attached pdf.

Reviewer #2

(Remarks to the Author)

This paper provides a framework for learning properties of quantum states that are not independent and identically distributed (iid). There is a long history of rigorous results on the sample complexity of learning quantum states and processes, but one of the drawbacks to many of these results is that their assumptions are substantially broken in real-world quantum information systems. One of those common assumptions is that of iid states, and so this work is addressing a problem of practical relevance as well as of fundamental scientific interest. In this work, the authors show how to adapt methods for learning quantum states in the iid setting to the non-iid setting, with a reasonably small overhead. In my opinion, this is a significant result. Furthermore, the paper is well-written and fairly approachable (at least if we condition on how technical the subject is). I recommend publication in Nature Communications, after the authors address my minor comments below.

1. In the abstract, the term "copy complexity" could be clarified, as readers unfamiliar with the formal state learning literature might not know what this means. Similarly, is there a way to be more descriptive about what "non-adaptive incoherent measurements" are in the abstract?
2. In the final sentence of the abstract, "can" -> "may" or "will".
3. The high-level results in this work are, I think, fairly intuitive. If I understand how this works correctly, the intuition is the following (1) by effectively randomized the label of each subsystem (by randomizing whether each system is a test or training system), this makes all N subsystems identically distributed, and (2) in the algorithms presented here, a certain number of the subsystems are measured with the results more-or-less discarded and, as long as you measure enough of the subsystems, this makes the remaining subsystems very close to independent. This is because the measurements necessarily destroy dependencies/correlations, and which set of systems you're measuring is random. If this intuition is correct it'd be great if this was communicated in the paper. If it's wrong it'd be great if you more clearly explain what the intuition is.
4. I found the distinction between 'delta' for protocols with and without calibration data (i.e., Eq (1) and the following unnumbered equation) unnecessarily cumbersome. Why not just have a single Eq. (1) where ρ is conditioned by p and any other output of the learning algorithm B ? I can see the reason for allowing B to have different outputs (i.e., calibration data c ranging from trivial to the entire measurement record), but I can't see why you would ever define 'delta' to not condition on everything that B outputs, in which case there is no need for two definitions but just flexibility in what c corresponds to. If I am wrong and (1) is still useful when there is calibration data that should be explained. [I also found the paragraph justifying why ρ is condition on p in Eq (1) unnecessary -- this seemed obvious to me, but I'm just one data point, and maybe other people will find it useful].
5. In the first sentence of 2.2 it isn't stated that A is an algorithm for iid learning but it should be for the sentence to make sense.
6. In Theorem 2.1 it would be worth re-stating what d is. I had forgotten by this point and initially thought it was undefined.
7. The description of algorithm B [line 106-110] was very terse and has no lead in. It starts "**The** algorithm B ", but so far there is no single algorithm B -- Theorem 2.1 just says there exists an algorithm B that etc etc. Unless I have misunderstood, this paragraph is describing an explicit algorithm that, by its existence, proves Theorem 2.1 and which could be used. It would also be helpful here to have an intuitive description of how and why this algorithm works (see point 3 above).
8. In line 114 there is a missing 'be'.
9. Footnote 1 would probably be better as a parenthetical in the main text (I didn't notice it and I just guessed, fortunately correctly, at what a measurement channel meant).
10. In line 149, it would help the reader if you briefly described what 'non-adaptive and incoherent' means.
11. The paragraph beginning on line 158 again starts with "**The** algorithm B " and this makes the reader guess that you are about to present an explicit algorithm achieving Theorem 2.3.
12. In line 164 and 165 it states that B outputs the calibration data w . It is strange to me that nothing is said about what is done with the calibration data. As far as I can tell, the algorithm does nothing with the data, and its role is purely in conditioning the state that we're trying to estimate properties of. Perhaps this is always what is meant by the term 'calibration data' in this context, but it is not a term that I am familiar with and I think many readers' reaction to a term like 'calibration data' is to assume that an action is going to be based on it. It would probably be helpful to explain what calibration data means the first time the term is used.

13. In Section 2.3 it is unclear if it must be *assumed* that the state is pure. Does the state verification protocol presented here break (i.e., potentially give incorrect results) if the state might actually be mixed? I suspect that is the case, but it wasn't clear in the paper.

Finally, I would like to point out that I did not check all of the mathematics in the Methods section. Everything appears to be correct, and all the results are plausible, but I do not have the time to check all the derivations.

Reviewer #3

(Remarks to the Author)
Comments in the pdf attachment

Reviewer #4

(Remarks to the Author)
I co-reviewed this manuscript with one of the reviewers who provided the listed reports. This is part of the Nature Communications initiative to facilitate training in peer review and to provide appropriate recognition for Early Career Researchers who co-review manuscripts.

Version 1:

Reviewer comments:

Reviewer #1

(Remarks to the Author)
I would like to thank the authors for the updated draft and a detailed list of responses to my previous review. I have the following two major comments:

- (1) I believe that the theorem statement of Theorem 4.7 is incorrect. I compiled a counterexample in the attached PDF. As a result, my previous comment 2 still holds.
- (2) I agree with the need to go beyond the setting of IID, which you spend a lot of effort in addressing. My real question is why your particular setting is an appropriate definition beyond IID. To address this, I suppose you can demonstrate that your definition suffices for important applications. However, in order to show that, you need to first fix the technique gap that I pointed up above, if your applications implicitly assume that the input state is permutationally invariant without loss of generality.

Reviewer #2

(Remarks to the Author)
In my previous review we (I and my early-career co-reviewer) recommended publication in Nature Communications if the authors addressed some minor concerns. The authors' response letter and changes to their manuscript have indeed addressed all of these concerns. In our opinion, the authors have also adequately responded to all of the criticisms of the other referees. We therefore recommend publication in Nature Communications.

Reviewer #3

(Remarks to the Author)
The authors have convincingly responded to all the comments in our review. In particular, they have provided more discussions in Section 4.3, expanded further on the verification of computation, and motivated the "non-iid" setting better. The authors have also better explained why they prefer to keep the calibration data separate from the prediction. Removing it would not significantly simplify the presentation, so we are fine with keeping it around.

Reviewer #4

(Remarks to the Author)
I co-reviewed this manuscript with one of the reviewers who provided the listed reports. This is part of the Nature Communications initiative to facilitate training in peer review and to provide appropriate recognition for Early Career Researchers who co-review manuscripts.

Reviewer #5

(Remarks to the Author)
"I co-reviewed this manuscript with one of the reviewers who provided the listed reports. This is part of the Nature Communications initiative to facilitate training in peer review and to provide appropriate recognition for Early Career Researchers who co-review manuscripts."

Version 2:

Reviewer comments:

Reviewer #1

(Remarks to the Author)

Thanks for the revision. I am happy that my previous counter-example was helpful and agree with the current technical statement.

However, without the original theorem statement, the current result looks conceptually weaker. Although this submission made progress toward the non-iid case, the current proposal is more like a technical outcome rather than a convincing first-principle alternative to the iid case.

I would suggest the authors explicitly discuss this limitation. Overall, this submission is still an interesting attempt.

Learning Properties of Quantum States Without the I.I.D. Assumption

Omar Fawzi, Richard Kueng, Damian Markham, and Aadil Oufkir

We thank the reviewers for their thorough feedback and comments. In the following we provide a point-by-point response to the reviewers' comments. The changes in the manuscript are highlighted by red color. Note that the numbering of the lines might be different between the previous submission and the actual one.

I. REVIEWER 1

A. Typos

1. In the equation below line 92, should \mathbb{P}_ρ be $\mathbb{P}_{c,p}$?
2. Line 114: “which should an informationally-completely” should be “which should be an informationally-complete”.
3. The equation (1F) below line 127 does not make sense. I guess you intend to write

$$\sup_{\Lambda_2, \dots, \Lambda_k} \left\| \text{id} \otimes \Lambda_2 \otimes \dots \otimes \Lambda_k \left(\rho^{A_1 \dots A_k} - \int d\nu(\sigma) \sigma^{\otimes k} \right) \right\|_1 \leq \sqrt{\frac{2k^2 \log(d)}{N - k}}.$$

4. Line 446: “for the states on conditioned” should be “for the states conditioned”.
5. In the 4th line of equation (6), $\rho_{l,r,w}^{A_1 A_2 \dots A_k}$ should be $\rho_{l,r,w}^{A_1 A_2 \dots A_k A_{k+1}}$

We thank the reviewer for finding these typos. We implemented all of them, the changes are in red color.

B. Comment 1

I feel the definition of non-iid setting needs better first-principle motivation and justification. In general, I feel the new definition of the target learning states and treating N as the corresponding copy complexity is at least confusing, and sometimes hard to make sense in the applications. For example, what is the first-principle operational meaning of learning a non-iid input state, especially when your definition of non-iid input states seems motivated by the technique rather than the first principles? I have more detailed comments in the attached PDF.

Our definition is operationally motivated by the task described in Figure 1. We imagine a black box from which we can request “copies”. On the first query, we receive system A_1 and on the k -th query, we receive A_k . Learning means making a statement about some of the outputs of the black box (e.g., the state is close to $|0\rangle$). With the i.i.d. assumption, the black box always outputs the same state. Removing the i.i.d. assumption, we have to specify the system about which we make the statement (this is the system that would be used for a later application for example). The most natural choice is to take a system at random among the ones that were requested. In fact, for any system index that is fixed in advance, no non-trivial statement can be made in general.

We updated Section 2.1 of the manuscript with this discussion, which we hope clearly motivates the definition.

C. Comment 2

It is not clear why you can assume your input state is permutation-invariant without loss of generality. (e.g., line 99-100). In your current definition, your learning target is the average of reduced quantum states over this permutation. Here you make an implicit assumption that your learning algorithm does not memorize the random permutation itself. One can also come up with a definition where the learning target is the reduced quantum state per permutation, where you calculate the success probability per each permutation, and average them to get the average success probability. These two definitions are generally not equivalent because your learning procedure could be non-linear. Your technique does not seem able to handle the other case. Please justify your setting.

Thank you for this comment. We indeed do not need permutation invariance of the state, the statement can be proved in expectation over the choice of a random permutation, the new version of Theorem 4.7 is now written in this way. To keep the notation simple (i.e., not explicitly mentioning the permutation), we still make the assumption of permutation invariance in most of the statements of the paper and we added a remark about the stronger statement after Theorem 2.1.

D. Comment 3

The concept of “learning” a non-i.i.d. state is awkward. For example, in the context of fidelity estimation, the goal is to estimate the fidelity between an unknown state and a known target state $|\psi\rangle$. This makes perfect sense in the i.i.d. scenario. However, if the input state is globally entangled across N subsystems, and suppose in one execution of the algorithm you get (c,p) and in another execution on the same input state you get (c',p') . It could be the case that $\langle\psi|\rho_{c,p}^{A_N}|\psi\rangle$ and $\langle\psi|\rho_{c',p'}^{A_N}|\psi\rangle$ are completely different. Then what’s the point of learning the fidelity of $\rho_{c,p}^{A_N}$ when you can’t guarantee that you will get the same reduced state next time?

In the non i.i.d. setting, repeating the learning procedure might indeed lead to very different outcomes. However, this is a feature and not a problem: the guarantee that is achieved by the learning task is that for the test system (which we denote N after applying the random permutation and is the system that we would want to use for another task) has the property being learned correspond to p . If we later run the whole process with the same devices and a new training and testing set, the outcome could be different as expected.

However, depending on the application, we only repeat the learning procedure when necessary. This is the case, for instance, of verification when we accept the post measurement state only when the fidelity is close to 1. As a bonus, in this case we have many remaining copies that are close to the ideal state and can be reused for other applications (see Figure 2, the copies on the right of the random bar). Hence, we only need to repeat the experiment until obtaining a state with high fidelity. On the other hand, while it is clear that the post measurement states are all equal to the marginal state *in expectation*, we also expect that, in situations when the state is close to i.i.d. or having weak correlations, the post measurement states are close to the marginal. Finally, we conjecture that for randomized measurement (optimal for many learning problems) lead to post measurement states close to the marginal with high probability.

E. Comment 4

I think it doesn’t make sense to call N “copy complexity”. In the i.i.d. scenario, we call it copy complexity because we want to understand how many copies of an unknown state

are needed to learn some property about the state. As we ask for more and more copies, we get a better and better estimate of the unknown state. In the non-i.i.d. scenario, for a general input state $\rho^{A_1 \dots A_N}$, the number N is basically “fixed”: You don’t have the choice of asking for more subsystems.

The word “copy complexity” is standard in the learning literature (which always makes the i.i.d. assumption) and we use it by analogy. In addition, we hope that the motivation provided in the answer to Comment 1 (IB) justifies it as the number of requested “copies” from the black box.

F. Comment 5

In section 2.3, you say that “Obtaining a statement beyond the i.i.d. assumption is particularly important for cryptographic applications”. Could you provide more arguments for that?

In cryptographic applications it is important to address all assumptions made in proofs carefully. Section 2.3 deals with the verification of pure states, which in turn can be used to verify applications for which they are the resource state, against a malicious adversary. In general, a malicious adversary could use the assumption of IID to form an attack which would invalidate security claims. For example, if they collaborate with the source, if they knew which copies were being tested, they could act honestly on those systems, and arrange for bad states to be used when they are used for an application - this would then pass the test, and be potentially fully corrupt the application.

For example, in measurement based quantum computing the difficult part is to generate a large, entangled state, called a graph state, then (universal) computation can be carried out by applying a sequence of ‘easy’ local measurements. In this sense the graph state is a resources state. If a ‘weak’ party, the user, with only local measurements wants to run a computation, they simply ask a powerful server to send them the graph state (even qubit by qubit if they don’t have a quantum memory). However, they may not trust the server, who may either not really have a powerful quantum computer, or may want to corrupt their computation by sending a bad state (if the user performed the same measurements on a non entangled state, the output would be garbage). It is possible to verify a graph state, also using single qubit measurements, hence the state can be verified in the same framework. Typically this is done by asking the server for many copies of the resource graph state and testing some of them, and then using the remaining to run the computation (see ref [22]). However, if the server knew that the user assumed IID, they could send any state they like, not IID, which may invalidate the verification. In particular, the usefulness of IID is that it assumes the same state that is tested is that which is used. However, if the server knew which copies were to be tested, and which copy the computation was run on, they could pass the verification but totally corrupt the computation.

We have included a deeper discussion on this topic in Section 2.3 of new draft that we hope clarifies this point.

II. REVIEWER 2

A. Comment 1

In the abstract, the term “copy complexity” could be clarified, as readers unfamiliar with the formal state learning literature might not know what this means. Similarly, is there a

way to be more descriptive about what “non-adaptive incoherent measurements” are in the abstract?

We reformulated the abstract. In particular we changed the term “copy complexity” to “training data size” and “non-adaptive incoherent measurements” to “non-adaptive single-copy measurements”.

B. Comment 2

In the final sentence of the abstract, “can” → “may” or “will”.

This sentence is removed in the new reformulation.

C. Comment 3

The high-level results in this work are, I think, fairly intuitive. If I understand how this works correctly, the intuition is the following (1) by effectively randomized the label of each subsystem (by randomizing whether each system is a test or training system), this makes all N subsystems identically distributed, and (2) in the algorithms presented here, a certain number of the subsystems are measured with the results more-or-less discarded and, as long as you measure enough of the subsystems, this makes the remaining subsystems very close to independent. This is because the measurements necessarily destroy dependencies/correlations, and which set of systems you’re measuring is random. If this intuition is correct it’d be great if this was communicated in the paper. If it’s wrong it’d be great if you more clearly explain what the intuition is.

Your intuition is correct. We tried to improve the communication of this intuition in the paper around lines 107 to 135 and in more detail around 495.

D. Comment 4

I found the distinction between ‘delta’ for protocols with and without calibration data (i.e., Eq (1) and the following unnumbered equation) unnecessarily cumbersome. Why not just have a single Eq. (1) where ρ is conditioned by p and any other output of the learning algorithm B ? I can see the reason for allowing B to have different outputs (i.e., calibration data c ranging from trivial to the entire measurement record), but I can’t see why you would ever define ‘delta’ to not condition on everything that B outputs, in which case there is no need for two definitions but just flexibility in what c corresponds to. If I am wrong and (1) is still useful when there is calibration data that should be explained. [I also found the paragraph justifying why ρ is condition on p in Eq (1) unnecessary – this seemed obvious to me, but I’m just one data point, and maybe other people will find it useful].

Based on the comments from Reviewer 2 and Reviewer 3 regarding “calibration data”, we have decided to concentrate on the setting in which a learning algorithm can output any calibration data in the main text (see the paragraph about error probability around line 428). We have also deferred the distinction between the post-measurement states conditioned only on the prediction and those conditioned on both the prediction and the calibration to Appendix B.

Allowing the learning algorithm to output any calibration data simplifies the analysis and permits non i.i.d. learning for a broad class of problems. See Appendix B where we were only able to prove non i.i.d. learning without calibration for a restricted class of learning problems.

Moreover the error terms in the error probability without calibration (Theorem B.3) are worse than the ones with calibration (Theorem 4.8).

We think that justifying why the target state should be the post measurement state is important since choosing the marginal state as a target state might be tempting at first.

E. Comment 5

In the first sentence of 2.2 it isn't stated that A is an algorithm for iid learning but it should be for the sentence to make sense.

We added that \mathcal{A} is designed for i.i.d. input states.

F. Comment 6

In Theorem 2.1 it would be worth re-stating what d is. I had forgotten by this point and initially thought it was undefined.

We edited Theorem 2.1 by adding “Let $\varepsilon > 0$, $1 \leq k < N/2$ and d be the dimension of the Hilbert spaces A_1, \dots, A_N ”.

G. Comment 7

*The description of algorithm B [line 106-110] was very terse and has no lead in. It starts “*The* algorithm B”, but so far there is no single algorithm B – Theorem 2.1 just says there exists an algorithm B that etc etc. Unless I have misunderstood, this paragraph is describing an explicit algorithm that, by its existence, proves Theorem 2.1 and which could be used. It would also be helpful here to have an intuitive description of how and why this algorithm works (see point 3 above).*

The algorithm B is exactly Algorithm 2 that achieves Theorem 2.1. We clarified this in the main text (line 117).

H. Comment 8

In line 114 there is a missing 'be'.

Thank you, we added it.

I. Comment 9

Footnote 1 would probably be better as a parenthetical in the main text (I didn't notice it and I just guessed, fortunately correctly, at what a measurement channel meant).

We moved the definition of measurement channels to the main text.

J. Comment 10

In line 149, it would help the reader if you briefly described what 'non-adaptive and incoherent' means.

We added a brief definition of non-adaptive incoherent algorithms: performing single copy measurements using a set of measurement devices chosen before starting the learning procedure.

K. Comment 11

The paragraph beginning on line 158 again starts with "The algorithm B" and this makes the reader guess that you are about to present an explicit algorithm achieving Theorem 2.3.

The algorithm B is exactly Algorithm 1 that achieves Theorem 2.3. We clarified this in the main text (line 178).

L. Comment 12

In line 164 and 165 it states that B outputs the calibration data w . It is strange to me that nothing is said about what is done with the calibration data. As far as I can tell, the algorithm does nothing with the data, and its role is purely in conditioning the state that we're trying to estimate properties of. Perhaps this is always what is meant by the term 'calibration data' in this context, but it is not a term that I am familiar with and I think many readers' reaction to a term like 'calibration data' is to assume that an action is going to be based on it. It would probably be helpful to explain what calibration data means the first time the term is used.

We agree with the Reviewer's comment that the role of the calibration data is to specify the target state which is the post measuring state conditioned on the prediction and the calibration data. At this stage, the algorithm does nothing with this calibration data since its role is predicting the property p . However, the calibration data and the post-measurement state could be used afterwards for other applications. We added an explanation of the calibration data around line 80.

M. Comment 13

*In Section 2.3 it is unclear if it must be *assumed* that the state is pure. Does the state verification protocol presented here break (i.e., potentially give incorrect results) if the state might actually be mixed? I suspect that is the case, but it wasn't clear in the paper.*

Since we have two states in hand, one of them is the unknown state ρ to be verified and the other one is the known target/ideal state $\sigma = |\Psi\rangle\langle\Psi|$. The question admits two possibilities: Do we need to assume ρ is pure? or do we need to assume σ is pure? We answer both:

- Assuming ρ is pure: We do not assume ρ to be pure. Our verification protocol works for an arbitrary input state $\rho^{A_1 \dots A_N}$ that can be mixed and entangled (see Proposition 4.20). We followed (General framework for verifying pure quantum states in the adversarial scenario, Huangjun Zhu and Masahito Hayashi) for the name "verification of pure state".

- Assuming $\sigma = |\Psi\rangle\langle\Psi|$ is pure: Verification of pure states via fidelity estimation is possible because $F(\rho, |\Psi\rangle\langle\Psi|) = \langle\Psi|\rho|\Psi\rangle$. The overlap $\langle\Psi|\rho|\Psi\rangle$ can be easily estimated (in the i.i.d. setting) since it is an expectation $\text{Tr}[\rho O]$ of a known observable $O = |\Psi\rangle\langle\Psi|$ under the unknown state ρ . In the non-i.i.d. setting we use our framework applied on the classical shadows (note that the observable $O = |\Psi\rangle\langle\Psi|$ has shadow norm at most 1). The same verification protocol breaks for non-pure states as the identity $F(\rho, \sigma) = \text{Tr}[\rho\sigma]$ is not true in general and hence shadow tomography is not applicable. Note that the known algorithms for estimating the fidelity (e.g., Improved Quantum Algorithms for Fidelity Estimation, András Gilyén, Alexander Poremba) require a number of copies polynomial in the rank of σ .

III. REVIEWER 3

A. Comment 1.1

- Authors give several motivations in first paragraph of the paper in the following statement: verified quantum computation [37] or tasks using entangled states in networks [50], such as authentication of quantum communication [6], anonymous communication [9] or distributed quantum sensing. Can they provide clear applications that work for this motivation? We strongly recommend providing more precise applications for the non-iid framework.

We have expanded the discussions and applications of our results for verification of pure states, and how this in turn can be applied to the different problems mentioned in the introduction, and why the non-iid setting is so important.

In section 2.3 we have make a more detailed explanation of how verification of pure states can be applied, and in section 4.4.2 we have outlined two specific examples where our results provide applications for networks of sensors and communication.

B. Comment 1.2

-Verified quantum computation is particularly discussed in Sec 2.3 - but there is a difference between verifying quantum computation and verifying pure quantum state. In what sense are the authors thinking of an application here? Is there a Kitaev's history state being used here?

The referee is of course correct that verification of states is not the same as computation and we regret this was not clear in the previous draft. The simplest application of state verification, for the verification of quantum computation is in reference [22], where the verification of a graph state is used to directly verify universal quantum computation (see also Reviewer 1 comment 3). In particular the verification of the computation itself follows directly as a consequence of the state verification. Indeed, in general once a resource state is verified, as long as one trusts the subsequent operations performed on it, the induced application is also verified (see e.g. [50]). It does not make use of Kitaev's history state.

We have expanded the discussion in section 2.3 which hopefully clarifies this issue.

C. Comment 1.3

- The recent work of Huang, Preskill, Soleimanifar gives conditionally efficient verification algorithm when one only has access to amplitudes of the state psi (that is to be

verified). Do the techniques in this submission apply here too? We understand that Huang, Preskill, Soleimanifar's work comes after the current submission.

Yes our result can be applied because their algorithm is non-adaptive.

D. Comment 2.1

Technique: the main technique is that of quantum de-finetti theorem, in particular the formalism due to Brandao and Harrow (BH). - Generally, BH does not require permutation invariance in the state (since its mostly based on classical measurements followed by chain rule of CMI, which is general - see Section 2.3 in the related paper <https://arxiv.org/pdf/1310.0017.pdf>). Why is permutation invariance assumed in the current work?

Thank you for this comment. We indeed do not need permutation invariance of the state, the statement can be proved in expectation over the choice of random permutation, the new version of Theorem 4.7 is now written in this way. To keep the notation simple (i.e., not explicitly mentioning the permutation), we still make the assumption of permutation invariance in most of the statements of the paper and we added a remark about the stronger statement after Theorem 2.1.

E. Comment 2.2

- The difference between Thm 2.2 and Equation 1F is that the former does not need mixture of product states. It's not clear why that is useful - in other words, why didn't Equation 1F already work for the authors. If a learning algorithm works well for all product states, shouldn't it work for mixtures of product states due to convexity of some relevant quantity?

The convexity argument might be used for another definition of non i.i.d. learning where we want to estimate, for example, properties of the marginal state. This can be done if the property we want to estimate, the algorithm's statistic/estimator and the approximation distance all behave nicely (e.g., convex) as well as a guarantee in terms of expectation and not with high probability.

In this work, we focus on predicting properties of the post-measurement state (see around line 88 for the motivation and for a counter example) and do not want to limit too much the problems we consider (the property we want to learn and the approximation distance) as well the algorithms (statistic/estimator). For these reasons, Theorem 2.2 is more suitable to us.

F. Comment 2.3

- Is it possible to extend Thm 2.2 to measurements that are adaptive but non-entangling?

It is not clear to us how to generalise Theorem 2.3 to adaptive algorithms at least with the same approach and performance. The reason is that adaptive algorithms can use up to $\mathcal{O}(d^{k-1})$ different POVMs where k is the copy complexity. Naively using the same method leads to an algorithm that measures all these POVMs and thus uses a number of copies $\mathcal{O}(k^2 d^{2k-2} / \epsilon^2)$ in the non-i.i.d. setting.

G. Comment 2.4

- In Theorem 4.8, why not set $k < \sqrt{N}$? The bound seems trivial if the RHS is > 1 .

Indeed we could set $k < \sqrt{N}$ or even $k < \sqrt{\frac{N}{4 \log(d)}}$.

H. Comment 2.5

- In Theorem 4.8, the role of r is quite unclear. It seems to specify the choice of the POVMs being performed. Why does r have to come from a permutation invariant distribution? It goes back to the previous question about why permutation invariance is needed in the first place

See the response to Comment 2.1 (IIID). r indeed specifies the choice of POVM that is performed. Without the permutation invariance assumption for q , it is not clear how to exchange the different registers X_j in the proof of the theorem.

I. Comment 3.1

Section 4.3 is a little difficult to read and needs to be significantly clarified. Here are some questions that come to one's mind after a few reads: Can the algorithm 1 be described in words? Why is so much randomization necessary in this algorithm? what is k_A vs k (will be clarified once the algorithm is described in words)? Some of the complexity of Algorithm 1 comes from Thm 4.8 in Sec 4.2. There are too many random variables to worry about. If the authors can simplify some of it, would be great.

To simplify Section 4.3, we implemented the following changes:

- We added a description of Algorithm 1 in words before stating formally the algorithm (see around lines 493-498): In words, given a non-adaptive incoherent algorithm \mathcal{A} that uses a set of measurement devices $\{\mathcal{M}_t\}_t$, Algorithm 1 measures a large number of the state's subsystems using measurement devices uniformly chosen from $\{\mathcal{M}_t\}_t$. This ensures that the (small) portion of measured subsystems intended for the learning algorithm approximately behave like i.i.d. copies. Then, in order to predict the property, Algorithm 1 applies the data processing of Algorithm \mathcal{A}) to the outcomes of these subsystems.
- The difference between k and k_A : k_A is the copy complexity in the i.i.d. setting and $k = k_A \log(k_A/\delta_A)$ is the number of copies in the non-i.i.d. setting we use for the prediction. We need slightly more copies to span all the k_A (possibly) different POVMs that a non-adaptive algorithm performs in the data acquisition part. The reason is that we sample at each time a POVM uniformly at random from the set of POVMs used by the non-adaptive algorithm. **The results are now stated with only k_A (we set $k = k_A \log(k_A/\delta_A)$), and we only differentiate between k and k_A in the proof.**
- We set $k = k_A \log(k_A/\delta_A)$, $\varepsilon' = \varepsilon$ and group the third and fourth terms in Theorem 4.8. Now the non-i.i.d. error probability is roughly bounded by the i.i.d. error probability and an additional error term accounting for the possibility of the input state being non i.i.d.

J. Comment 3.2

Calibration information is an unclear aspect. It seems to be put in place to have a general theorem - but note that 'less is more'. It might be worth moving this notion to supplementary info, if it's not a crucial concept. As an example of the confusion, consider example 3 of Page 12. The string x could easily be the prediction and one could defer the job of deriving $|x|$ from x to the learner. Then why is the distinction between c and p so much emphasized?

Based on the comments from Reviewer 2 and Reviewer 3 regarding “calibration data”, we have decided to concentrate on the setting in which a learning algorithm can output any calibration data in the main text (see the paragraph about error probability around line 428). We have also deferred the distinction between the post-measurement states conditioned only on the prediction and those conditioned on both the prediction and the calibration to Appendix B.

K. Comment 3.3

- Example 4 could fit better in the supplementary part. For those who know the BH proof, thm 4.8's proof is clear enough. For those who don't know the BH proof, Example 4 is complex enough to not help.

We moved Example 4 to the supplementary part (Appendix A).

Learning Properties of Quantum States Without the I.I.D. Assumption

Omar Fawzi, Richard Kueng, Damian Markham, and Aadil Oufkir

We thank the reviewers for their important feedback and comments. In the following we provide a response to the Reviewer 1 comments. The changes in the manuscript are highlighted by red color. Note that the numbering of the lines might be different between the previous submissions and the actual one.

I. REVIEWER 1

A. Comment 1

I believe that the theorem statement of Theorem 4.7 is incorrect. I compiled a counter-example in the attached PDF. As a result, my previous comment 2 still holds.

We thank the Reviewer for pointing out this mistake. The inequality in Theorem 4.7 should be

$$\begin{aligned} & \mathbb{E}_{\mathbf{j}, l, \mathbf{r} \sim q^N} \left[\sum_{\mathbf{w}} \langle \mathbf{w} | \left(\bigotimes_{i=l+1}^{k+N/2} \Lambda_{r_i} \right) (\rho^{A_{j_1} \dots A_{j_N}}) | \mathbf{w} \rangle \left\| \text{id} \otimes \left(\bigotimes_{i=2}^{k+1} \Lambda_{r_i} \right) \left(\rho_{l, \mathbf{r}, \mathbf{w}}^{A_{j_1} \dots A_{j_{k+1}}} - \bigotimes_{i=1}^{k+1} \rho_{l, \mathbf{r}, \mathbf{w}}^{A_{j_i}} \right) \right\|_1 \right] \\ & \leq \sqrt{\frac{4k^2 \log(d)}{N}}, \end{aligned} \quad (\text{I.1})$$

and if the state ρ is permutation invariant (or at least has the same post-measurement marginals) the inequality (I.1) becomes

$$\begin{aligned} & \mathbb{E}_{\mathbf{j}, l, \mathbf{r} \sim q^N} \left[\sum_{\mathbf{w}} \langle \mathbf{w} | \left(\bigotimes_{i=l+1}^{k+N/2} \Lambda_{r_i} \right) (\rho^{A_{j_1} \dots A_{j_N}}) | \mathbf{w} \rangle \left\| \text{id} \otimes \left(\bigotimes_{i=2}^{k+1} \Lambda_{r_i} \right) \left(\rho_{l, \mathbf{r}, \mathbf{w}}^{A_{j_1} \dots A_{j_{k+1}}} - \left(\rho_{l, \mathbf{r}, \mathbf{w}}^{A_{j_N}} \right)^{\otimes k+1} \right) \right\|_1 \right] \\ & \leq \sqrt{\frac{4k^2 \log(d)}{N}}. \end{aligned}$$

This error is now corrected.

B. Comment 2 (from previous revision)

It is not clear why you can assume your input state is permutation-invariant without loss of generality. (e.g., line 99-100). In your current definition, your learning target is the average of reduced quantum states over this permutation. Here you make an implicit assumption that your learning algorithm does not memorize the random permutation itself. One can also come up with a definition where the learning target is the reduced quantum state per permutation, where you calculate the success probability per each permutation, and average them to get the average success probability. These two definitions are generally not equivalent because your learning procedure could be non-linear. Your technique does not seem able to handle the other case. Please justify your setting.

We also corrected the comment that we had added to the previous revision. We now clearly state that we start the process by applying a random permutation that the learner does not have

access to (see around lines 70 - 75). We show that in fact for the alternative definition where the permutation is available to the learner, even very simple learning tasks are not possible (see around lines 102 - 115 and Appendix A). We would actually like to thank the referee for his detailed example concerning the previous version of Theorem 4.7 as we use a similar example in Appendix A. We also point out that for some verification tasks (e.g., the prediction is of the form Accept/Reject and we consider the expected fidelity to a pure state), the two definitions are equivalent.

C. Comment 2

I agree with the need to go beyond the setting of IID, which you spend a lot of effort in addressing. My real question is why your particular setting is an appropriate definition beyond IID. To address this, I suppose you can demonstrate that your definition suffices for important applications. However, in order to show that, you need to first fix the technique gap that I pointed up above, if your applications implicitly assume that the input state is permutationally invariant without loss of generality.

We hope the previous discussion motivates our choice of definition. Our objective was to encompass general learning tasks (i.e., not only specific verification scenarios) and in such a setting, we argued it is essential to hide the permutation from the learner.

1 Summary of Results

This paper shows that given any learning problem (under reasonable assumptions), an algorithm designed for i.i.d. input states can be adapted to handle non-i.i.d. input states, albeit at the expense of an increase in copy complexity.

1.1 Framework of learning with non-i.i.d. input states

The **input state** is a *permutation-invariant* state $\rho^{A_1 A_2 \dots A_N}$ over N subsystems $A_1 \cong A_2 \cong \dots \cong A_N$, each of dimension d . Here N is called the **copy complexity**. We say that ρ is an **i.i.d. state** if it has the form $\rho = \sigma^{\otimes N}$.

A **learning algorithm** \mathcal{B} is a measurement channel (i.e., a quantum-to-classical channel) applied to the subsystems $A_1 \dots A_{N-1}$. The learning algorithm outputs a “prediction” p and possibly some “side information” c . We denote $\rho_{c,p}^{A_N}$ to be the reduced state of subsystem A_N conditioned on observing the outcome (c, p) .

The goal is to design learning algorithms such that, with high probability, the output p is an approximately correct prediction of a property of the quantum state $\rho_{c,p}^{A_N}$. We evaluate the performance of the learning algorithm by its error probability:

$$\delta_{\mathcal{B}}(N, \rho^{A_1 \dots A_N}, \varepsilon) := \Pr_{(c,p) \sim \mathcal{B}(\rho^{A_1 \dots A_{N-1}})} [p \text{ is not a correct prediction for } \rho_{c,p}^{A_N} \text{ with precision } \varepsilon].$$

For this paper, one needs an extra **robustness assumption** of the property of quantum states to be learned: If p is a correct prediction of the property for state ρ with precision ε , and $\|\rho - \xi\|_1 \leq \varepsilon'$, then p is also a correct prediction of the property for state ξ with precision $\varepsilon + \varepsilon'$.

1.2 Main result 1: transforming a general learning algorithm

Given a learning algorithm \mathcal{A} designed for i.i.d. input states, one constructs a learning algorithm \mathcal{B} that works for arbitrary permutation-invariant input states $\rho^{A_1 \dots A_N}$:

1. Choose $1 \leq k < \frac{N}{2}$, which is a hyper-parameter of the algorithm.
2. Run \mathcal{A} on the first k subsystems $A_1 A_2 \dots A_k$ and obtain outcome p .
3. Pick $l \sim \text{Unif}\{1, 2, \dots, \frac{N}{2}\}$ and perform a fixed measurement $\mathcal{M}_{\text{dist}}$ on each of the next l subsystems $A_{k+1} A_{k+2} \dots A_{k+l}$, obtaining an output string w .
4. Return prediction p along with side information (l, w) .

The paper proves the following bound on the performance of the learning algorithm:

$$\sup_{\rho} \delta_{\mathcal{B}}(N, \rho^{A_1 \dots A_N}, 2\varepsilon) \leq \sup_{\sigma} \delta_{\mathcal{A}}(k, \sigma^{\otimes k}, \varepsilon) + \mathcal{O}\left(\sqrt{\frac{k^3 d^2 \log d}{N \varepsilon^2}}\right).$$

Thus, *any i.i.d. learning algorithm can be transformed into a non-i.i.d. one with an overhead in copy complexity that is polynomial in the dimension d .*

1.3 Main result 2

Let \mathcal{A} be a learning algorithm designed for i.i.d. input states and performing *non-adaptive incoherent* measurements, which means that, on k -copy input states, \mathcal{A} applies a measurement of the form $\mathcal{M}_1^A \otimes \mathcal{M}_2^A \otimes \cdots \otimes \mathcal{M}_k^A$ and then outputs a prediction based on the measurement outcomes. One constructs a learning algorithm \mathcal{B} that works for arbitrary permutation-invariant input states $\rho^{A_1 \dots A_N}$:

1. Choose $1 \leq k < m < \frac{N}{2}$, which are hyper-parameters of the algorithm.
2. Pick $l \sim \text{Unif} \{1, 2, \dots, \frac{N}{2}\}$ and $r = (r_1, \dots, r_N) \stackrel{\text{i.i.d.}}{\sim} \text{Unif} \{1, \dots, k\}$.
3. According to coupon collector's problem, (r_1, \dots, r_m) contains all elements in $\{1, \dots, k\}$ with high probability. Run \mathcal{A} on those k subsystems among the first m subsystems $A_1 A_2 \dots A_m$, obtaining outcome p .
4. Perform measurement $\mathcal{M}_{r_{m+1}}^A \otimes \mathcal{M}_{r_{m+2}}^A \otimes \cdots \otimes \mathcal{M}_{r_{m+l}}^A$ on subsystems $A_{m+1} A_{m+2} \dots A_{m+l}$, obtaining an output string w .
5. Return prediction p as well as side information (l, r, w) .

The paper proves the following bound on the performance of the learning algorithm: with a proper choice of m ,

$$\sup_{\rho} \delta_{\mathcal{B}}(N, \rho^{A_1 \dots A_N}, 2\varepsilon) \leq \sup_{\sigma} \delta_{\mathcal{A}}(k, \sigma^{\otimes k}, \varepsilon) + \mathcal{O}\left(\sqrt{\frac{k^2 (\log N)^2 \log d}{N \varepsilon^2}}\right).$$

The copy complexity overhead has a much better dependence in d than that in Main Result 1.

The central technique used in proving the result is a new “randomized local” quantum de Finetti theorem.

1.4 Applications

The authors apply the two results to many learning tasks: shadow tomography, verification of pure states, fidelity estimation, state tomography, and testing mixedness of states.

2 Typos

1. In the equation below line 92, should \mathbb{P}_{ρ} be $\mathbb{P}_{c,p}$?
2. Line 114: “which should an informationally-completely” should be “which should be an informationally-complete”.
3. The equation (1F) below line 127 does not make sense. I guess you intend to write

$$\sup_{\Lambda_2, \dots, \Lambda_k} \left\| \text{id} \otimes \Lambda_2 \otimes \cdots \otimes \Lambda_k \left(\rho^{A_1 \dots A_k} - \int d\nu(\sigma) \sigma^{\otimes k} \right) \right\|_1 \leq \sqrt{\frac{2k^2 \log(d)}{N - k}}.$$

4. Line 446: “for the states on conditioned” should be “for the states conditioned”.
5. In the 4th line of equation (6), $\rho_{l,r,w}^{A_1 A_2 \dots A_k}$ should be $\rho_{l,r,w}^{A_1 A_2 \dots A_{k+1}}$.

3 Comments

1. The concept of “learning” a non-i.i.d. state is awkward. For example, in the context of fidelity estimation, the goal is to estimate the fidelity between an unknown state and a known target state $|\psi\rangle$. This makes perfect sense in the i.i.d. scenario. However, if the input state is globally entangled across N subsystems, and suppose in one execution of the algorithm you get (c, p) and in another execution on the same input state you get (c', p') . It could be the case that $\langle\psi|\rho_{c,p}^{A_N}|\psi\rangle$ and $\langle\psi|\rho_{c',p'}^{A_N}|\psi\rangle$ are completely different. Then what’s the point of learning the fidelity of $\rho_{c,p}^{A_N}$ when you can’t guarantee that you will get the same reduced state next time?
2. I think it doesn’t make sense to call N “copy complexity”. In the i.i.d. scenario, we call it copy complexity because we want to understand how many copies of an unknown state are needed to learn some property about the state. As we ask for more and more copies, we get a better and better estimate of the unknown state. In the non-i.i.d. scenario, for a general input state $\rho^{A_1\dots A_N}$, the number N is basically “fixed”: You don’t have the choice of asking for more subsystems.
3. In section 2.3, you say that “Obtaining a statement beyond the i.i.d. assumption is particularly important for cryptographic applications”. Could you provide more arguments for that?

In the usual framework of learning theory (classical or quantum), the inputs are given as independent and identically distributed samples. The learning algorithm uses some of these samples to make a prediction and then tests the prediction against rest of the samples. This work considers the realistic scenario where the inputs can be arbitrarily entangled across the copies and tries to understand how many more samples would be needed to achieve the same learning goal. Since the entangled input can always be made symmetric by a random permutation, the authors' main observation is that the powerful quantum de-finetti theorem can be used.

The main result of the authors is that as long as non-adaptive measurement strategy is used, the learning can be achieved with mildly more number of copies. Since non-adaptive strategies are standard learning methods (and arguably easier to implement than adaptive entangled strategies), this is a powerful result. An important application of the result is verification of quantum states (i.e. - are the samples from a pure state ψ ?) by performing measurements that do not depend on ψ , except in the classical postprocessing.

However, the paper does not yet meet the bar of Nature Communications, since it is not written in its most accessible sense and the applications are not very clearly explored. The authors can find below a series of questions and comments addressing this aspect.

1. Applications:

- Authors give several motivations in first paragraph of the paper in the following statement:

verified quantum computation [37] or tasks using entangled states in networks [50], such as authentication of quantum communication [6], anonymous communication [9] or distributed quantum sensing.

Can they provide clear applications that work for this motivation? We strongly recommend providing more precise applications for the non-iid framework.

-Verified quantum computation is particularly discussed in Sec 2.3 - but there is a difference between verifying quantum computation and verifying pure quantum state. In what sense are the authors thinking of an application here? Is there a Kitaev's history state being used here?

- The recent work of Huang, Preskill, Soleimanifar gives conditionally efficient verification algorithm when one only has access to amplitudes of the state ψ (that is to be verified). Do the techniques in this submission apply here too? We understand that Huang, Preskill, Soleimanifar's work comes after the current submission.

2. Technique: the main technique is that of quantum de-finetti theorem, in particular the formalism due to Brandao and Harrow (BH).

- Generally, BH does not require permutation invariance in the state (since its mostly based on classical measurements followed by chain rule of CMI, which is general - see Section 2.3 in the related paper <https://arxiv.org/pdf/1310.0017.pdf>). Why is permutation invariance assumed in the current work?

- The difference between Thm 2.2 and Equation IF is that the former does not need mixture of product states. It's not clear why that is useful - in other words, why didn't Equation IF already work for the authors. If a learning algorithm works well for all product states, shouldn't it work for mixtures of product states due to convexity of some relevant quantity?

- Is it possible to extend Thm 2.2 to measurements that are adaptive but non-entangling?

- In Theorem 4.8, why not set $k < \sqrt{N}$? The bound seems trivial if the RHS is > 1 .

- In Theorem 4.8, the role of r is quite unclear. It seems to specify the choice of the POVMs being performed. Why does r have to come from a permutation invariant distribution? It goes back to the previous question about why permutation invariance is needed in the first place.

3. Presentation

- Section 4.3 is a little difficult to read and needs to be significantly clarified. Here are some questions that come to one's mind after a few reads: Can the algorithm 1 be described in words? Why is so much randomization necessary in this algorithm? what is k_A vs k (will be clarified once the algorithm is described in words)?

Some of the complexity of Algorithm 1 comes from Thm 4.8 in Sec 4.2. There are too many random variables to worry about. If the authors can simplify some of it, would be great.

- Calibration information is an unclear aspect. It seems to be put in place to have a general theorem - but note that 'less is more'. It might be worth moving this notion to supplementary info, if it's not a crucial concept. As an example of the confusion, consider example 3 of Page 12. The string x could easily be the prediction and one could defer the job of deriving $|x|$ from x to the learner. Then why is the distinction between c and p so much emphasized?

- Example 4 could fit better in the supplementary part. For those who know the BH proof, thm 4.8's proof is clear enough. For those who don't know the BH proof, Example 4 is complex enough to not help.

I don't think your claim that you don't need permutation invariance of the state is correct. In particular, one step in the new version of Theorem 4.7 seems unjustified: How did you deduce the last line from the second-to-last line in Eq (6)?

The statement of Theorem 4.7 itself seems counterintuitive. Let me provide an easy counterexample to Theorem 4.7: Consider $k = d$ and let N be large enough so that your assumption $1 \leq k < \sqrt{\frac{N}{4 \log(d)}}$ is satisfied. For concreteness, let's say $N = d^4$. Let the state be the product state

$$\rho^{A_1 \dots A_N} = |1\rangle\langle 1|^{\otimes d^3} \otimes |2\rangle\langle 2|^{\otimes d^3} \otimes \dots \otimes |d\rangle\langle d|^{\otimes d^3}.$$

Let $\{\Lambda_r\}_{r \in \mathcal{R}}$ consist of a single measurement channel $\{\Lambda\}$ (i.e., \mathcal{R} has cardinality one) chosen to be the standard basis measurement, i.e.,

$$\Lambda(\sigma) = \langle 1|\sigma|1\rangle|1\rangle\langle 1| + \dots + \langle d|\sigma|d\rangle|d\rangle\langle d|.$$

There is no need to specify q^N in this case since it is a probability measure on a set of a single element. Now, no matter what values the random permutation $\mathbf{j} = (j_1, \dots, j_N)$ and the random number $l \in \{k+1, \dots, k + \frac{N}{2}\}$ take, if you measure the systems $A_{j_{l+1}}, \dots, A_{j_{k+N/2}}$ using the measurement Λ , you always get a single possible outcome \mathbf{w} (which depends on \mathbf{j} and l but there is no other randomness). Moreover, since ρ is a product state, the post-measurement state of the remaining systems is simply the reduced state of ρ on those systems. Hence the states $\rho_{l,r,\mathbf{w}}^{A_{j_1} \dots A_{j_{k+1}}}$ and $\rho_{l,r,\mathbf{w}}^{A_{j_N}}$ in your notation are simply $\rho^{A_{j_1} \dots A_{j_{k+1}}}$ and $\rho^{A_{j_N}}$, respectively. Therefore the left-hand side of your claimed inequality reduces to

$$\text{LHS} = \mathbb{E}_{\mathbf{j}} \left[\left\| \left(\text{id} \otimes \Lambda^{\otimes k} \right) \left(\rho^{A_{j_1} \dots A_{j_{k+1}}} - (\rho^{A_{j_N}})^{\otimes k+1} \right) \right\|_1 \right].$$

Furthermore, since each system of ρ is already diagonal in the standard basis, the channel Λ does nothing to it:

$$\text{LHS} = \mathbb{E}_{\mathbf{j}} \left[\left\| \rho^{A_{j_1} \dots A_{j_{k+1}}} - (\rho^{A_{j_N}})^{\otimes k+1} \right\|_1 \right].$$

Obviously, $\left\| \rho^{A_{j_1} \dots A_{j_{k+1}}} - (\rho^{A_{j_N}})^{\otimes k+1} \right\|_1 = 2$ whenever any one of $\rho^{A_{j_1}}, \dots, \rho^{A_{j_{k+1}}}$ is different from $\rho^{A_{j_N}}$. One can then calculate that

$$\begin{aligned} \text{LHS} &= 2 \times \left(1 - \Pr_{\mathbf{j}} \left[\rho^{A_{j_1}} = \dots = \rho^{A_{j_{k+1}}} = \rho^{A_{j_N}} \right] \right) \\ &= 2 \times \left(1 - \frac{d^3 - 1}{d^4 - 1} \times \frac{d^3 - 2}{d^4 - 2} \times \dots \times \frac{d^3 - d - 1}{d^4 - d - 1} \right) \\ &> 2 \times \left(1 - \frac{1}{d} \times \frac{1}{d} \times \dots \times \frac{1}{d} \right) \\ &\approx 2 \end{aligned}$$

On the other hand, the right-hand side of your claimed inequality is

$$\text{RHS} = \sqrt{\frac{4 \log(d)}{d^2}} \rightarrow 0 \text{ as } d \rightarrow \infty.$$